# Effectiveness of Distributed Gradient Descent with Local Steps for Overparameterized Models

## Abstract

In distributed training of machine learning models, gradient descent with *local iterative steps*, commonly known as Local (Stochastic) Gradient Descent (Local-(S)GD) or Federated averaging (FedAvg), is a very popular method to mitigate communication burden. In this method, gradient steps based on local datasets are taken independently in distributed compute nodes to update the local models, which are then aggregated intermittently. In the interpolation regime, Local-GD can converge to zero training loss. However, with many potential solutions corresponding to zero training loss, it is not known which solution Local-GD converges to. In this work we answer this question by analyzing implicit bias of Local-GD for classification tasks with *linearly separable data*. For the interpolation regime, our analysis shows that the aggregated global model obtained from Local-GD, with *arbitrary number* of local steps, converges exactly to the model that would be obtained if all data were in one place (centralized model) "in direction". We also obtain the same implicit bias for exactly solved local problems with a modified aggregation. Our analysis provides a new view to understand why Local-GD can still perform well with a very large number of local steps even for heterogeneous data. Lastly, we also discuss the extension of our results to Local-SGD and non-separable data.

## 1 Introduction

In this era of large machine learning models, distributed training is an essential part of machine learning pipelines. It can happen in a data center with thousands of connected compute nodes (Sergeev & Del Balso, 2018; Huang et al., 2019), or across several data centers and millions of mobile devices in federated learning (Konečný et al., 2016; Kairouz et al., 2019). In such a network, the communication cost is usually the bottleneck in the whole system. To alleviate the communication burden, and also to preserve privacy to some extent, one common strategy is to perform multiple local updates before sending the information to other nodes, which is called Local Gradient Descent (Local-GD) (McMahan et al., 2017; Stich, 2019; Lin et al., 2019). In a network with $M$ compute nodes, the goal is to train a global model to fit the distributed datasets:

$$\min_{w \in \mathbb{R}^d} f(w) \qquad \text{with } f(w) \equiv \frac{1}{M} \sum_{i=1}^{M} f_i(w), \tag{1}$$

where $w \in \mathbb{R}^d$ is the single model to be trained and $f_i(w)$ is the local loss function for $i^{th}$ compute node. The local loss $f_i(w)$ is the average of the loss function evaluated at model $w$ for the high-dimensional samples and their corresponding labels, $\{x_s, y_s\}_{s \in S_i}$, where $S_i$ is the local dataset, and $N_i = |S_i|$ is the number of local samples. The samples of the local dataset are obtained iid from the local distribution $D_i$.

In each round of Local-GD, a central node sends its current model, referred to as the **global model**, to all compute nodes. Each compute node runs $L$ local gradient descent steps on the global model using its loss $f_i$ on this model to obtain a local model. Each compute node sends its local model back to the central node, where these local models are aggregated, by averaging, to obtain the global model for the next round. The detailed algorithm of Local-GD is described in Algorithm 1.

In modern machine learning, most deep neural networks, where Local-GD has impressive performance, operate in the *overparameterized regime*, where the dimension $d$ of the model is more than the total number of samples $MN$. In this case, there are multiple solutions corresponding to zero training loss. The main question here is:

*Q: Which solution would the aggregated model trained by Local-GD converge to?*

**Contributions.** In this work, we answer this question by analyzing *implicit bias* of Local-GD on classification tasks for linearly separable data. From the implicit bias of Local-GD, we can characterize the dynamics of the global model across rounds. We compare the global model with the ***centralized model*** obtained from running gradient descent (GD) on a dataset consisting of all distributed datasets as if all these datasets were located on the central node. The centralized model is obtained from existing results for the implicit bias of linearly separable data (Soudry et al., 2018). Note that these results cannot be directly applied to Local-GD. For globally linearly separable dataset, we show that the global model converges to the centralized model with any arbitrary number of local steps on heterogeneous data under $\mathcal{O}(1/L)$ learning rate. This learning rate is common in existing analyses of distributed learning (Karimireddy et al., 2020; Koloskova et al., 2020; Crawshaw et al., 2025b). The meaning of this work lies in: 1). providing a theoretical explanation to the phenomenon that Local-GD can work well with a very large number of local steps in practice; 2). showing which solution with zero training loss the Local-GD would converge to. When Local-SGD is considered to be an algorithm that samples a mini-batch of local dataset without replacement at each local step (see Assumption 4), we can obtain the same implicit bias result as Local-GD (see Theorem 5) since each local batch is still a subset of global dataset.

We also consider a special case where each local problem with a weakly regularized term is exactly solved. This setting *simulates* the behavior of Local-GD with a very large number of local steps. With a Modified Aggregation, (see Section 4.4 we can guarantee that the global model can converge to the centralized model. This result provides the implicit bias of massive local updates without the restrictive learning rate of $\mathcal{O}(1/L)$.

**Practical Implications.** In practical implementation of distributed training on large models, the performance of Local-GD is surprisingly good even with heterogeneous data distribution (McMahan et al., 2017; Charles et al., 2021). Also, the number of local steps can be very large in Local-GD type algorithms and real-world systems, for example, up to 500 local steps in distributed training of large language models (LLM) (Douillard et al., 2023; Jaghouar et al., 2024). Since our results show the Local-GD can converge to centralized model with *arbitrary* number of local steps, it helps explain why Local-GD can still work well with a large number of local steps in practice. In this work we consider linear models as an appropriate starting point to investigate the implicit bias of Local-GD. A popular example of linear models used in practical machine learning pipelines is fine-tuning last layer on pretrained large models or adding linear layers in transfer learning (Donahue et al., 2014; Kornblith et al., 2019) and deployment of LLM (Devlin, 2018; Jiang et al., 2020). Thus we also add an experiment of fine-tuning last layer of neural network to show broader impact of our analysis.

## 1.1 Related Works

**Convergence of Local-GD.** When the data distribution is homogeneous, several works analyze convergence of Local (Stochastic) GD (Stich, 2019; Yu et al., 2019; Khaled et al., 2020). With an "appropriately small" number of local steps, the dominating convergence rate is not affected. Various assumptions have been made to handle data heterogeneity and develop convergence analysis (Li et al., 2020b; Karimireddy et al., 2020; Khaled et al., 2020; Reddi et al., 2021; Wang et al., 2020; Crawshaw et al., 2023). For strongly convex and smooth loss functions, the number of local steps should not be larger than $\mathcal{O}(\sqrt{T})$ for i.i.d data (Stich, 2019) and non-i.i.d. data (Li et al., 2020b). However, in practice Local-GD (FedAvg) works well in many applications (McMahan et al., 2017; Charles et al., 2021), even in training large language models (Douillard et al., 2023; Jaghouar et al., 2024). In Wang et al. (2024), the authors argue that existing theoretical assumption does not align with practice and proposed a client consensus hypothesis to explain the effectiveness of FedAvg in heterogeneous data. But they do not consider the impact of overparameterization on distributed training. There are some works incorporating the property of zero training loss of overparameterized neural networks into the conventional convergence analysis of FedAvg (Huang et al., 2021; Deng et al., 2022a; Song et al., 2023; Qin et al., 2022). However, they do not guarantee which model FedAvg can converge to, which is especially important for overparameterized set ups since there are multiple solutions with zero training loss. Our work is different from these works as: 1. We analyze which point the Local-GD can converge to, which is a more elementary problem before obtaining the convergence rate; 2. We use implicit bias as a technical tool to analyze the overparameterized FL. Woodworth et al. (2020b) provides a lower bound of Local-GD for convex and smooth functions, however,

the lower bounds requires an upper bound on the $\ell_2$ norm of the optimum. For logistic regression, the optimum is achieved at $\|w^\star\|_2 \to \infty$, hence the bounds do not apply to logistic regression. Further, in the overparameterized setting, all the data points can share an optimum, so the terms of noise at optimum is zero in Woodworth et al. (2020b). This further makes their bounds invalid for the problem of overparameterized logistic regression.

**Implicit Bias.** Soudry et al. (2018) is the first work to show the gradient descent converges to a max-margin direction on linearly separable data with a linear model and exponentially-tailed loss function. Ji & Telgarsky (2019a) has provided an alternative analysis and extended this to non-separable data. The theory of implicit bias has been further developed, for example, for wide two-layer neural networks (Chizat & Bach, 2020), deep linear models (Ji & Telgarsky, 2019b), linear convolutional networks (Gunasekar et al., 2018b), two-layer ReLU networks (Kou et al., 2024) etc. Beyond gradient descent, more algorithms have been considered, including gradient descent with momentum (Gunasekar et al., 2018a), SGD (Nacson et al., 2019), Adam (Cattaneo et al., 2023), AdamW (Xie & Li, 2024). Recently, implicit bias has also been used to characterize the dynamics of continual learning, on linear regression (Evron et al., 2022; Goldfarb & Hand, 2023; Lin et al., 2023), and linear classification (Evron et al., 2023; Jung et al., 2025). In Evron et al. (2023), gradient descent on continually learned tasks is related to Projections onto Convex Sets (POCS) and shown to converge to a *sequential* max-margin scheme. In our work we consider the implicit bias of gradient descent in distributed setting, which is related to a different parallel projection scheme by projecting onto constraint sets *simultaneously*.

**Parallel Projection.** Parallel projection methods are a family of algorithms to find a common point across multiple constraint sets by projecting onto these sets in parallel. These methods are widely used in feasibility problems in signal processing and image reconstruction (Bauschke & Combettes, 2011). The straightforward average of multiple projections is known as the simultaneous iterative reconstruction technique (SIRT) in Gilbert (1972). Then de Pierro & Iusem (1984) studied the convergence of PPM for a relaxed version, and Combettes (1994) further generalized the result to inconsistent feasibility problems. In Combettes (1997), an extrapolated parallel projection method was proposed to accelerate the convergence. We note that Jhunjhunwala et al. (2023) used this extrapolation to accelerate FedAvg. However, it was just inspired by the similarity between parallel projection method and FedAvg, while in this work we rigorously prove the relation between PPM and FedAvg using implicit bias of gradient descent.

---

**Algorithm 1** LOCAL-GD.

---
1: **Input:** learning rate $\eta$.
2: Initialize $w_0^0$
3: **for** $k=0$ to $K-1$ **do**
4:     The aggregator sends global model $w_0^k$ to all compute nodes.
5:     **for** $i=1$ to $i=M$ **do**
6:         compute node $i$ updates local model starting from $w_0^k$: $w_i^{k,0}=w_0^k$.
7:         **for** $l=0$ to $L-1$ **do**
8:             $w_i^{k,l+1}=w_i^{k,l}-\eta\nabla f_i(w_i^{k,l})$.
9:         **end for**
10:         compute node $i$ sends back the updated local model $w_i^{k+1}=w_i^{k,L}$.
11:     **end for**
12:     The aggregator aggregates all the local models: $w_0^{k+1}=\frac{1}{M}\sum_{i=1}^M w_i^{k+1}$.
13: **end for**
14: **Output:** $w_0^K$.

---

## 2 Motivating Observation in Linear Regression

In this section we first give some observations in linear regression as a motivating example. The behavior of linear regression is very well-understood in high-dimensional statistics.

**Setting:** At each compute node $i$, the dataset $S_i$ consists of $N$ tuples of samples and their corresponding labels, $(x,y) \in \mathbb{R}^d \times \mathbb{R}$. Denote $X_i = [x_{i1}, x_{i2}, ..., x_{iN}]^T \in \mathbb{R}^{N \times d}$ as the data matrix at $i$-th compute node, and $y_i = [y_{i1}, y_{i2}, ..., y_{iN}] \in \mathbb{R}^N$ as the label vector. Let $X_c = [X_1^T, ..., X_M^T]^T \in \mathbb{R}^{MN \times d}$ be the data matrix consisting of all the local data, and $y_c = [y_1^T, ..., y_M^T]^T \in \mathbb{R}^{MN \times 1}$ be the label vector consisting of the local labels.

We consider a special case of Local-GD in Algorithm 1 where the number of local steps is very large. At each round, the aggregator sends the global model $w_0$ to all the compute nodes. Each compute node minimizes the squared loss $f_i(w_i) = \frac{1}{2N}\|y_i - X_i w_i\|^2$ by *a large number* of gradient descent steps *until convergence*. Then each compute node sends back the local model and the aggregator aggregates all the local models to get the updated global model.

**Underparameterized Regime:** When the number of local samples is larger than the dimension $d$, it is known that local model would converge to the ordinary least square solution $w_i^{k+1} = (X_i^T X_i)^{-1} X_i^T y_i$ regardless of initial point $w_i^k$. In the meanwhile, the centralized model with all the training samples is $w_c = (X_c^T X_c)^{-1} X_c^T y_c$. However, the average of local models $w_0 = \sum_{i=1}^M (X_i^T X_i)^{-1} X_i^T y_i$ is not identical to the centralized model unless the data is homogeneously distributed and all $X_i^T X_i$ are proportional. So a large number of local steps can hurt the convergence to centralized model with heterogeneous data distribution.

**Overparameterized Regime:** When the dimension is larger than the number of samples at each compute node $(d > N)$, there are multiple solutions corresponding to zero squared loss. However, it is known that gradient descent would converge to the minimum norm solution in the feasible set, which corresponds to a minimum Euclidean distance to the initial point (Gunasekar et al., 2018a; Evron et al., 2022), i.e., the solution of the optimization problem

$$\min_{w_i} \quad \|w_i - w_0^k\|^2 \quad \text{s.t.} \quad X_i w_i = y_i. \tag{2}$$

We can obtain the closed form solution as $w_i^{k+1} = (I - P_i)w_0^k + X_i^\dagger y_i$, where $P_i \triangleq X_i^T(X_i X_i^T)^{-1} X_i$ and $X_i^\dagger \triangleq X_i^T(X_i X_i^T)^{-1}$. We observe that $P_i$ is the projection operator to the row space of $X_i$, and $X_i^\dagger$ is the pseudo inverse of $X_i$. Meanwhile the centralized model converges to the minimum norm solution $w_c = X_c^T(X_c X_c^T)^{-1} y_c$. Denote $\bar{P} = \frac{1}{M}\sum_{i=1}^M P_i$. In the training process the difference between global model and centralized model is iteratively projected onto the null space of span of row spaces of $X_i$s. It implies that the difference on the span of data matrix gradually decreases until zero. Note that Based on the evolution of the difference, we can prove the following theorem:

**Theorem 1.** *For the linear regression problem, suppose the initial point $w_0^0$ is 0 and $d > MN$, and there exists minimum non-zero eigenvalue of $\bar{P}$ denoted by $\theta_{\min}$. Then the output of Local-GD, $w_0^K$, converges to the centralized solution $w_c$ as the number of communication rounds $K \to \infty$ as $\|w_0^K - w_c\| \le (1 - \theta_{\min})^K \|w_c\|$.*

The proof is deferred to Appendix B. The key step is to show the initial difference is already in the data space, and no residual in the null space of row spaces of $X_i$s. The convergence to the centralized model is at exponential rate. Due to the linearity of the regression problem, we can theoretically show the global model can exactly converge to the centralized model with implicit bias on overparameterized regime. It implies that, even if we use a large number of local steps to exactly solve the local problems on very heterogeneous data, the performance of Local-GD is equivalent to train a model with all the data in one place.

## 3 Implicit Bias of Local-GD for Classification

For classification task, we also would like to know whether the global model can converge to the centralized model with any number of local steps. Now we investigate a binary classification task with linear models.

### 3.1 Setting

Suppose, for each compute node $i$, the dataset $S_i$ consists of $N_i$ tuples of samples and their corresponding labels, $(x,y) \in \mathbb{R}^d \times \{+1,-1\}$. We denote $X_i \in \mathbb{R}^{N_i \times d}$ as the data matrix at $i$-th compute node, and $y_i \in \{+1,-1\}^{N_i}$ as the label vector. The global dataset is the set of $M$ local datasets $S = \cup_{i=1}^M S_i$.

We consider a linear model $w \in \mathbb{R}^d$ for the binary classification task. The local loss at $i$-th compute node is

$$f_i(w) = \sum_{s \in S_i} g(y_s x_s^T w), \tag{3}$$

where $g(u)$ is a loss function decreasing to zero when $u \to \infty$, such as logistic loss $g(u) = \ln(1 + e^{-u})$.

We study LocalGD with an *arbitrary number* of gradient descent steps. To describe our main results, we have the following notations and assumptions. We denote the whole data matrix as $X \in \mathbb{R}^{N \times d}$, where $N = \sum_{i=1}^M N_i$.

We write $\sigma_{\max} = \sqrt{\theta_{\max}(X^T X)}$ as the maximum singular value of data matrix $X$, where $\theta$ represents eigenvalues of a square matrix. We need an assumption of global separability on whole dataset.

**Assumption 1.** For all the data samples $(x_s, y_s) \in S$, there exists $w \in \mathbb{R}^d$ such that $y_s x_s^T w > 0$.

Note that linear separability is a common assumption in the analysis of learning in overparameterized regime (Nacson et al., 2019; Soudry et al., 2018; Evron et al., 2023). For our distributed case, this implies that all clients share at least 1 minimizer, which imposes an extremely mild condition on the data heterogeneity among clients. In the overparametrized setting, $d \geq mn$, hence, there are likely several such solutions separating the whole dataset. Since there are multiple solutions separating the whole dataset, we define a particular max-margin solution on global dataset:

$$\hat{w} = \arg\min_{w \in \mathbb{R}^d} \|w\| \quad \text{s.t.} \quad y_s x_s^T w \geq 1, \quad \forall s \in S. \tag{4}$$

It has been proven that gradient descent would implicitly lead the linear model to this max-margin solution in direction, i.e., convergence of model direction to $\hat{w}/\|\hat{w}\|$ (Soudry et al., 2018). We define the maximum margin as

$$\gamma = \max_{w \in \mathbb{R}^d, \|w\|=1} \min_s y_s x_s^T w \tag{5}$$

which is strictly positive since the global dataset is linearly separable. The data points reaching this margin are support vectors of the global dataset.

To establish convergence, we require additional regularity assumptions on the loss function.

**Assumption 2.** The loss function $g(u)$ is a positive, differentiable, $\beta$-smooth function, monotonically decreasing to zero, and $\limsup_{u \to -\infty} g' < 0$.

**Assumption 3.** The negative loss derivative $-g'(u)$ has a tight exponential tail. That is, there exists positive constants $\mu_+$, $\mu_-$ and $\bar{u}$ such that $\forall u > \bar{u}$:

$$(1 - \exp(-\mu_- u))e^{-u} \leq -g'(u) \leq (1 + \exp(-\mu_+ u))e^{-u}. \tag{6}$$

Note that these assumptions are also used in centralized learning of overparameterized models (Soudry et al., 2018; Nacson et al., 2019; Evron et al., 2023), and the logistic loss satisfies all the assumptions. With our setting completely defined, we state our main results.

### 3.2 Loss Convergence and Implicit Bias of Local-GD

Our main result is on the asymptotic convergence of the model parameter $w_0$ and loss $f(w)$ for Local-GD.

**Theorem 2.** *Under assumptions 1, 2, 3, if the learning rate satisfies $\eta < \min\left(\frac{1}{2L\sigma_{max}^2 \beta}, \frac{\gamma^2}{4L\sigma_{\max}^3 \beta(\gamma + \sigma_{\max})}\right)$, then for the process of Local-GD, we have,*

- *Every data point is classified correctly finally: $\lim_{k \to \infty} x_s^T w_0^k = \infty, \forall s \in S$.*
- *The global model obtained from Local-GD will behave as*

$$w_0^k = \log(Lk)\hat{w} + \rho^k, \quad \text{and,} \quad \left\| \frac{w_0^k}{\|w_0^k\|} - \frac{\hat{w}}{\|\hat{w}\|} \right\| = O\left(\frac{1}{\log(\eta L k)}\right) \tag{7}$$

  *and $\|\rho^k\| < \infty$ for all $k$. This implies, the normalized global model converges to the global max-margin solution.*
- *The loss function $f(w_0^k)$ decreases to zero asymptotically as $f(w_0^k) = O\left(\frac{1}{\eta L k}\right)$ when $k$ is sufficiently large.*

The proof is deferred to Appendix C. The technical challenges lie in that we need to control the residual term $\rho^k$ with the *local steps* and *aggregations*, which are handled by a refined analysis in distributed context. This theorem implies the global model can eventually correctly classify all the training samples after many rounds of communication. Given that centralized model also converges to the global max-margin solution from prior

results, the global model from Local-GD actually converges to the exact centralized model in direction. Further, this holds for a step size $\eta \propto \frac{1}{L}$, and does not require any additional modifications to the objective, for instance, any regularization on the difference between local and global models during local steps.

**Impact of local steps.** In this analysis, the number of local steps can be arbitrary. Although the magnitude of model vector would diverge to infinity, the direction of aggregated model still converges to the direction of global max-margin solution. Thus, the number of local steps does not influence the asymptotic convergence to the centralized model, which is very different from underparameterized regime. Our loss convergence rate is $\mathcal{O}(1/\eta Lk)$, which matches the rate in Crawshaw et al. (2025a).

**Learning rate.** Theorems 2 needs the learning rate to be small as $\mathcal{O}(1/L)$, which has also been used by existing works (Karimireddy et al., 2020; Koloskova et al., 2020; Crawshaw et al., 2025b) on Local-GD and Local-SGD. This means the model does not move so far after one round of local iterations. In the next section we try to explore whether the global model still converges to max-margin solution with a learning rate independent of $L$ in a special case.

*Remark* 3. The proof of Theorem 2 is inspired by the analysis of implicit bias of SGD (Nacson et al., 2019). Intuitively, we can regard one local dataset as a "batch" in SGD for sampling without replacement. But we perform multiple gradient steps in the same "batch", not just one step of gradient descent. The challenge is to handle local steps in the same local dataset and the aggregation after one round of local training.

*Remark* 4. In this paper we mainly focus on the linearly separable data, which is a standard assumption in implicit bias analysis and also widely used in recent works (Zhang et al., 2024; Crawshaw et al., 2025b; Jung et al., 2025). For non-separable case, Ji & Telgarsky (2019a) has shown gradient descent converges to a ray along the direction of max-margin solution of largest linearly separable subset. However, there is still an assumption on the data: in fact, one needs a positive margin on the separable part of data to show both convergence in risk or parameters. Nevertheless, Ji & Telgarsky (2019a) clearly shows strict linear separability is not the main reason for the convergence of gradient descent to a max-margin solution. Since even without this assumption, GD still converges to a variant form of max-margin solution. It is possible to use the same idea in Local-GD. Intuitively, in the case where local datasets are linearly separable but global dataset is non-separable, although local training would guide local models to local max-margin solutions, the aggregations would force the global model to converge to the max-margin solution of largest linearly separable subset of global dataset, which is the centralized solution.

### 3.3 Extension to Local-SGD

It is straightforward to extend our analysis of Local-GD to Local-SGD that chooses samples without replacement. At each local step of $i$-th compute node, the update is $w_i^{k,l+1} = w_i^{k,l} - \eta \sum_{s \in \mathcal{B}_{i,l}} \nabla g(y_s x_s^T w_i^{k,l})$, where $\mathcal{B}_{i,l}$ is the mini-batch of samples at $l$-th local step. We consider the following setting of sampling:

**Assumption 4** (Sampling without replacement.)**.** At every communication round, each compute node run stochastic gradient descent with $E$ epochs, where $E$ is an positive integer. Within each epoch, the mini-batches with batch size $B$ are $\{S_{i,0}, S_{i,1}, ..., S_{i,l'}\}$, which partition the local dataset $S_i$, where $l' = N/B$ is the number of local steps for one epoch.

Under this setting, each sample is exactly chosen once inside one epoch of local updates. At each round, the local datasets are passed $E$ times, which is a practically common way. Under this assumption, the number of local steps is $L = E \cdot L'$. We give a formal description of Local-SGD as Algorithm 2.

As remarked, the proof of Local-GD (Theorem 2) is inspired by the implicit bias analysis of SGD. Thus we can extend it to Local-SGD without significant changes. In Local-GD we regard one local dataset as a large "batch" in SGD for sampling without replacement. Instead of performing multiple steps on the same "batch" at one compute node, Local-SGD performs multiple steps on mini-batches of this larger "batch". At each step, the stochastic gradient contains less samples than a full gradient, but those samples are still a subset of the local "batch". After slight modification we can obtain the same asymptotic results as Theorem 2 for Local-SGD.

**Theorem 5.** *Under assumptions 1, 2, 3, 4, if the learning rate satisfies* $\eta \leq \min\left(\frac{1}{2L\sigma_{max}^2 \beta}, \frac{\gamma^2}{4L\sigma_{\max}^3 \beta(\gamma + \sigma_{\max})}\right)$, *then for the process of Local-SGD, we have,*

- *Every data point is classified correctly finally:* $\lim_{k \to \infty} x_s^T w_0^k = \infty, \forall s \in S$.

---

**Algorithm 2** LOCAL-SGD.

---

1: **Input:** learning rate $\eta$.
2: Initialize $w_0^0$
3: **for** $k=0$ to $K-1$ **do**
4:    The aggregator sends global model $w_0^k$ to all compute nodes.
5:    **for** $i=1$ to $i=M$ **do**
6:       compute node $i$ updates local model starting from $w_0^k$: $w_i^{k,0}=w_0^k$.
7:       **for** $l=0$ to $L-1$ **do**
8:          Choose a mini-batch of samples $\mathcal{B}_l$ and calculate the gradient: $G(w_i^{k,l})=\sum_{s\in\mathcal{B}_l}g'(x_s^T w_i^{k,l})x_s$
9:          $w_i^{k,l+1}=w_i^{k,l}-\eta G(w_i^{k,l})$.
10:       **end for**
11:       compute node $i$ sends back the updated local model $w_i^{k+1}=w_i^{k,L}$.
12:    **end for**
13:    The aggregator aggregates all the local models: $w_0^{k+1}=\frac{1}{M}\sum_{i=1}^M w_i^{k+1}$.
14: **end for**
15: **Output:** $w_0^K$.

---

- *The global model obtained from Local-GD will behave as*

$$w_0^k=\log(Lk)\hat{w}+\rho^k, \quad and, \quad \left\|\frac{w_0^k}{\|w_0^k\|}-\frac{\hat{w}}{\|\hat{w}\|}\right\|=O\left(\frac{1}{\log(\eta Lk)}\right) \tag{8}$$

   *and $\|\rho^k\|<\infty$ for all $k$. This implies, the normalized global model converges to the global max-margin solution.*

- *The loss function $f(w_0^k)$ decreases to zero asymptotically as $f(w_0^k)=O\left(\frac{1}{\eta Lk}\right)$ when $k$ is sufficiently large.*

The proof is deferred to Appendix D. The main difference from proofs of Theorem 2 is on the proof of Claim 1. We can see the Local-SGD does not change the asymptotic property of local steps with increasing number of rounds. This result aligns with Nacson et al. (2019), which obtains the same implicit bias and convergence rate of SGD compared to GD. Choosing a mini-batch for one step of update does not change the implicit bias compared to choosing a full batch at every update.

## 4    Implicit Bias of Exactly-Solved Local Problems

### 4.1   Setting

In this section, we consider a special case, where we aim to solve a local optimization problem with exponential loss and a weakly regularized term for each compute node. The local problem is solved exactly (to reach the local optima) with a large number of local steps. This setting is an analogy of Local-GD with massive local steps and the learning rate is independent of $L$.

**Algorithm.** At each round, the aggregator sends the global model $w_0$ to all the compute nodes. Each compute node minimizes an *exponential loss* with a *weakly regularized term* by many gradient descent steps *until convergence*. That is, each compute node solves the following problem:

$$\min_{w\in\mathbb{R}^d} f_i(w) \qquad \text{where } f_i(w)\equiv\sum_{s\in S_i}\exp\left(-y_s x_s^T w\right)+\frac{\lambda}{2}\|w-w_0^k\|^2 \tag{9}$$

where $\lambda$ is a regularization parameter close to 0.

Then each compute node sends back the local model and the aggregator aggregates all the local models to get the updated global model (i.e., they follow Algorithm 1 with $f_i(w_i)$ as specified here).

Regularization methods are very common in distributed learning to force the local models move not too far from global model (Li et al., 2020a; 2021; T Dinh et al., 2020). Here we consider the weakly regularized term, $\lambda\to0$, to give

theoretical insights of Local-GD on classification tasks. Experimentally the $\lambda$ is set to be extremely small that does not affect the minimization of exponential loss. For the local loss functions, we have one assumption on smoothness:

**Assumption 5.** *For each compute node, the local loss function $f_i(w)$ is $B$-smooth for any round of local steps $k$.*

**Learning Rate.** In the following analysis of implicit bias, we actually exploit the property of local minimizers. Since local problem (9) is a strongly convex problem for $\lambda > 0$, we can run local gradient descent to find the unique minimizer with a learning rate $\eta \leq \frac{2}{B}$ for a large number of local steps $L$. That's the only requirement of learning rate, which is not dependent of number of local steps $L$. In other words, the learning rate is only needed to be sufficiently small to ensure local convergence at each round.

## 4.2 Implicit Bias of Exactly-Solved Local Problems and Relation to Parallel Projection Method

We consider the whole algorithmic process on classification and use another auxiliary sequence of global models, denoted as $\bar{w}_0^k, k = 0,1,2,....$ Starting from an initial point $\bar{w}_0^0$, the central node sends global model $\bar{w}_0^k$ to all the compute nodes at $k$-th iteration round. Each compute node solves the following *Local Max-Margin* problem to obtain $\bar{w}_i^{k+1}$:

$$\bar{w}_i^{k+1} = \arg\min_{w \in \mathbb{R}^d} \|w - \bar{w}_0^k\| \quad \text{s.t.} \quad y_s x_s^T w \geq 1, \quad \forall s \in S_i. \tag{10}$$

Then the compute node sends the local model back. The central node averages the local models to get $\bar{w}_0^{k+1} = \frac{1}{M}\sum_{i=1}^{M}\bar{w}_i^{k+1}$. We can show the solution $w_0^K$ converges in direction to the global model from Local Max-Margin problems $\bar{w}_0^K$.

**Lemma 1.** *For almost all datasets sampled from a continuous distribution satisfying Assumption 1, with initialization $w_0^0 = \bar{w}_0^0 = 0$, we have $w_0^k \to \ln\left(\frac{1}{\lambda}\right)\bar{w}_0^k$, and the residual $\|w_0^k - \ln\left(\frac{1}{\lambda}\right)\bar{w}_0^k\| = \mathcal{O}(k\ln\ln\frac{1}{\lambda})$, as $\lambda \to 0$. It implies that at any round $k = o\left(\frac{\ln(1/\lambda)}{\ln\ln(1/\lambda)}\right)$, $w_0^k$ converges in direction to $\bar{w}_0^k$:*

$$\lim_{\lambda \to 0} \frac{w_0^k}{\|w_0^k\|} = \frac{\bar{w}_0^k}{\|\bar{w}_0^k\|}. \tag{11}$$

The proof is deferred in Appendix E. The proof sketch is similar to the continual learning work Evron et al. (2023), but we have to handle the parallel local updates for each dataset from the same initial model and the aggregation, which is different from the sequential updates where for each dataset the model is trained from the previous model and there is no need to do aggregation.

Based on this equivalence between Exactly-solved Local Problems for linear classification and Local Max-Margin scheme, we can further analyze the performance of distributed training with a large number of local steps. Instead of a closed-form solution for the Local Max-Margin problem (10), we treat it as a projection of the aggregated global model onto a convex set $C_i$: $\bar{w}_i^{k+1} = P_i(\bar{w}_0^k)$, which is formed by the constraints in (10) and exactly the local feasible set defined in Assumption 1. Here we slightly overload the notation $P_i$, which was used as the projection matrix in linear regression since the readers can get a sense of the same effect of them. The aggregation is actually to average the local projected points: $\bar{w}_0^{k+1} = \frac{1}{M}\sum_{i=1}^{M}P_i(\bar{w}_0^k)$.

The sequence of Local Max-Margin schemes is therefore projections to local (convex) feasible sets followed by aggregation, which is the Parallel Projection Method (PPM) in literature (Gilbert, 1972; Combettes, 1994). Using Lemma 1, we establish the relation between Exactly-solved Local Problems and PPM: the model from Exactly-solved Local Problems converges to the model from PPM in direction.

## 4.3 Convergence to Global Feasible Set

Now we use the properties of PPM to characterize the performance of Exactly-solved Local Problems in classification. In Combettes (1994), the convergence of PPM has been provided for a relaxed version. The direct average considered in this work can be seen as a special case of the relaxed version, and the following lemma holds.

**Lemma 2** (Theorem 1 and Proposition 8, (Combettes, 1994))**.** *Suppose all the local feasible sets $C_i, i = 1,2,...$ are closed and convex, and the intersection $\bar{C}$ is not empty. Then for any initial point $\bar{w}_0^0$, the global model $\bar{w}_0$ generated by PPM converges to a point in the global feasible set $\bar{C}$.*

This lemma guarantees that $\bar{w}_0^K$ will converge to the intersection of the convex sets after many rounds of iteration, however we are not sure which exact point it would converge to.

Combining Lemma 1, Lemma 2 and the fact that centralized model would converge to the minimum norm solution in global feasible set, we immediately have:

**Theorem 6.** *For linear classification problem with exponential loss, suppose initial point is $w_0^0 = 0$. The aggregated global model $w_0^K$ obtained by Local-GD with a large number of local steps converges in direction to one point in the global feasible set $\bar{C}$, while the centralized model converges in direction to the minimum norm point in the same set.*

Here we cannot guarantee the global model obtained by Exactly-solved Local Problems to converge exactly to the centralized model in classification, but show that it converges to the same global feasible set as the centralized solution. To theoretically support that the aggregated model converges to the centralized model, we propose a slightly Modified Aggregation by just changing the aggregation method, and showing that it converges to the centralized model exactly.

### 4.4 Modified Aggregation: Convergence to Centralized Model

In Combettes (1996) it was shown that if the aggregation method is modified to incorporate the influence of the initial point $\bar{w}_0^0$ in PPM, then the sequence generated by PPM will converge to a specific point in global feasible set $\bar{C}$ with minimum distance to this initial point. Denote $P_c(\cdot)$ as the projection operator onto the global feasible set $\bar{C}$. Formally we have the following lemma.

**Lemma 3** (Theorem 5.3, (Combettes, 1996)). *Suppose $\bar{C}$ is not empty. For any initial point $\bar{w}_0^0$, when the local models are aggregated as*

$$\bar{w}_0^{k+1} = (1-\alpha^{k+1})\bar{w}_0^0 + \alpha^{k+1}\left(\frac{1}{M}\sum_{i=1}^{M} P_i(\bar{w}_0^k)\right), \tag{12}$$

*where $\{\alpha^k\}$ satisfy $(i)\lim_{k\to\infty}\alpha^k = 1, (ii)\sum_{k\geq 0}(1-\alpha^k) = \infty, (iii)\sum_{k\geq 0}|\alpha^{k+1} - \alpha^k| < \infty$, then the global model generated by PPM will converge to the point $P_c(\bar{w}_0^0)$.*

The sequence generated by PPM would converge to the point in global feasible set, $\bar{C}$, with minimum distance to $\bar{w}_0^0$. The modified aggregation method is a linear combination of initial point and current average of local projected points. One example of the sequence $\{\alpha^k\}$ satisfying the conditions is $\alpha^k = 1 - \frac{1}{k+1}$.

If we start from $\bar{w}_0^0 = 0$, then the point $P_c(\bar{w}_0^0)$ is exactly the minimum norm point in the global feasible set. It shows the PPM can exactly converge to the minimum norm point as the centralized model. Based on this result, we propose a Modified Aggregation, with the replacement of Line 9 in Algorithm 1 with

$$w_0^{k+1} = (1-\alpha^k)w_0^0 + \alpha^k\left(\frac{1}{M}\sum_{i=1}^{M} w_i^k\right). \tag{13}$$

We still need to prove a lemma analogous to Lemma 1 to establish the equivalence between Modified Aggregation and Modified PPM, which is omitted here due to space limit (Please refer to Appendix E and the proof is very similar to proof in Lemma 1). From the equivalence, Lemma 3, and implicit bias of the centralized model, we can have the following theorem:

**Theorem 7.** *For linear classification problem with local loss (9), suppose the initial point is $w_0^0 = 0$. Then the global model $w_0^K$ obtained by Modified Local-GD converges in direction to the centralized model obtained from (4).*

Unlike the vanilla Local-GD, which is only guaranteed to converge to the global feasible set, the Modified Aggregation is guaranteed to converge to the centralized model in direction. Note that if we start from $\bar{w}_0^0 = 0$, the aggregation in Modified Aggregation becomes $w_0^{k+1} = \frac{k}{k+1}\left(\frac{1}{M}\sum_{i=1}^{M} w_i^k\right)$, which is just a *scaling* of vanilla aggregation with a parameter less than 1. Thus we can see experimentally they usually converge to the same point and Modified Aggregation converges slightly slower. With Modified Aggregation, we can theoretically show the global model still converges to centralized model in direction with a learning rate independent of $L$.

## 5 Discussions and Comparisons

In this section we further discuss the insights of our theoretical results and its comparison to relevant works. Firstly, we would like to emphasize that the core question of this work is which solution the aggregated model trained by Local-GD would converge to in *overparameterized* regime. There are several models achieving zero training loss, but only one of which is the centralized model. To motivate the discussion below, we give a concrete example where the average of local max-margin solutions is not the global max-margin solution.

**A counter-example in overpameterized linear classification.** Consider the case where dimension is $d > 4$. Suppose $e_1$ and $e_2$ are unit vectors with only 1 non-zero entry. For compute node 1, the data samples are $(x = e_1, y = 1)$ and $(x = -e_1, y = -1)$. For compute node 2, the data samples are $(x = 0.5(e_1 + e_2), y = 1)$ and $x = (-0.5(e_1 + e_2), y = -1)$. Thus the max-margin unit-norm classifier of compute node 1 is $e_1$, and that of compute node 2 is $(e_1 + e_2)/\sqrt{2}$. Their average is not equal to the max-margin classifier of the centralized solution, which is $(e_1 + e_2)/\sqrt{2}$, as the centralized solution's support vectors are only from compute node 2 and not from compute node 1.

It implies the implicit bias analysis of Local-GD is still non-trivial with small learning rate. The $\mathcal{O}(1/L)$ learning rate makes every round to be seen as a "large GD step", thus the local model is away from the local max-margin solution at each round. After sufficient aggregations, the final aggregated model converges to the centralized max-margin solution. With a constant larger learning rate, the local model can approach closer to the local max-margin solution with massive local steps, but the average of local max-margin solutions is normally not the global max-margin solution. Below we further discuss the intuitions and consequences of using $\mathcal{O}(1/L)$ learning rate and larger learning rate.

$\mathcal{O}(1/L)$ **learning rate.** In many previous analysis of Local-GD, the $\mathcal{O}(1/L)$ learning rate is needed to ensure global function to decrease every round. Then the loss converges to optimal value in the underparameterized setting. In overparameterized setting, we still require $\mathcal{O}(1/L)$ learning rate to make global function decrease after every round as a round-level descent lemma (see (48) in Appendix C.1), and then derive the implicit bias results similar to centralized GD. But the price is convergence rate of global loss is sub-optimal as $\mathcal{O}(1/k)$ with small learning rate. Some works have shown increasing local steps $L$ does not improve worst-case amount of communication for smooth, convex optimization (Woodworth et al., 2020a, Theorem 5),(Koloskova et al., 2020, Theorem 6).

**Larger learning rate.** For the specific problem of distributed logistic regression, Crawshaw et al. (2025b, Corollary 3) shows that a two-stage Local-GD algorithm can improve this worst-case bound and Crawshaw et al. (2025a) improves their results with vanilla Local-GD and constant step sizes. However, the first stage of Crawshaw et al. (2025b) still requires $\eta = \mathcal{O}(\frac{1}{L})$. On the other hand, both Crawshaw et al. (2025b) and Crawshaw et al. (2025a) are attempts to extend the analysis of GD on logistic regression with large step sizes. Then the global function is not guaranteed to decrease every round and the loss oscillates at the edge of stability phase. The price is several warm-up rounds to handle the initial instability, where the number of warm-up rounds is proportional to the learning rate. However, they can only show that the loss converges, but not the solution the model converges to in overparameterized setting since there are many solutions with zero loss. In this work we considered a special case where each local problem with exponential loss is exactly solved. It is not vanilla Local-GD algorithm but it includes the case of infinite local steps with larger step size independent of $L$. We conclude with exactly solved local problem, the vanilla average aggregation is not guaranteed to converge to centralized model. Only with the proposed Modified Aggregation we can ensure the same implicit bias result as centralized GD. This is non-trivial especially considering the gradient descent can diverge catastrophically under the exponential loss in the edge of stability regime where the learning rates are large (Theorem 4.2 in Wu et al. (2023)). Finally, it is a promising future direction to explore whether we can incorporate the implicit bias of large learning rate with logistic loss into Local-GD and obtain faster convergence rate.

**Comparison to other relevant works.** Another line of work (Gu et al., 2023; 2024) approximates Local-Stochastic Gradient Descent (Local-SGD) by an SDE to obtain an appropriate scaling between $L$ and $\eta$. Note that our analysis is exact for both Local-GD and Local-SGD, for finite $\eta$. We also extend our results to Local-SGD. Further, Gu et al. (2023; 2024) do not characterize the exact implicit bias, which we do for linearly separable data. For overparameterized non-linear models, several works (Deng et al., 2022b; Song et al., 2023; Maralappanavar et al., 2025) analyze convergence in loss value of Local-GD, but do not provide any guarantees on it's implicit bias. Additionally, several works compare the performance of Local-GD and GD on the whole dataset (Patel et al., 2024; Woodworth et al., 2020a) with differences in certain regimes. For overparametrized linear models, we establish that there is no difference between the final model learned by either of these methods.

## 6 Experiments

We conducted various experiments on linear classification and neural network fine-tuning. We compared the **global model**, i.e., the output of Local-GD (Algorithm 1), with the ***centralized model***, i.e., the model obtained from running GD on a dataset consisting of all distributed datasets at one place, in different scenarios.

### 6.1 Experiments on Linear Regression

We simulated 10 compute nodes, each with 50 training samples. The label vector $y_i$ at $i$-th compute node is exactly generated as (14), where ground truth model $w_i^*$ is Gaussian vector with each element following $\mathcal{N}(0,4)$. Each ground truth model at different compute nodes is independently generated, thus the datasets can be very different from each other. The data matrix $X_i$ also follows Gaussian distribution, with each element being $\mathcal{N}(0,1)$, and $z_i$ is a Gaussian vector with $\mathcal{N}(0,0.04)$. In Local-GD, the number of local steps is $L=200$, number of rounds is also $K=200$, and the learning rate $\eta=0.0001$. Actually it just take a few local steps to converge locally at each round, but we set a large number of local steps to show it can be large at $\mathcal{O}(\sqrt{T})$, where $T=L*K$ is the number of total iterations. We tested the global model (G) from Local-GD on squared loss, centralized model (C) trained from global dataset on squared loss, closed form of global model (G-Closed) in (18), closed form of centralized model (C-Closed) as solution of problem (19). The centralized model is trained 10000 steps with learning rate 0.0001.

Fig. 1(a) displays the difference between global model and centralized model, global model and its closed form, and centralized model and its closed form, with respect to model dimension. The difference between two models is $\|w_1-w_2\|/d$. Since it is always locally overparameterized, the difference between global model and the closed form is always zero. The difference between global model and centralized model has an obvious peak around 500, which is the number of total samples. The phenomenon that global model converges exactly to centralized model only happens when the model is sufficiently overparameterized. Fig. 1(b) shows the generalization error of global model and centralized model in linear regression. Since the data matrix is Gaussian, the generalization error of model $w$ can be computed as $\frac{1}{M}\sum_{i=1}^{M}\|w-w_i^*\|^2$. We plot the generalization error divided by $d$. It is shown the global model and centralized model can get the same performance when model is sufficiently overparameterized.

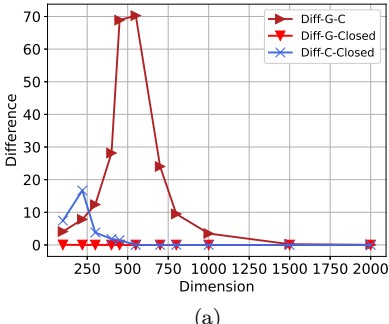 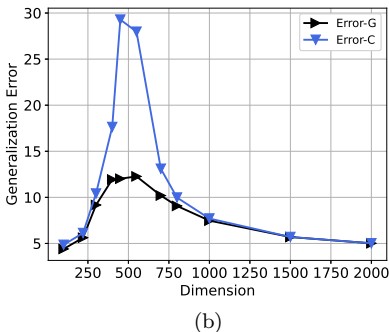

(a)            (b)

Figure 1: (a) Difference between global and centralized models plotted against increasing dimension. (b) Generalization error with respect to dimension.

### 6.2 Linear Classification

For linear classification, we have 10 compute nodes with 50 training samples at each. The dataset is generated as $y_{ij} = \text{sign}(x_{ij}^T w_i^*)$, where ground truth model is $w_i^* = w^* + z_i$, and $w^*$ is a Gaussian vector randomly chosen, $z_i$ is a Gaussian noise. The data matrix $X_i$ is a Gaussian matrix. This setting makes sure the datasets across compute nodes are different from each other, meanwhile they are not totally different such that there may be a non-empty global feasible set.

We tested four models for linear classification. The global model (G) is trained exactly with Local-GD and logistic loss. The centralized model (C) is trained with gradient descent on the global dataset. The global model from Modified Aggregation (G-Mod) is trained with exponential loss and regularization term as $\lambda=0.0001$. The

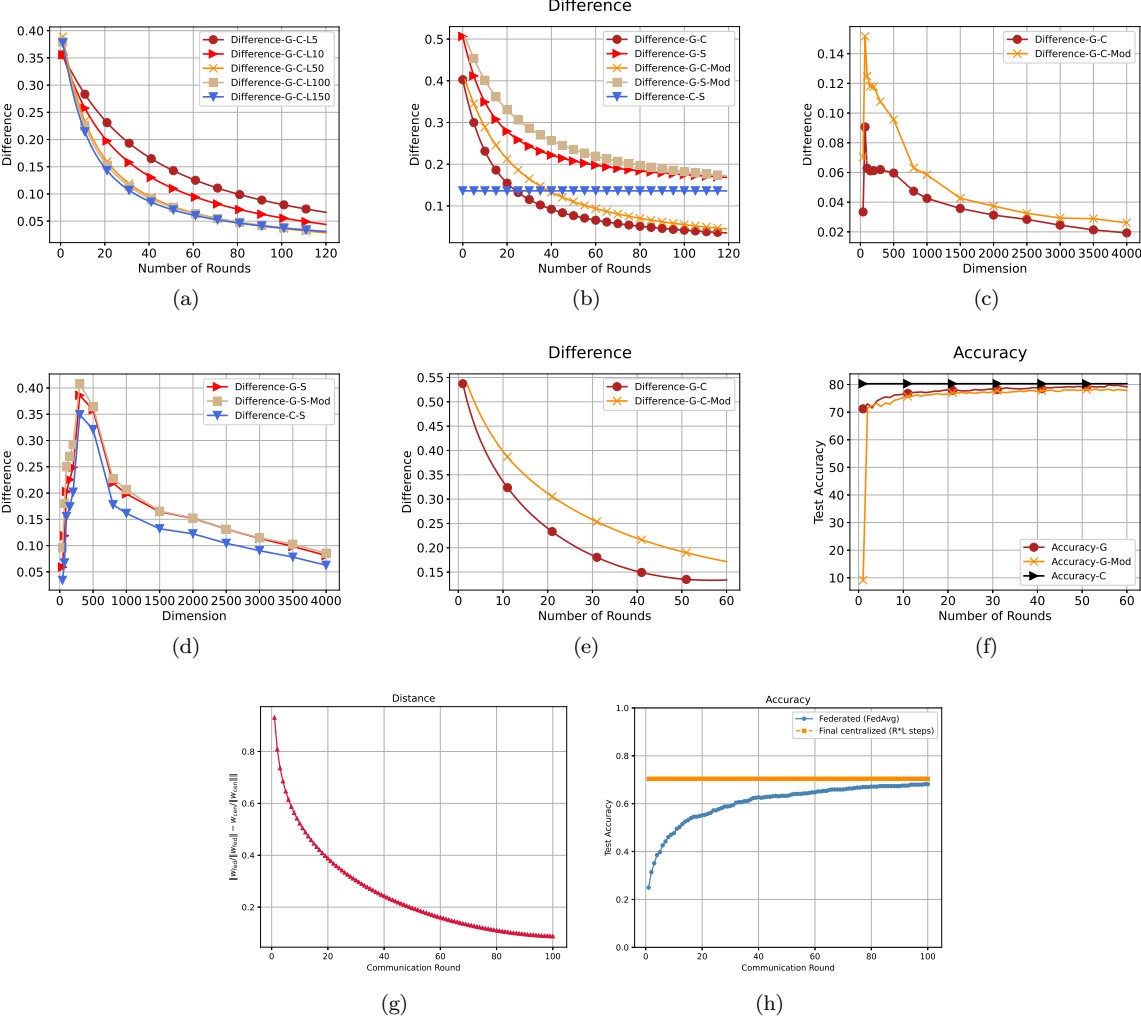

Figure 2: (a) Difference between global model and centralized model with $L$. (b) Difference between global model and centralized model with $K$. (c) Difference between global model and centralized model with $d$. (d) Difference from SVM model with $d$. (e) Difference between global linear layer and centralized linear layer for CIFAR10. (f) Test accuracy of neural network fine-tuning for CIFAR10. (g) Difference between global linear layer and centralized linear layer for FEMNIST. (f) Test accuracy of neural network fine-tuning for FEMNIST.

centralized SVM model (S) (max-margin solution) is obtained by solving problem (4) via standard scikit-learn package. Note that centralized model and SVM model are the final trained model in the plots. The learning rate of (local) gradient descent is $\eta=0.01$. Since our theory claimed the convergence is established in direction, the difference here for two models $w_1, w_2$ is defined after normalization $\|w_1/\|w_1\| - w_2/\|w_2\|\|$.

In Fig. 2(a), we show the difference between global model from Local-GD and centralized model with different number of local steps. The model dimension is chosen as $d=1500$, ensuring it is globally overparameterized. The centralized model is trained with 20000 gradient descent steps. It is seen the difference can approach zero for all the $L$, and larger $L$ can result in faster convergence to the centralized model.

In Figs. 2(b), 2(c), 2(d), the number of local steps is fixed as $L=150$ for Local-GD and Modified Aggregation, and the number of communication rounds is fixed as $K=120$ for all the dimensions. Fig. 2(b) shows the difference between these models with respect to the number of rounds $K$ when dimension is $d=1500$. We can see both global model and modified global model converges to the centralized model in direction, and the centralized model is close to the SVM model but there is small gap. Fig. 2(c) displays the difference with respect to dimension $d$. It

is seen the difference between global model and centralized model gradually decreases with larger dimensions. The modified global model is almost the same as the centralized model but the gap is slightly larger since it converges slower than vanilla global model with same number of rounds. Fig. 2(d) shows the difference from SVM model with dimension. The gap between the models to SVM model also decreases with larger $d$.

### 6.3 Fine-Tuning of Pretrained Neural Network

We further fine-tuned the ResNet50 model pretrained with ImageNet dataset on CIFAR10 and FEMNIST datasets Caldas et al. (2018). Only the final linear layer is trained during the process, while the rest of model is fixed. For CIFAR10 dataset, the 50000 samples are distributed on 10 compute nodes. For $i$-th compute node, the half of local dataset belongs to the same class, and the other half consists of rest of 9 classes evenly, which forms a heterogeneous data distribution. The centralized model is trained with the whole CIFAR10 dataset. The models are trained with cross entropy loss and Local SGD. The learning rate is 0.01 and the batch size is 128. The number of local steps is $L=60$ and number of communication rounds is $K=60$. The centralized model is trained with the same learning rate for 3600 steps. For FMNIST dataset from Leaf benchmark, there are 805263 samples across 3597 nodes. The data distribution with 62 classes is unbalanced and very heterogeneous. We randomly choose $M$ nodes from the dataset. The $K$ are $L$ are 100. We plot the difference between the linear layer and test accuracy with number of rounds in Fig. 2(e) and 2(f) for CIFAR10 dataset and Fig. 2(g) and 2(h) for FEMNIST dataset. Again the difference is defined in direction. We can see the difference gradually decreases to a small error floor and the accuracy of global models and centralized model is very similar at last.

We put additional experimental results on linear classification with a heterogeneous Dirichlet distribution, and on impact of $M$ for FEMNIST dataset in Appendix A.

## 7 Conclusions

In this work we analyzed the implicit bias of GD in distributed setting, and characterized the dynamics of the global model trained from Local-GD and Local-SGD. We showed that Local-GD can converge to a centrally trained model for linearly separable data with a constant learning rate $\mathcal{O}(1/L)$, and a Modified Aggregation can have the same convergence for a exactly-solved local problems. Our analysis provided a new perspective why Local-GD works well in practice even with a large number of local steps on heterogeneous data. Nevertheless, there are a couple of directions remaining open. Firstly, our analysis applies to linear models. How to extend the implicit bias of Local-GD to advanced neural network architectures is a worthwhile direction for future work. Secondly, we adopted linear separability in the analysis. We showed in Section 3.2 that it is possible to extend the analysis to non-separable case.

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

## Contents

# A Additional Experiments

## A.1 Linear Classification with Dirichlet Distribution

In federated learning, the Dirichlet distribution is usually used to generate heterogeneous datasets across the compute nodes (Hsu et al., 2019; Chen & Chao, 2021; Reguieg et al., 2023). For binary classification problem, the Dirichlet distribution $\text{Dir}(\alpha)$ is used to unbalance the positive and negative samples. In the experiments we have 10 compute nodes. We generate 500 samples as $y_i = \text{sign}(x_i^T w^*)$ for $i \in [500]$ and use $\text{Dir}(\alpha)$ to distribute the 500 samples across 10 compute nodes. Note that the number of samples at each compute node is not necessarily identical. Fig. 3 shows performance of Local-GD for linear classification with different parameter $\alpha$ in Dirichlet distribution. The $\lambda$ is set to be 0.0001 and model dimension is fixed as $d = 1500$. The number of local steps $L$ is 150 and number of communication rounds $R$ is 150. The learning rate is 0.01. The centralized model is trained with the same learning rate for 22500 steps. We can see the global model and modified global model still converge to the centralized model in direction and get similar test accuracy.

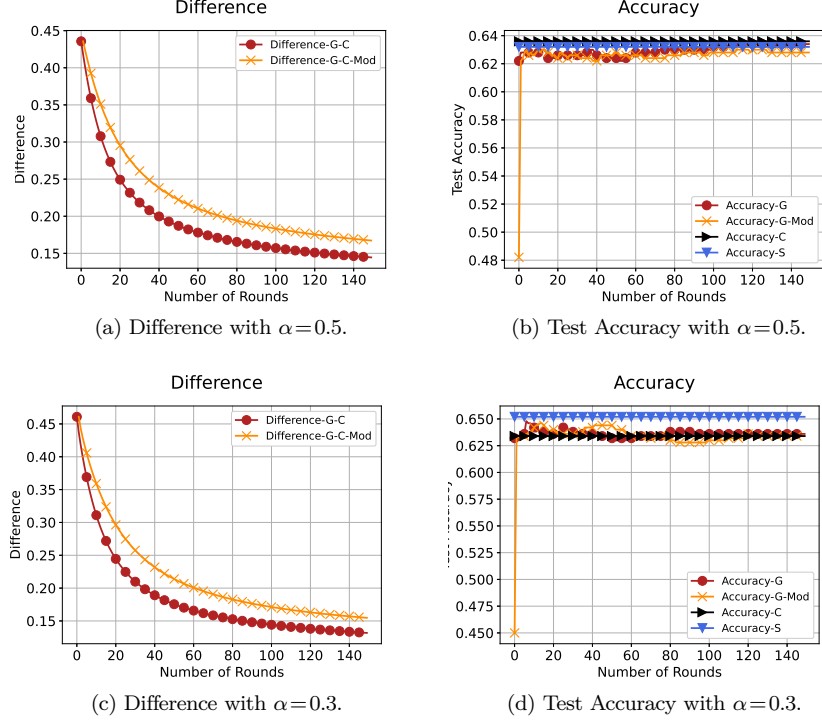

(a) Difference with $\alpha = 0.5$.  (b) Test Accuracy with $\alpha = 0.5$.

(c) Difference with $\alpha = 0.3$.  (d) Test Accuracy with $\alpha = 0.3$.

Figure 3: Local-GD on linear classification with Dirichlet distribution.

## A.2 Impact of number of compute nodes on FEMNIST dataset

Here we present the experimental results with different number of compute nodes $M$ on FEMNIST dataset. Recall that there are 805263 samples across 3597 nodes. The data distribution with 62 classes is unbalanced and very heterogeneous. We randomly choose $M$ nodes from the dataset. Here we choose $M = 5, 10, 20$ nodes from the whole datasets to form heterogeneous datasets. We can observe the performance are similar with different $M$, and less compute nodes converge slightly slower.

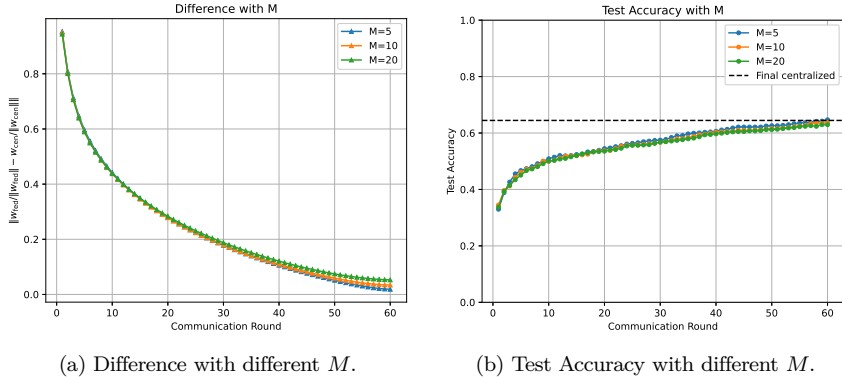

(a) Difference with different $M$.  (b) Test Accuracy with different $M$.

Figure 4: Local-GD on FEMNIST dataset with different $M$.

# B    Local-GD for Linear Regression in Overparameterized Regime

In this section we give a extended description of Section 2 about linear regression in overparameterized regime.

## B.1    Setting

The behavior of linear regression is very well-understood in high-dimensional statistics; and we can clearly convey our key message based on this fundamental setting.

At each compute node $i$, the dataset $S_i$ consists of $N$ tuples of samples and their corresponding labels, $(x,y) \in \mathbb{R}^d \times \mathbb{R}$. We assume the label $y_{ij}$ is generated by

$$y_{ij} = x_{ij}^T w_i^* + z_{ij} \tag{14}$$

where $w_i^* \in \mathbb{R}^d$ is the ground truth model at $i$-th compute node, and $z_{ij}$ is the added noise. Denote $X_i = [x_{i1}, x_{i2}, ..., x_{iN}]^T \in \mathbb{R}^{N \times d}$ as the data matrix at $i$-th compute node, and $y_i = [y_{i1}, y_{i2}, ..., y_{iN}] \in \mathbb{R}^N$ as the label vector, $z_i \in \mathbb{R}^N$ as the noise vector. In heterogeneous setting, the $w_i^*$ can be very different to each other. Note that the convergence to centralized model does not rely on the generative model. We just make this assumption on generative model for deriving a more clear form of the aggregated global model.

**Algorithm.** At each round, the aggregator sends the global model $w_0$ to all the compute nodes. Each compute node minimizes the squared loss $f_i(w_i) = \frac{1}{2N} \| y_i - X_i w_i \|^2$ by a large number of gradient descent steps *until convergence.* Then each compute node sends back the local model and the aggregator aggregates all the local models to get the updated global model. The detailed algorithm is Local-GD in Algorithm 1 with $f_i(w_i)$ replaced in the update. Since minimizing squared loss is a quadratic problem, it is expected to reach convergence locally with a small number of gradient descent steps.

## B.2    Implicit Bias of Local GD in Linear Regression

For each local problem, when the dimension of the model is larger than the number of samples at each compute node $(d > N)$, i.e., locally overparameterized, there are multiple solutions corresponding to zero squared loss. However, gradient descent will lead the model converge to a specific solution, which corresponds to a minimum Euclidean distance to the initial point (Gunasekar et al., 2018a; Evron et al., 2022). Formally, the solution $w_i^{k+1}$ obtained at $k$-th round and $i$-th node will converge to the solution of the optimization problem

$$\min_{w_i} \quad \| w_i - w_0^k \|^2 \quad \text{s.t.} \quad X_i w_i = y_i. \tag{15}$$

We can obtained the closed form solution of this optimization problem as (see Proof of Lemma 4 in Appendix B.4.1)

$$
\begin{aligned}
w_i^{k+1} &= \left(I - X_i^T (X_i X_i^T)^{-1} X_i\right) w_0^k + X_i^T (X_i X_i^T)^{-1} y_i \\
&= \left(I - X_i^T (X_i X_i^T)^{-1} X_i\right) w_0^k \\
&\quad + X_i^T (X_i X_i^T)^{-1} X_i w_i^* + X_i^T (X_i X_i^T)^{-1} z_i.
\end{aligned}
\tag{16}
$$

Denote $P_i \triangleq X_i^T (X_i X_i^T)^{-1} X_i$ and $X_i^\dagger \triangleq X_i^T (X_i X_i^T)^{-1}$. The local model can be rewritten as $w_i^{k+1} = (I - P_i) w_0^k + P_i w_i^* + X_i^\dagger z_i$. We observe that $P_i$ is the projection operator to the row space of $X_i$, and $X_i^\dagger$ is the pseudo inverse of $X_i$. After one round of iterations, the local model is actually an interpolation between the initial global model $w_0^k$ at this round and the ground-truth model $w_i^*$, plus a noise term. We then obtain the closed form of global model by aggregation. After many rounds of communication, we can obtain the final trained global model from Local-GD.

**Lemma 4.** *When the local overparameterized linear regression problems are exactly solved by gradient descent, then after $K$ rounds of communication, the global model $w_0^K$ obtained from Local-GD is*

$$
w_0^K = (I - \bar{P})^K w_0^0 + \sum_{k=0}^{K-1} (I - \bar{P})^k (\bar{Q} + \bar{Z}),
\tag{17}
$$

*where $\bar{P} = \frac{1}{M} \sum_{i=1}^M P_i, \bar{Q} = \frac{1}{M} \sum_{i=1}^M P_i w_i^*, \bar{Z} = \frac{1}{M} \sum_{i=1}^M X_i^\dagger z_i$.*

Note that $\bar{P}, \bar{Q}, \bar{Z}$ are constant after the data is generated. Since we only know the $\{X_i, y_i\}_{i=1}^M$ in the training process, we can also write it as

$$
w_0^K = (I - \bar{P})^K w_0^0 + \sum_{k=0}^{K-1} (I - \bar{P})^k \bar{Y},
\tag{18}
$$

where $\bar{Y} = \frac{1}{M} \sum_{i=1}^M X_i^\dagger y_i$. Then we can directly get the final model from the training set.

**Singularity of $\bar{P}$.** If $\bar{P}$ is invertible, we can further simplify the form of global model. However, since $P_i \in \mathbb{R}^{d \times d}$ is the projection operator onto row space of $X_i$, its rank is at most N. The $\bar{P}$ is the average of $P_i$s, thus its rank is at most $MN$. Note that we consider the overparameterized regime both locally and globally, i.e., $d \gg MN$. Then $\bar{P}$ is singular, and the sum $\sum_{k=0}^{K-1} (I - \bar{P})^k$ approaches $KI$ when $d$ becomes very large. We cannot get more properties of the final global model from (18), but we can compare it to the centralized model trained with all of the data.

### B.3 Convergence to Centralized Model

Let $X_c = [X_1^T, ..., X_M^T]^T \in \mathbb{R}^{MN \times d}$ be the data matrix consisting of all the local data, and $y_c = [y_1^T, ..., y_M^T]^T \in \mathbb{R}^{MN \times 1}$ be the label vector consisting of the local labels. If we train the centralized model from initial point 0 with squared loss, then the gradient descent will lead the model to the solution of the optimization problem

$$
\min_w \quad \|w\|^2 \quad \text{s.t.} \quad X_c w = y_c
\tag{19}
$$

We can write the closed form of centralized model as $w_c = X_c^T (X_c X_c^T)^{-1} y_c$.

Due to the constraint in problem (19), for each compute node $i$, we have $X_i w_c = y_i$. We replace $y_i$ in the local model (16), then we have

$$
w_i^{k+1} - w_c = (I - P_i)(w_0^k - w_c).
\tag{20}
$$

The RHS is projecting the difference between global model and centralized model onto null space of $X_i$. After averaging all the local models at the aggregator, we have

$$
w_0^{k+1} - w_c = (I - \bar{P})(w_0^k - w_c).
\tag{21}
$$

In the training process the difference between global model and centralized model is iteratively projected onto the null space of span of row spaces of $X_i$s. It implies that the difference on the span of data matrix gradually decreases until zero. Based on the evolution of the difference, we can prove the Theorem 1 and we restate it here:

**Theorem 8.** *For the linear regression problem, suppose the initial point $w_0^0$ is 0 and $d > MN$, and there exists minimum non-zero eigenvalue of $\bar{P}$ denoted by $\theta_{\min}$. Then the output of Local-GD, $w_0^K$, converges to the centralized solution $w_c$ as the number of communication rounds $K \to \infty$ as $\|w_0^K - w_c\| \leq (1 - \theta_{\min})^K \|w_c\|$.*

The proof is in Appendix B.4.2. The key step is to show the initial difference is already in the data space, and no residual in the null space of row spaces of $X_i$s. The convergence to the centralized model is at exponential rate.

Due to the linearity of the regression problem, we can theoretically show the global model can exactly converge to the centralized model with implicit bias on overparameterized regime. Note that the proof does not rely on the generative model and assumption on data heterogeneity. It implies that, even if we use a large number of local steps to exactly solve the local problems on very heterogeneous data, the performance of Local-GD is equivalent to train a model with all the data in one place.

### B.4 Proofs in Linear Regression

### B.4.1 Proof of Lemma 4

At each compute node, the local model converges to the solution of problem

$$\min_{w_i} \quad \|w_i - w_0^k\|^2 \quad \text{s.t.} \quad X_i w_i = y_i. \tag{22}$$

Using Lagrange multipliers, we can write the Lagrangian as

$$\frac{1}{2}\|w_i - w_0^k\|^2 + \beta^T(X_i w_i - y_i) \tag{23}$$

Setting the derivative to 0, we know the optimal $\tilde{w}_i$ satisfies

$$\tilde{w}_i - w_0^k + X_i^T \beta = 0, \tag{24}$$

and then

$$\tilde{w}_i = w_0^k - X_i^T \beta. \tag{25}$$

Also by the constraint $y_i = X_i \tilde{w}_i$, we can get

$$y_i = X_i w_0^k - (X_i X_i^T)\beta. \tag{26}$$

Since the model is overparameterized $(d > N)$, $X_i X_i^T \in \mathbb{R}^{d \times d}$ is invertible. Then we have

$$\beta = -(X_i X_i^T)^{-1}(y_i - X_i w_0^k). \tag{27}$$

Plugging the $\beta$ back, we can get the closed form solution as

$$\tilde{w}_i = w_0^k + X_i^T(X_i X_i^T)^{-1}(y_i - X_i w_0^k). \tag{28}$$

We update the local model $w_i^{k+1} = \tilde{w}_i$.

We can also write the closed form solution as

$$\begin{aligned}
w_i^{k+1} &= w_0^k + X_i^T(X_i X_i^T)^{-1}(y_i - X_i w_0^k) \\
&= \left(I - X_i^T(X_i X_i^T)^{-1}X_i\right)w_0^k + X_i^T(X_i X_i^T)^{-1}y_i
\end{aligned} \tag{29}$$

If we plug in the generative model $y_i = X_i w_i^* + z_i$, then the solution is

$$\begin{aligned}
w_i^{k+1} &= \left(I - X_i^T(X_i X_i^T)^{-1}X_i\right)w_0^k + X_i^T(X_i X_i^T)^{-1}X_i w_i^* + X_i^T(X_i X_i^T)^{-1}z_i \\
&= (I - P_i)w_0^k + P_i w_i^* + X_i^\dagger z_i.
\end{aligned} \tag{30}$$

where $P_i = X_i^T(X_iX_i^T)^{-1}X_i$ is the projection operator to the row space of $X_i$, and $X_i^\dagger = X_i^T(X_iX_i^T)^{-1}$ is the pseudo inverse of $X_i$. It is an interpolation between the initial global model $w_0^k$ and the local true model $w_i^*$, plus a noise term.

After aggregating all the local models, the global model is

$$w_0^{k+1} = \frac{1}{m}\sum_{i=1}^{m}(I-P_i)w_0^k + \frac{1}{m}\sum_{i=1}^{m}P_iw_i^* + \frac{1}{m}\sum_{i=1}^{m}X_i^\dagger z_i$$
$$= (I-\bar{P})w_0^k + \bar{Q} + \bar{Z}, \tag{31}$$

where $\bar{P} = \frac{1}{m}\sum_{i=1}^{m}P_i, \bar{Q} = \sum_{i=1}^{m}P_iw_i^*, \bar{Z} = \frac{1}{m}\sum_{i=1}^{m}X_i^\dagger z_i$.

After $K$ rounds of communication, the global model is

$$w_0^K = (I-\bar{P})^K w_0^0 + \sum_{k=0}^{K-1}(I-\bar{P})(\bar{Q}+\bar{Z}). \tag{32}$$

If we start from $w_0^0 = 0$, then the solution will converge to $\sum_{k=0}^{K-1}(I-\bar{P})(\bar{Q}+\bar{Z})$.

### B.4.2  Proof of Theorem 1

We know the difference between global model and centralized model is iteratively projected onto the null space of span of row spaces of $X_i$s:

$$w_0^{k+1} - w_c = (I-\bar{P})(w_0^k - w_c). \tag{33}$$

We can formally describe it as follows. Since the problem is overparameterized globally, we can assume each $X_i$ has full rank $N$. We apply singular value decomposition (SVD) to $X_i$ as $X_i = U_i\Sigma_iV_i^T$, where $U_i \in \mathbb{R}^{N\times N}, V_i \in \mathbb{R}^{d\times N}$. Then $P_i = X_i^T(X_iX_i^T)^{-1}X_i = V_iV_i^T$, which is the projection matrix to the row space of $X_i$.

We apply eigenvalue decomposition on $\bar{P}$ to get $\bar{P} = Q\Sigma Q^T$, where $Q \in \mathbb{R}^{d\times n'}$ and $n'$ is the rank of $\bar{P}$. It satisfies $N \le n' \le MN$. Since $\bar{P}$ is a linear combination of $P_i$s, the space of column space of $Q$ is the space spanned by all the vectors $v_{ij}, i=1,...,M, j=1,...,N$.

We also construct a matrix $Q' \in \mathbb{R}^{d\times(d-n')}$, which consists of orthonomal vectors perpendicular to $Q$. We can project the difference onto column space of $Q$ and $Q'$ respectively.

$$Q^T(w_0^{k+1}-w_c) = Q^T(I-Q\Sigma Q^T)(w_0^k-w_c) = (I-\Sigma)Q^T(w_0^k-w_c)$$
$$Q'^T(w_0^{k+1}-w_c) = Q'^T(I-Q\Sigma Q^T)(w_0^k-w_c) = Q'^T(w_0^k-w_c) \tag{34}$$

After $K$ rounds of communication, we can decomposite $w_0^K - w_c$ into two parts:

$$w_0^K - w_c = QQ^T(w_0^K-w_c) + Q'Q'^T(w_0^K-w_c). \tag{35}$$

Then we can obtain

$$w_0^K - w_c = QQ^T(w_0^K-w_c) + Q'Q'^T(w_0^K-w_c)$$
$$= Q(I-\Sigma)^K Q^T(w_0^0-w_c) + Q'Q'^T(w_0^0-w_c).$$

It shows the initial difference on the column space of $Q$ continues to decrease until zero if $K$ is sufficiently large. And the initial difference on the null space of $Q$ remains constant.

To show the difference $w_0^K - w_c$ goes to zero entirely, we just need to choose an initial point such that initial difference is on the column space of $Q$. When we choose $w_0^0 = 0$, the initial difference is $w_c$ itself. Moreover, the centralized solution $w_c = X_c^T(X_cX_c^T)^{-1}y_c$ exactly lies in the data space spanned by vectors $\{v_{ij}\}_{i=1,j=1}^{M,N}$ since it is a linear combination of columns of $X_c^T$. So if we start from $w_0^0 = 0$, then $w_0^K - w_c$ will go to zero when $K$ is sufficiently large.

When starting from 0, the difference between the global model and the centralized model becomes

$$
\begin{aligned}
\|w_0^K - w_c\|^2 &= \|Q(I-\Sigma)^K Q^T w_c\|^2 \\
&= \left(Q(I-\Sigma)^K Q^T w_c\right)^T \left(Q(I-\Sigma)^K Q^T w_c\right) \\
&= \left(Q^T w_c\right)^T (I-\Sigma)^{2K} \left(Q^T w_c\right).
\end{aligned}
\tag{36}
$$

Since $I-\Sigma$ is a diagonal matrix, we can get

$$
\|w_0^K - w_c\|^2 \leq (1-\theta_{\min})^{2K} \|Q^T w_c\|^2,
\tag{37}
$$

where $\theta_{\min}$ is the minimum eigenvalue of matrix $\bar{P}$. Also since $Q$ is an orthogonal matrix, we have $\|Q^T w_c\|^2 = \|w_c\|^2$. Then we can get

$$
\|w_0^K - w_c\| \leq (1-\theta_{\min})^K \|w_c\|.
\tag{38}
$$

It shows the difference between trained global model and centralized model converge to zero at an exponential rate.

# C  Proofs of Implicit Bias for Linear Classification in Section 3

We give the detailed proofs of Theorem 2 in this section. The proof framework is inspired by the analysis of implicit bias of SGD (Nacson et al., 2019). Intuitively, we can regard one local dataset as a "batch" in SGD for sampling without replacement. But we perform multiple gradient steps in the same "batch", not just one step of gradient descent. The challenge is to handle local steps in the same local dataset and the aggregation after one round of local training. Here we restate the Theorem 2.

**Theorem 9.** *Under assumptions 1, 2, 3, if the learning rate satisfies $\eta \le \min\left(\frac{1}{2L\sigma_{max}^2\beta}, \frac{\gamma^2}{4L\sigma_{\max}^3\beta(\gamma+\sigma_{\max})}\right)$, then for the process of Local-GD, we have,*

- ***Claim 1:*** *Every data point is classified correctly finally:* $\lim_{k\to\infty} x_s^T w_0^k = \infty, \forall s \in S$.
- ***Claim 2:*** *The global model obtained from Local-GD will behave as*

$$w_0^k = \log(Lk)\hat{w} + \rho^k, \quad and, \quad \left\|\frac{w_0^k}{\|w_0^k\|} - \frac{\hat{w}}{\|\hat{w}\|}\right\| = O\left(\frac{1}{\log Lk}\right) \tag{39}$$

*and $\|\rho^k\| < \infty$ for all $k$. This implies, the normalized global model converges to the global max-margin solution.*

- ***Claim 3:*** *The loss function $f(w_0^k)$ decreases to zero as $f(w_0^k) = O\left(\frac{1}{Lk}\right)$.*

For the three claims in Theorem 2, we will give separable (but sequential) proofs below.In the proofs of linear classification, for ease of notation, we redefine the samples $y_s x_s$ to $x_s$ to subsume the labels.

## C.1  Proof of Claim 1

In this proof, we rely on the key property of linearly separable data.

**Lemma 5** (Lemma 2 and (17) in Nacson et al. (2019)). *Suppose that Assumptions 1 and 2 hold. For any $w \in \mathbb{R}^d$,*

$$\|\nabla f(w)\| \ge \frac{\gamma}{M}\sqrt{\sum_{s\in S}[g'(x_s^T w)]^2}.$$

**Lemma 6.** *Suppose that Assumptions 1 and 2 hold and $k \in \mathbb{N}$. Then we have*

$$\|w_i^{k,l} - w_0^k + \eta(l\nabla f_i(w_0^k))\| \le \frac{\eta^2 L\sigma_{\max}^3\beta Ml}{\gamma(1 - l\eta\beta\sigma_{\max}^2)}\|\nabla f(w_0^k)\|. \tag{40}$$

$$\|w_i^{k,l} - w_0^k\| \le \frac{\eta L\sigma_{\max}M}{\gamma(1 - l\eta\beta\sigma_{\max}^2)}\|\nabla f(w_0^k)\|. \tag{41}$$

$$\|\nabla f(w_i^{k,l}) - \nabla f(w_0^k)\| \le \frac{\eta L\sigma_{\max}^3\beta M}{\gamma(1 - l\eta\beta\sigma_{\max}^2)}\|\nabla f(w_0^k)\|. \tag{42}$$

The proof can be seen in Section C.1.1.

Note that $f(w) = \frac{1}{M}\sum_{i=1}^M f_i(w) = \frac{1}{M}\sum_{s\in S}g(x_s^T w)$, and $g(u)$ is a $\beta$-smooth function from Assumption 2. Then $f(w)$ is a $\frac{\beta\sigma_{\max}^2}{M}$-smooth function. Then we can get

$$\begin{aligned}
&f(w_0^{k+1}) - f(w_0^k) - \frac{\sigma_{\max}^2\beta}{2M}\|w_0^{k+1} - w_0^k\|^2 \\
&\le \langle\nabla f(w_0^k), (w_0^{k+1} - w_0^k)\rangle \\
&= \langle\nabla f(w_0^k), w_0^{k+1} - w_0^k - \eta L\nabla f(w_0^k) + \eta L\nabla f(w_0^k)\rangle \\
&\le -\eta L\|\nabla f(w_0^k)\|^2 + \|\nabla f(w_0^k)\|\|w_0^{k+1} - w_0^k + \eta L\nabla f(w_0^k)\|,
\end{aligned} \tag{43}$$

where the second inequality is from Cauchy-Schwarz inequality.

For the second term, we have

$$\|w_0^{k+1}-w_0^k+\eta L\nabla f(w_0^k)\|$$

$$=\|\frac{1}{M}\sum_{i=1}^M w_i^{k+1}-w_0^k+\eta L\frac{1}{M}\sum_{i=1}^M\nabla f_i(w_0^k)\|$$

$$\leq\frac{1}{M}\sum_{i=1}^M\|w_i^{k+1}-w_0^k+\eta L\nabla f_i(w_0^k)\|$$

$$\leq\frac{1}{M}\sum_{i=1}^M\frac{\eta^2 L^2\sigma_{\max}^3\beta M}{\gamma(1-L\eta\beta\sigma_{\max}^2)}\|\nabla f(w_0^k)\|$$

$$=\frac{\eta^2 L^2\sigma_{\max}^3\beta M}{\gamma(1-L\eta\beta\sigma_{\max}^2)}\|\nabla f(w_0^k)\| \tag{44}$$

where the first inequality is triangle inequality and second inequality is from Lemma 6.

We also have

$$\|w_0^{k+1}-w_0^k\|^2=\|\frac{1}{M}\sum_{i=1}^M w_i^{k+1}-w_0^k\|^2$$

$$\leq\frac{1}{M}\sum_{i=1}^M\|w_i^{k+1}-w_0^k\|^2$$

$$\leq\frac{\eta^2 L^2\sigma_{\max}^2 M^2}{\gamma^2(1-L\eta\beta\sigma_{\max}^2)^2}\|\nabla f(w_0^k)\|^2 \tag{45}$$

where the second inequality is from Lemma 6. Plug above two inequalities into (43), we can get

$$f(w_0^{k+1})-f(w_0^k)\leq-\eta L\left(1-\frac{\eta L\sigma_{\max}^3\beta M}{\gamma(1-L\eta\beta\sigma_{\max}^2)}-\frac{\eta L\sigma_{\max}^4\beta M}{2\gamma^2(1-L\eta\beta\sigma_{\max}^2)^2}\right)\|\nabla f(w_0^k)\|^2 \tag{46}$$

If we choose $\eta\leq\frac{1}{2L\sigma_{\max}^2\beta}$, then $\frac{1}{1-L\eta\beta\sigma_{\max}^2}\leq 2$. Thus we can obtain

$$f(w_0^{k+1})-f(w_0^k)\leq-\eta L\left(1-\eta L\sigma_{\max}^3\beta M(\frac{2}{\gamma}+\frac{2\sigma_{\max}}{\gamma^2})\right)\|\nabla f(w_0^k)\|^2$$

$$=-\eta L(1-\eta L\beta')\|\nabla f(w_0^k)\|^2 \tag{47}$$

where $\beta'=\frac{2\sigma_{\max}^3\beta M(\gamma+\sigma_{\max})}{\gamma^2}$.

If we also choose $\eta\leq\frac{1}{2L\beta'}$, then

$$f(w_0^{k+1})-f(w_0^k)\leq-\frac{\eta L}{2}\|\nabla f(w_0^k)\|^2, \tag{48}$$

which means the loss continues to decrease.

Combining the two condition on step size, we require

$$\eta\leq\min\left(\frac{1}{2L\sigma_{max}^2\beta},\frac{\gamma^2}{4L\sigma_{\max}^3\beta M(\gamma+\sigma_{\max})}\right). \tag{49}$$

Summing up from $k=0$ to $\infty$, we have

$$\sum_{k=0}^{\infty}\|\nabla f(w_0^k)\|^2\leq\frac{2(f(w_0^0)-f(w_0^{\infty}))}{\eta L}\leq\frac{2f(w_0^0)}{\eta L}<\infty \tag{50}$$

The boundedness means $\lim_{k\to\infty}\|\nabla f(w_0^k)\|^2=0$. From Lemma 5, we can also know $\lim_{k\to\infty}g'(x_s^Tw_0^k)=0,\forall s\in S$. From Assumption 2, $g'(u)\to 0$ only when $u\to\infty$, thus $x_s^Tw_0^k\to\infty,\forall s\in S$, which means all the training samples can be correctly classified. This proves Claim 1 in Theorem 2.

We also bound the change of weights across iterations here, which is useful in the proof of Claim 2. since $\nabla f_i(w)=\sum_{s\in S_i}g'(x_s^Tw)x_s$ we can have

$$\frac{1}{M}\sum_{i=1}^{M}\|w_i^{k,l+1}-w_i^{k,l}\|=\frac{1}{M}\sum_{i=1}^{M}\eta\|\nabla f_i(w_i^{k,l})\|$$

$$=\frac{1}{M}\sum_{i=1}^{M}\eta\|\sum_{s\in S_i}g'(x_s^Tw_i^{k,l})x_s\|$$

$$\leq\frac{1}{M}\sum_{i=1}^{M}\eta\sigma_{\max}\sqrt{\sum_{s\in S_i}\left(g'(x_s^Tw_i^{k,l})\right)^2}$$

$$\leq\frac{1}{M}\sum_{i=1}^{M}\eta\sigma_{\max}\sqrt{\sum_{s\in S}\left(g'(x_s^Tw_i^{k,l})\right)^2}$$

$$\leq\frac{\eta\sigma_{\max}}{\gamma}\sum_{i=1}^{M}\|\nabla f(w_i^{k,l})\|, \tag{51}$$

where the first inequality is from the fact $\|\sum_{s\in S}a_sx_s\|\leq\sigma_{\max}\sqrt{\sum_{s\in S}a_s^2}$ for $\forall a_s\in\mathbb{R}$, the second inequality is due to $S_i\subset S$, and the final inequality is from Lemma 5. Further we can obtain

$$\|\nabla f(w_i^{k,l})\|\leq\|\nabla f(w_0^k)\|+\|\nabla f(w_i^{k,l})-\nabla f(w_0^t))\|$$

$$\leq\|\nabla f(w_0^k)\|+\frac{\eta L\sigma_{\max}^3\beta M}{\gamma(1-l\eta\beta\sigma_{\max}^2)}\|\nabla f(w_0^k)\|$$

$$=\left(1+\frac{\eta L\sigma_{\max}^3\beta M}{\gamma(1-l\eta\beta\sigma_{\max}^2)}\right)\|\nabla f(w_0^k)\| \tag{52}$$

where the second inequality is from Lemma 6. Then we have

$$\frac{1}{M}\sum_{i=1}^{M}\|w_i^{k,l+1}-w_i^{k,l}\|^2\leq\frac{1}{M}\sum_{i=1}^{M}\frac{\eta^2\sigma_{\max}^2M^2}{\gamma^2}\left(1+\frac{\eta L\sigma_{\max}^3\beta M}{\gamma(1-l\eta\beta\sigma_{\max}^2)}\right)^2\|\nabla f(w_0^k)\|^2$$

$$\leq\frac{\eta^2\sigma_{\max}^2M^2}{\gamma^2}\left(1+\frac{\eta L\sigma_{\max}^3\beta M}{\gamma(1-L\eta\beta\sigma_{\max}^2)}\right)^2\|\nabla f(w_0^k)\|^2 \tag{53}$$

Summing up all the changes, we can finally have

$$\frac{1}{M}\sum_{k=0}^{\infty}\sum_{l=1}^{L-1}\sum_{i=1}^{M}\|w_i^{k,l+1}-w_i^{k,l}\|^2\leq\frac{\eta^2\sigma_{\max}^2LM^2}{\gamma^2}\left(1+\frac{\eta L\sigma_{\max}^3\beta M}{\gamma(1-L\eta\beta\sigma_{\max}^2)}\right)^2\sum_{k=0}^{\infty}\|\nabla f(w_0^k)\|^2<\infty. \tag{54}$$

### C.1.1 Proof of Lemma 6

*Proof.* We start from the update rule:

$$w_i^{k,l}=w_0^k-\eta\left(\sum_{l'=0}^{l-1}\nabla f_i(w_i^{k,l'})\right). \tag{55}$$

Define $\Delta := w_i^{k,l} - w_0^k + \eta(l\nabla f_i(w_0^k))$. Then by triangle inequality, we have

$$
\begin{aligned}
\|\Delta\| &= \| -\eta\sum_{l'=0}^{l-1}\nabla f_i(w_i^{k,l'}) + \eta l\nabla f_i(w_0^k)) \| \\
&= \eta\|\sum_{l'=0}^{l-1}\Big(\nabla f_i(w_i^{k,l'}) - \nabla f_i(w_0^k)\Big)\| \\
&\leq \eta\sum_{l'=0}^{l-1}\|\nabla f_i(w_i^{k,l'}) - \nabla f_i(w_0^k)\| \\
&\leq \eta\beta_i\sum_{l'=0}^{l-1}\|w_i^{k,l'} - w_0^k\|
\end{aligned}
\tag{56}
$$

where $\beta_i$ is the smoothness parameter of $f_i(w)$. Since each local dataset of a subset of global dataset, $\forall i\in[1,M], \beta_i\leq\beta\sigma_{\max}^2$.

In addition, since $\nabla f_i(w) = \sum_{s\in S_i}g'(x_s^Tw)x_s$ we can have

$$
\begin{aligned}
&\|w_i^{k,l} - w_0^k\| \\
&= \|w_i^{k,l} - w_0^k + \eta l\nabla f_i(w_0^k) - \eta l\nabla f_i(w_0^k)\| \\
&\leq \|w_i^{k,l} - w_0^k + \eta l\nabla f_i(w_0^k)\| + \eta\|l\sum_{s\in S_i}g'(x_s^Tw_0^k)x_s\| \\
&\leq \|\Delta\| + \eta l\sigma_{\max}\sqrt{\sum_{s\in S_i}\big(g'(x_s^Tw_0^k)\big)^2} \\
&\leq \|\Delta\| + \eta L\sigma_{\max}\sqrt{\sum_{s\in S}\big(g'(x_s^Tw_0^k)\big)^2} \\
&\leq \|\Delta\| + \frac{\eta L\sigma_{\max}M}{\gamma}\|f(w_0^k)\|
\end{aligned}
\tag{57}
$$

where the second inequality is from the fact $\|\sum_{s\in S}a_sx_s\| \leq \sigma_{\max}\sqrt{\sum_{s\in S}a_s^2}$ for $\forall a_s\in\mathbb{R}$, the third inequality is due to $S_i\subset S$, and the final inequality is from Lemma 5. Then we plug in $\|\Delta\|$ and get

$$
\|w_i^{k,l} - w_0^k\| \leq \eta\beta\sigma_{\max}^2\sum_{l'=0}^{l-1}\|w_i^{k,l'} - w_0^k\| + \frac{\eta L\sigma_{\max}M}{\gamma}\|f(w_0^k)\|.
\tag{58}
$$

Now we use another lemma from Nacson et al. (2019):

**Lemma 7** (Lemma 4 in Nacson et al. (2019)). *Let $\epsilon$ and $\theta$ be positive constants. If $\delta_k\leq\theta+\epsilon\sum_{u=0}^{k-1}\delta_u$, then*

$$
\delta_k \leq \frac{\theta}{1-k\epsilon} \quad \text{and} \quad \sum_{u=0}^{k-1}\delta_u \leq \frac{k\theta}{1-k\epsilon}.
$$

Directly applying this lemma to (58), we can obtain

$$
\|w_i^{k,l} - w_0^k\| \leq \frac{\eta L\sigma_{\max}M}{\gamma(1-l\eta\beta\sigma_{\max}^2)}\|\nabla f(w_0^k)\|.
\tag{59}
$$

Then we further have

$$
\|\Delta\| \leq \eta\beta\sigma_{\max}^2\sum_{l'=0}^{l-1}\|w_i^{k,l'} - w_0^k\| \leq \frac{\eta^2 L\sigma_{\max}^3\beta Ml}{\gamma(1-l\eta\beta\sigma_{\max}^2)}\|\nabla f(w_0^k)\|.
\tag{60}
$$

By smoothness, we also have

$$\|\nabla f(w_i^{k,l}) - \nabla f(w_0^k)\| \le \sigma_{\max}^2 \beta \|w_i^{k,l} - w_0^k\| \le \frac{\eta L \sigma_{\max}^3 \beta M}{\gamma(1 - l\eta\beta\sigma_{\max}^2)} \|\nabla f(w_0^k)\|. \tag{61}$$

$\square$

### C.2 Proof of Claim 2

In this section, we prove our implicit bias result. Recall that $\hat{w}$ is the global max-margin solution defined in (4). We denote the set of support vectors in $S$ as $V$. Thus the max-margin solution is $\hat{w} = \sum_{s \in S} \alpha_s x_s$, where $\alpha_s > 0, \forall s \in V; \alpha_s = 0, \forall s \notin V$. We further define a vector $\tilde{w}$, which satisfies

$$\alpha_s = \eta\exp(-x_s^T \tilde{w}) \quad \forall s \in V. \tag{62}$$

From Lemma 12 in Soudry et al. (2018), this solution exists for almost every dataset. We also denote the minimum margin to a non-support vector as

$$\theta = \min_{s \notin V} x_s^T \hat{w} > 1. \tag{63}$$

We will use the following Lemma:

**Lemma 8.** *There exists $m_i(k,l)$ such that*

$$L\sum_{u=1}^{k-1} \frac{1}{u} \frac{1}{M} \sum_{s \in V} \alpha_s x_s + \frac{l}{k} \sum_{s \in V_i} \alpha_s x_s = \frac{L}{M}\log(k)\hat{w} + \frac{L}{M}\zeta\hat{w} + m_i(k,l), \quad \forall l \in [1,L] \tag{64}$$

$$m_i(k+1,0) \triangleq \frac{1}{M}\sum_{i=1}^{M} m_i(k,L), \quad \forall i \in [1,M] \tag{65}$$

*where $\|m_i(k,l)\| = o(k^{-1})$ and $\|m_i(k,l+1) - m_i(k,l)\| = \mathcal{O}(k^{-1})$. $\zeta$ is Euler-Mascheroni constant, which is used to calculate $\sum_{u=1}^{k} \frac{1}{u} = \log k + \zeta + \mathcal{O}(k^{-1})$.*

Now we define $r_i^{k,l}, \rho_i^{k,l}$ as

$$\begin{aligned} w_i^{k,l} &= \log(Lk)\hat{w} + \rho_i^{k,l} \\ &= \log(Lk)\hat{w} + \tilde{w} + \frac{M}{L}m_i(k,l) + r_i^{k,l}, \quad \forall l \in [1,L]. \end{aligned} \tag{66}$$

Also, define $r_0^{k+1} = \frac{1}{M}\sum_{i=1}^{M} r_i^{k,L}$ and $\rho^{k+1} = \frac{1}{M}\sum_{i=1}^{M} \rho_i^{k,L}$. Thus

$$w_0^k = \frac{1}{M}\sum_{i=1}^{M} w_i^{k,L} = \log(Lk)\hat{w} + \rho^k = \log(Lk)\hat{w} + \tilde{w} + \frac{M}{L}\frac{1}{M}\sum_{i=1}^{M} m_i(k,l) + r_0^k \tag{67}$$

We also define

$$\rho_i^{k,0} = \rho^k, \quad r_i^{k,0} = r_0^k \tag{68}$$

Then for $l = 0$, we have

$$w_i^{k+1,0} = w_0^{k+1} = \log(Lk)\hat{w} + \tilde{w} + m_i(k+1,0) + r_i^{k+1,0}. \tag{69}$$

We aim to bound $\|\rho^k\|$, and we can see that it is enough to prove $\|r_0^k\|$ is bounded to achieve this goal.

We first write for a constant $k_1 > 0$ (defined later) and all $K \geq k_1$

$$
\begin{aligned}
\|r_0^K\|^2 - \|r_0^{k_1}\|^2 &= \sum_{u=k_1}^{K} \|r_0^{u+1}\|^2 - \|r_0^u\|^2 \\
&\leq \sum_{u=k_1}^{K} \frac{1}{M} \sum_{i=1}^{M} \left( \|r_i^{u,L}\|^2 - \|r_i^{u,0}\|^2 \right) \\
&= \frac{1}{M} \sum_{u=k_1}^{K} \sum_{l=0}^{L-1} \sum_{i=1}^{M} \|r_i^{u,l+1}\|^2 - \|r_i^{u,l}\|^2 \\
&= \frac{1}{M} \sum_{u=k_1}^{K} \sum_{l=0}^{L-1} \sum_{i=1}^{M} 2 \left\langle r_i^{u,l+1} - r_i^{u,l}, r_i^{u,l} \right\rangle + \|r_i^{u,l+1} - r_i^{u,l}\|^2
\end{aligned}
\tag{70}
$$

We will handle the inner product and squared norm items respectively. Here we need a lemma to characterize the behavior of inner product $\langle r_i^{u,l+1} - r_i^{u,l}, r_i^{u,l} \rangle$, which can be adapted from a lemma in Nacson et al. (2019) and its proof is omitted here:

**Lemma 9** (Adapted from Lemma 6 in Nacson et al. (2019)). *Under Assumptions 1, 2, 3, $\exists \tilde{k}, C_1, C_2 > 0$ such that $\forall k > \tilde{k}$,*

$$
\langle r_i^{k,l+1} - r_i^{k,l}, r_i^{k,l} \rangle \leq C_1 (Lk)^{-\theta} + \frac{C_2 M}{L} k^{-1-0.5\tilde{\mu}}, \forall l \in [0, L-1]
\tag{71}
$$

*, where $\tilde{\mu} = \min\{\mu_+, \mu_-, 0.25\}$.*

Let $a_i^{k,l} = \frac{M}{L}(m_i(k,l+1) - m_i(k,l))$ and we know $\|m_i(k,l+1) - m_i(k,l)\| = \mathcal{O}(k^{-1})$ from Lemma 8. Then we can handle the squared norm item:

$$
\begin{aligned}
&\frac{1}{M} \sum_{u=k_1}^{K} \sum_{l=0}^{L-1} \sum_{i=1}^{M} \|r_i^{u,l+1} - r_i^{u,l}\|^2 \\
&= \frac{1}{M} \sum_{u=k_1}^{K} \sum_{l=0}^{L-1} \sum_{i=1}^{M} \|w_i^{k,l+1} - w_i^{k,l} - a_i^{k,l}\|^2 \\
&= \frac{1}{M} \sum_{u=k_1}^{K} \sum_{l=0}^{L-1} \sum_{i=1}^{M} \|w_i^{u,l+1} - w_i^{u,l}\|^2 + \frac{1}{M} \sum_{u=k_1}^{K} \sum_{l=0}^{L-1} \sum_{i=1}^{M} 2 \left\langle w_i^{u,l} - w_i^{u,l+1}, a_i^{t,k} \right\rangle + \frac{1}{M} \sum_{u=k_1}^{K} \sum_{l=0}^{L-1} \sum_{i=1}^{M} \|a_i^{u,l}\|^2 \\
&\leq \frac{1}{M} \sum_{u=k_1}^{T} \sum_{l=0}^{L-1} \sum_{i=1}^{M} \|w_i^{u,l+1} - w_i^{u,l}\|^2 + \frac{2}{M} \sqrt{\sum_{u=k_1}^{K} \sum_{l=0}^{L-1} \sum_{i=1}^{M} \|w_i^{u,l+1} - w_i^{u,l}\|^2 \sum_{u=k_1}^{K} \sum_{l=0}^{L-1} \sum_{i=1}^{M} \|a_i^{u,l}\|^2} \\
&\quad + \frac{1}{M} \sum_{u=k_1}^{K} \sum_{l=0}^{L-1} \sum_{i=1}^{M} \|a_i^{u,l}\|^2
\end{aligned}
\tag{72}
$$

Since $\|a_i^{k,l}\| = \mathcal{O}(\frac{M}{Lk})$, we can find a $k_1$ such that $\forall k \geq k_1, \forall l \in [0, L-1], \forall i \in [1, M]$ we have $\|a_i^{k,l}\| \leq \frac{M}{Lk}$. Also, we know $\frac{1}{M}\sum_{k=t_1}^{K}\sum_{l=0}^{L-1}\sum_{i=1}^{M}\|w_i^{k,l+1} - w_i^{k,l}\|^2 < \infty$ from the proof of Claim 1 (54). Then we can obtain

$$\frac{1}{M}\sum_{u=k_1}^{K}\sum_{l=0}^{L-1}\sum_{i=1}^{M}\|r_i^{u,l+1} - r_i^{u,l}\|^2$$

$$\leq \frac{1}{M}\sum_{u=k_1}^{K}\sum_{l=0}^{L-1}\sum_{i=1}^{M}\|w_i^{u,l+1} - w_i^{u,l}\|^2 + 2\sqrt{\frac{1}{M}\sum_{u=k_1}^{K}\sum_{l=0}^{L-1}\sum_{i=1}^{M}\|w_i^{u,l+1} - w_i^{u,l}\|^2 \frac{1}{M}\sum_{u=k_1}^{K}\sum_{l=0}^{L-1}\sum_{i=1}^{M}\frac{M^2}{L^2}u^{-2}}$$

$$+ \frac{1}{M}\sum_{u=k_1}^{K}\sum_{l=0}^{L-1}\sum_{i=1}^{M}\frac{M^2}{L^2}u^{-2}$$

$$< \infty. \tag{73}$$

With Lemma 9 and the fact that $\forall c > 1, \sum_{u=1}^{\infty} u^{-c} < \infty$, we can finally get

$$\|r_0^k\|^2 - \|r_0^{k_1}\|^2 \leq \frac{1}{M}\sum_{u=k_1}^{K}\sum_{l=0}^{L-1}\sum_{i=1}^{M}\left(2\left\langle r_i^{u,l+1} - r_i^{u,l}, r_i^{u,l}\right\rangle + \|r_i^{u,l+1} - r_i^{u,l}\|^2\right) < \infty. \tag{74}$$

The $\|r_0^k\|^2$ is bounded, then $\|\rho^k\|$ is also bounded. We can know $w_0^k$ converges to $\hat{w}$ in direction: $w_0^{k+1} = \log(Lk)\hat{w} + \rho^k$.

Then we can analyze the dependence of $\|\rho^k\|$ on $L$. From (54) and the condition on learning rate $\eta = \mathcal{O}(L^{-1})$ we can know

$$\frac{1}{M}\sum_{u=k_1}^{K}\sum_{l=0}^{L-1}\sum_{i=1}^{M}\|w_i^{u,l+1} - w_i^{u,l}\|^2 \leq \mathcal{O}(L^{-1})\sum_{k=0}^{\infty}\|\nabla f(w_0^k)\|^2. \tag{75}$$

Then we can write (73) as

$$\frac{1}{M}\sum_{u=k_1}^{K}\sum_{l=0}^{L-1}\sum_{i=1}^{M}\|r_i^{u,l+1} - r_i^{u,l}\|^2$$

$$\leq \mathcal{O}(L^{-1})\sum_{u=k_1}^{\infty}\|\nabla f(w_0^u)\|^2 + 2\sqrt{\mathcal{O}(L^{-1})\sum_{u=k_1}^{\infty}\|\nabla f(w_0^u)\|^2 \cdot \frac{M^2}{L}\sum_{u=k_1}^{K}u^{-2}} + \frac{M^2}{L}\sum_{u=k_1}^{K}u^{-2}$$

$$\leq \mathcal{O}(L^{-1})\left(\sum_{u=k_1}^{\infty}\|\nabla f(w_0^u)\|^2 + \sqrt{\sum_{u=k_1}^{\infty}\|\nabla f(w_0^u)\|^2 \sum_{u=k_1}^{K}u^{-2}} + \sum_{u=k_1}^{K}u^{-2}\right) \tag{76}$$

From Lemma 9, since $\theta > 1$ we can know

$$\langle r_i^{k,l+1} - r_i^{k,l}, r_i^{k,l}\rangle \leq C_1(Lk)^{-\theta} + \frac{C_2 M}{L}k^{-1-0.5\tilde{\mu}} = \mathcal{O}(L^{-1})(k^{-\theta} + k^{-1-0.5\tilde{\mu}}) \tag{77}$$

Then we can obtain

$$\|r_0^k\|^2 - \|r_0^{k_1}\|^2$$

$$\leq \frac{1}{M}\sum_{u=k_1}^{K}\sum_{l=0}^{L-1}\sum_{i=1}^{M}\left(2\left\langle r_i^{u,l+1} - r_i^{u,l}, r_i^{u,l}\right\rangle + \|r_i^{u,l+1} - r_i^{u,l}\|^2\right)$$

$$\leq \mathcal{O}(1)\sum_{u=k_1}^{K}(u^{-\theta} + u^{-1-0.5\tilde{\mu}}) + \mathcal{O}(L^{-1})\left(\sum_{u=k_1}^{\infty}\|\nabla f(w_0^u)\|^2 + \sqrt{\sum_{u=k_1}^{\infty}\|\nabla f(w_0^u)\|^2 \sum_{u=k_1}^{K}u^{-2}} + \sum_{u=k_1}^{K}u^{-2}\right)$$

$$< \infty \tag{78}$$

and the dominating term on $L$ is $\mathcal{O}(1)$.

Now we aim to derive the convergence rate of the direction. Since from (62) the $\tilde{w}$ depends on learning rate $\eta$, we further define $\tilde{w}'$ as the solution of

$$\alpha_s = \exp(-x_s^T \tilde{w}') \quad \forall s \in V. \tag{79}$$

Recall that $x_s^T \hat{w} = 1$ as $\hat{w}$ is the max-margin solution. We can get $\tilde{w} = \tilde{w}' + \log(\eta)\hat{w}$. The $\tilde{w}'$ is no longer dependent on $\eta$. Then we can write $w_0^{k+1} = \log(\eta L k)\hat{w} + \rho'^k$, where $\rho'^k = \rho^k - \log(\eta)\hat{w}$. Next we can calculate the directional convergence rate:

$$
\begin{aligned}
&\frac{w_0^{k+1}}{\|w_0^{k+1}\|} \\
&= \frac{\log(\eta L k)\hat{w} + \rho'^k}{\sqrt{\rho'^{kT}\rho'^k + \hat{w}^T\hat{w}\log^2(\eta L k) + 2\rho'^{kT}\hat{w}\log(Lk)}} \\
&= \frac{\rho'^k/\log(Lk) + \hat{w}}{\|\hat{w}\|\sqrt{1 + \frac{2\rho'^{kT}\hat{w}}{\|\hat{w}\|^2\log(\eta L k)} + \frac{\|\rho'^k\|^2}{\|\hat{w}\|^2\log^2(\eta L k)}}} \\
&= \frac{1}{\|\hat{w}\|}\left(\frac{\rho'^k}{\log(\eta L k)} + \hat{w}\right)\left[1 - \frac{\rho'^{kT}\hat{w}}{\|\hat{w}\|^2\log(\eta L k)} + \left(\frac{3}{2}\left(\frac{\rho'^{kT}\hat{w}}{\|\hat{w}\|^2}\right)^2 - \frac{\|\rho'^k\|^2}{2\|\hat{w}\|^2}\right)\frac{1}{\log^2(\eta L k)} + O\left(\frac{1}{\log^3(\eta L k)}\right)\right] \\
&= \frac{\hat{w}}{\|\hat{w}\|} + \left(\frac{\rho'^k}{\|\hat{w}\|} - \frac{\hat{w}}{\|\hat{w}\|}\frac{\rho'^{kT}\hat{w}}{\|\hat{w}\|^2}\right)\frac{1}{\log(\eta L k)} + \mathcal{O}(\frac{1}{\log^2(\eta L k)}) \\
&= \frac{\hat{w}}{\|\hat{w}\|} + \left(I - \frac{\hat{w}\hat{w}^T}{\|\hat{w}\|^2}\right)\frac{\rho'^k}{\|\hat{w}\|}\frac{1}{\log(\eta L k)} + \mathcal{O}(\frac{1}{\log^2(\eta L k)}),
\end{aligned}
\tag{80}
$$

where the third equality is from $\frac{1}{\sqrt{1+x}} = 1 - \frac{1}{2}x + \frac{3}{4}x^2 + \mathcal{O}(x^3)$. Thus we can get

$$\left\|\frac{w_0^k}{\|w_0^k\|} - \frac{\hat{w}}{\|\hat{w}\|}\right\| = \mathcal{O}\left(\frac{1}{\log(\eta L k)}\right). \tag{81}$$

### C.2.1 Proof of Lemma 8

*Proof.* We first write

$$
\begin{aligned}
&L\sum_{u=1}^{k-1}\frac{1}{u}\frac{1}{M}\sum_{s\in V}\alpha_s x_s + \frac{l}{k}\sum_{s\in V_i}\alpha_s x_s \\
&= \frac{L}{M}\hat{w}\sum_{u=1}^{k-1}\frac{1}{u} + \frac{l}{k}\sum_{s\in V_i}\alpha_s x_s \\
&= \frac{L}{M}\hat{w}(\log(k) + \zeta + \mathcal{O}(k^{-1})) + \frac{l}{k}\sum_{s\in V_i}\alpha_s x_s \\
&= \frac{L}{M}\log(k)\hat{w} + \frac{L\zeta}{M}\hat{w} + \mathcal{O}(k^{-1})\frac{L}{M}\hat{w} + \frac{l}{k}\sum_{s\in V_i}\alpha_s x_s,
\end{aligned}
\tag{82}
$$

where the first equality is definition of $\hat{w}$, the second equality is from the fact

$$\sum_{u=1}^{k}\frac{1}{u} = \log k + \zeta + \mathcal{O}(k^{-1}) \tag{83}$$

$$\text{and} \quad \log k - \log(k-1) = \mathcal{O}(k^{-1}). \tag{84}$$

Then we define

$$m_i(k,l) = L\sum_{u=1}^{k-1}\frac{1}{u}\frac{1}{M}\sum_{s\in V}\alpha_s x_s + \frac{l}{k}\sum_{s\in V_i}\alpha_s x_s - \frac{L}{M}\log(k)\hat{w} - \frac{L\zeta}{M}\hat{w}, \quad \forall l\in[1,L] \tag{85}$$

and

$$m_i(k+1,0) = \frac{1}{M}\sum_{i=1}^{M}m_i(k,L) = L\sum_{u=1}^{k}\frac{1}{u}\frac{1}{M}\sum_{s\in V}\alpha_s x_s - \frac{L}{M}\log(k)\hat{w} - \frac{L\zeta}{M}\hat{w}, \quad \forall i\in[1,M]. \tag{86}$$

We can obviously see $\|m_i(k,l)\| = \mathcal{O}(k^{-1})$. For the difference, we can get

$$\|m_i(k,l+1) - m_i(k,l)\| = \|\frac{1}{k}\sum_{s\in V_i}\alpha_s x_s\| = \mathcal{O}(k^{-1}), \quad \forall l\in[1,L-1] \tag{87}$$

$$\|m_i(k,1) - m_i(k,0)\| = \|\frac{1}{k}\sum_{s\in V_i}\alpha_s x_s - \frac{L}{M}(\log(k+1) - \log k)\| = \mathcal{O}(k^{-1}). \tag{88}$$

$\square$

### C.2.2  Proof of Lemma 9

*Proof.* By Assumption 3, there exist positive constants $\mu_+, \mu_-, \bar{u}$ such that

$$\big(1 - \exp(-\mu_- u)\big)e^{-u} \le -g'(u) \le \big(1 + \exp(-\mu_+ u)\big)e^{-u}, \qquad \forall u > \bar{u}. \tag{89}$$

For every $l\in[1,L]$ and every node $i\in[M]$,

$$w_i^{k,l} = \log(Lk)\hat{w} + \bar{w} + \frac{M}{L}m_i(k,l) + r_i^{k,l}, \tag{90}$$

where $m_i(k,l)$ is defined by

$$L\sum_{u=1}^{k-1}\frac{1}{u}\frac{1}{M}\sum_{s\in V}\alpha_s x_s + \frac{l}{k}\sum_{s\in V_i}\alpha_s x_s = \frac{L}{M}\log k\hat{w} + \frac{L}{M}\zeta\hat{w} + m_i(k,l). \tag{91}$$

We first show that, for every $l\in[0,L-1]$,

$$r_i^{k,l+1} - r_i^{k,l} = -\eta\sum_{s\in S_i}g'\big(x_s^\top w_i^{k,l}\big)x_s - \frac{M}{Lk}\sum_{s\in V_i}\alpha_s x_s. \tag{92}$$

If $l\in[1,L-1]$, we get directly

$$\frac{M}{L}\big(m_i(k,l+1) - m_i(k,l)\big) = \frac{M}{Lk}\sum_{s\in V_i}\alpha_s x_s. \tag{93}$$

Both $w_i^{k,l+1}$ and $w_i^{k,l}$ have the same log-coefficient $\log(Lk)$, so subtracting (90) at $l+1$ and at $l$ and using the local update $w_i^{k,l+1} - w_i^{k,l} = -\eta\nabla f_i(w_i^{k,l})$ yields (92).

If $l=0$, now $w_i^{k,0} = w_0^k$ includes $\log(L(k-1))$, and $m_i(k,0)$ is the averaged term of $M$ compute nodes. Using (91) at $(k-1,L)$ for every $j$,

$$m_j(k-1,L) = L\sum_{u=1}^{k-2}\frac{1}{u}\frac{1}{M}\sum_{s\in V}\alpha_s x_s + \frac{L}{k-1}\sum_{s\in V_j}\alpha_s x_s - \frac{L}{M}\log(k-1)\hat{w} - \frac{L}{M}\zeta\hat{w}.$$

Averaging over $j$ and using $\sum_{j=1}^{M}\sum_{s\in V_j}\alpha_s x_s = \sum_{s\in V}\alpha_s x_s$ (because $\{V_j\}$ partitions $V$),

$$m_i(k,0) = L\sum_{u=1}^{k-1}\frac{1}{u}\frac{1}{M}\sum_{s\in V}\alpha_s x_s - \frac{L}{M}\log(k-1)\hat{w} - \frac{L}{M}\zeta\hat{w}.$$

Combining with (91) at $l=1$ gives

$$m_i(k,1)-m_i(k,0) = \frac{1}{k}\sum_{s\in V_i}\alpha_s x_s - \frac{L}{M}\log\frac{k}{k-1}\hat{w},$$

hence

$$\frac{M}{L}\big(m_i(k,1)-m_i(k,0)\big) = \frac{M}{Lk}\sum_{s\in V_i}\alpha_s x_s - \log\frac{k}{k-1}\hat{w}. \tag{94}$$

Now using (90) at round $k$ (with $\log(Lk)$) and (90) at $l=0$ (with $\log(L(k-1))$),

$$r_i^{k,1}-r_i^{k,0} = \big(w_i^{k,1}-w_i^{k,0}\big) - \log\frac{k}{k-1}\hat{w} - \frac{M}{L}\big(m_i(k,1)-m_i(k,0)\big)$$

$$= -\eta\nabla f_i(w_0^k) - \log\frac{k}{k-1}\hat{w} - \Big[\frac{M}{Lk}\sum_{s\in V_i}\alpha_s x_s - \log\frac{k}{k-1}\hat{w}\Big]$$

$$= -\eta\sum_{s\in S_i}g'\big(x_s^\top w_0^k\big)x_s - \frac{M}{Lk}\sum_{s\in V_i}\alpha_s x_s.$$

So (92) holds for $l=0$ as well.

Then we calculate the inner product $\langle r_i^{k,l+1}-r_i^{k,l},r_i^{k,l}\rangle$. Splitting $S_i=(S_i\backslash V_i)\cup V_i$, (92) gives

$$\langle r_i^{k,l+1}-r_i^{k,l},r_i^{k,l}\rangle = \underbrace{-\eta\sum_{s\in S_i\backslash V_i}g'\big(x_s^\top w_i^{k,l}\big)x_s^\top r_i^{k,l}}_{=:T_1} - \underbrace{\sum_{s\in V_i}\Big[\eta g'\big(x_s^\top w_i^{k,l}\big)+\frac{M\alpha_s}{Lk}\Big]x_s^\top r_i^{k,l}}_{=:T_2}. \tag{95}$$

We note that our analysis is asymptotic in terms of large enough $k$, so we define the following stages. Let

$$\tilde{k}_5 := \min\big\{k'\,\big|\,\forall k\geq k',\forall l\in[0,L],\forall i\in[M],\forall s\in S\colon x_s^\top w_i^{k,l}\geq \bar{u}\big\}, \tag{96}$$

$$\tilde{k}_6 := \min\big\{k'\,\big|\,\forall k\geq k',\forall l,\forall i,\forall s\colon x_s^\top w_i^{k,l}\geq 0\big\}, \tag{97}$$

$$\tilde{k}_7 := \min\big\{k'\,\big|\,\forall k\geq k',\forall l,\forall i,\forall s\colon \exp\big(-\tfrac{M}{L}x_s^\top m_i(k,l)\big)\leq 2\big\}, \tag{98}$$

$$\tilde{k}_8 := \min\big\{k'\,\big|\,\forall k\geq k',\forall l,\forall i,\forall s\colon \exp\big(-\tfrac{M}{L}x_s^\top m_i(k,l)\big)\geq \tfrac{3}{4}\big\}, \tag{99}$$

$$\tilde{k}_9 := \min\big\{k'\,\big|\,\forall k\geq k',\forall l,\forall i,\forall s\colon \exp\big(-\mu_- x_s^\top w_i^{k,l}\big)\leq \tfrac{1}{4}\big\}. \tag{100}$$

Such $\tilde{k}_5-\tilde{k}_9$ exist because, by Theorem 2, $x_s^\top w_0^k\to\infty$ as $k\to\infty$ for every $s\in S$ and $\lim_{k\to\infty}x_s^\top w_i^{k,l}=\infty$ uniformly in $(i,l,s)$. Also $\lim_{k\to\infty}\|m_i(k,l)\|=0$ by Lemma 8.

Then we set $\tilde{k}:=\max\{\tilde{k}_5,\tilde{k}_6,\tilde{k}_7,\tilde{k}_8,\tilde{k}_9\}$.

We now aim to bound $T_1$.

For $s\in S_i\backslash V_i$ we have $s\notin V$, hence $x_s^\top\hat{w}\geq\theta>1$. Drop the terms with $x_s^\top r_i^{k,l}\leq 0$, which make the upper bound smaller, we can have

$$T_1 \leq -\eta\sum_{\substack{s\in S_i\backslash V_i\\x_s^\top r_i^{k,l}>0}}g'\big(x_s^\top w_i^{k,l}\big)x_s^\top r_i^{k,l}. \tag{101}$$

Apply (89) in the stage $k\geq\tilde{k}_5$ and $\exp(-\mu_+ x_s^\top w_i^{k,l})\leq 1$ in the stage $k\geq\tilde{k}_6$,

$$T_1 \leq 2\eta\sum_{\substack{s\in S_i\backslash V_i\\x_s^\top r_i^{k,l}>0}}\exp\big(-x_s^\top w_i^{k,l}\big)x_s^\top r_i^{k,l}. \tag{102}$$

Substituting (90),

$$\eta\exp(-x_s^\top w_i^{k,l}) = \eta e^{-c_l x_s^\top \hat{w}} e^{-x_s^\top \bar{w}} \exp\left(-\tfrac{M}{L} x_s^\top m_i(k,l) - x_s^\top r_i^{k,l}\right) \tag{103}$$

$$= \frac{\alpha_s}{e^{c_l x_s^\top \hat{w}}} \exp\left(-\tfrac{M}{L} x_s^\top m_i(k,l) - x_s^\top r_i^{k,l}\right). \tag{104}$$

Here the second equality uses the definition of $\bar{w}$. Hence, using $\exp(-(M/L)x_s^\top m_i(k,l)) \leq 2$ in the stage $k \geq \tilde{k}_7$, and $xe^{-x} \leq 1$ for $x \geq 0$ on the factor $e^{-x_s^\top r_i^{k,l}} x_s^\top r_i^{k,l}$, we can obtain

$$T_1 \leq 2 \sum_{s \in S_i \setminus V_i} \alpha_s e^{-c_l x_s^\top \hat{w}} \exp\left(-\tfrac{M}{L} x_s^\top m_i(k,l)\right) e^{-x_s^\top r_i^{k,l}} x_s^\top r_i^{k,l} \tag{105}$$

$$\leq 4 \sum_{s \in S_i \setminus V_i} \alpha_s e^{-c_l x_s^\top \hat{w}}. \tag{106}$$

Finally $x_s^\top \hat{w} \geq \theta$ and $e^{-c_l} \leq \frac{1}{L(k-1)} \leq \frac{2}{Lk}$, so

$$T_1 \leq C_1 (Lk)^{-\theta}, \qquad \text{where} \quad C_1 := 4 \cdot 2^\theta \cdot |S| \cdot \max_s \alpha_s. \tag{107}$$

Then we aim to bound $T_2$.

For every $s \in S$, define

$$A_{s,i}^{k,l} = \begin{cases} 1 + \exp(-\mu_+ x_s^\top w_i^{k,l}) & \text{if } x_s^\top r_i^{k,l} > 0, \\ 1 - \exp(-\mu_- x_s^\top w_i^{k,l}) & \text{if } x_s^\top r_i^{k,l} \leq 0. \end{cases}$$

By (89),

$$-\eta g'\left(x_s^\top w_i^{k,l}\right) x_s^\top r_i^{k,l} \leq \eta A_{s,i}^{k,l} \exp(-x_s^\top w_i^{k,l}) x_s^\top r_i^{k,l} \quad (\forall s \in S). \tag{108}$$

Using (108) inside $T_2$, which is over $V_i \subset V$, where $x_s^\top \hat{w} = 1$ and $\alpha_s = \eta\exp(-x_s^\top \bar{w})$,

$$T_2 \leq \sum_{s \in V_i} \left[\eta A_{s,i}^{k,l} \exp(-x_s^\top w_i^{k,l}) - \frac{M\alpha_s}{Lk}\right] x_s^\top r_i^{k,l}$$

$$= \sum_{s \in V_i} \frac{M\alpha_s}{Lk} \left[\frac{A_{s,i}^{k,l}}{M\delta_l} E_{s,i}^{k,l} - 1\right] x_s^\top r_i^{k,l}, \tag{109}$$

where

$$\delta_l := \begin{cases} 1, & l \in [1, L-1], \\ (k-1)/k, & l = 0, \end{cases} \qquad E_{s,i}^{k,l} := \exp\left(-\tfrac{M}{L} x_s^\top m_i(k,l) - x_s^\top r_i^{k,l}\right).$$

Since $\delta_l \to 1$, for clarity we treat $\delta_l = 1$, the $l = 0$ case differs by a factor $k/(k-1) \in [1,2]$ that is absorbed in $C_2$.

By Lemma 8, $|(M/L)x_s^\top m_i(k,l)| = o(k^{-\tilde{\mu}})$. Fix once and for all a threshold $C_7 > 0$ (to be chosen large enough at the end). We analyze each term in (109) according to four cases on $\rho := x_s^\top r_i^{k,l}$.

**Case 1.** $0 \leq \rho \leq C_7 k^{-0.5\tilde{\mu}}$

$A_{s,i}^{k,l} \leq 2$ ($k \geq \tilde{k}_6$) and $E_{s,i}^{k,l} \leq 2 \cdot 1 = 2$ ($k \geq \tilde{k}_7$, $e^{-\rho} \leq 1$), so

$$\frac{A_{s,i}^{k,l} E_{s,i}^{k,l}}{M} - 1 \leq \frac{4}{M} - 1 \leq 3.$$

Hence

$$\frac{M\alpha_s}{Lk}\left[\tfrac{A_{s,i}^{k,l} E_{s,i}^{k,l}}{M} - 1\right]\rho \leq \frac{M\alpha_s}{Lk} \cdot 3 \cdot C_7 k^{-0.5\tilde{\mu}} = \frac{3C_7 \alpha_s M}{L} k^{-1-0.5\tilde{\mu}}. \tag{110}$$

**Case 2.** $-C_7 k^{-0.5\tilde{\mu}} \leq \rho \leq 0$

Now $A_{s,i}^{k,l} \leq 1$, and $E_{s,i}^{k,l} \leq 2 \cdot e^{-\rho} \leq 2 \cdot e^{C_7 k^{-0.5\tilde{\mu}}} \leq 4$ for $k \geq \tilde{k}$ large (pick $\tilde{k}$ so $C_7 \tilde{k}^{-0.5\tilde{\mu}} \leq \log 2$). Thus $|A_{s,i}^{k,l} E_{s,i}^{k,l}/M - 1| \leq 4/M + 1 \leq 5$. The $|\rho| \leq C_7 k^{-0.5\tilde{\mu}}$ bound gives, exactly as in (110),

$$\frac{M\alpha_s}{Lk}\left[\frac{A_{s,i}^{k,l} E_{s,i}^{k,l}}{M} - 1\right]\rho \leq \frac{5C_7\alpha_s M}{L} k^{-1-0.5\tilde{\mu}}. \tag{111}$$

**Case 3.** $\rho > C_7 k^{-0.5\tilde{\mu}}$ **(large positive $\rho$)**

We must show that $\frac{A_{s,i}^{k,l} E_{s,i}^{k,l}}{M} - 1 \leq 0$, then the whole term in (109) is non-positive.

Bound $A_{s,i}^{k,l} = 1 + \exp(-\mu_+ x_s^\top w_i^{k,l})$. Substituting (90),

$$\exp(-\mu_+ x_s^\top w_i^{k,l}) = e^{-\mu_+ c_l}\exp(-\mu_+ x_s^\top \bar{w})\exp\left(-\mu_+\left(\frac{M}{L} x_s^\top m_i(k,l) + \rho\right)\right).$$

With $\rho > 0$, the last factor is $\leq 2$, hence $\exp(-\mu_+ x_s^\top w_i^{k,l}) \leq C_8 k^{-\mu_+}$ for some constant $C_8 > 0$. Therefore $A_{s,i}^{k,l} \leq 1 + C_8 k^{-\mu_+}$.

For $E_{s,i}^{k,l}$ apply $\exp(x) \leq 1 + x + x^2$ (valid for $|x| \leq 1$, true for large $k$ on both summands), and $|x_s^\top m_i(k,l)| = o(k^{-1})$:

$$\exp\left(-\frac{M}{L} x_s^\top m_i(k,l)\right) \leq 1 - \frac{M}{L} x_s^\top m_i(k,l) + \left(\frac{M}{L} x_s^\top m_i(k,l)\right)^2 = 1 + o(k^{-\tilde{\mu}}),$$
$$\exp(-\rho) \leq 1 - \rho + \rho^2 \leq 1 - C_7 k^{-0.5\tilde{\mu}} + C_7^2 k^{-\tilde{\mu}}.$$

Multiplying,

$$E_{s,i}^{k,l} \leq 1 - C_7 k^{-0.5\tilde{\mu}} + o(k^{-\tilde{\mu}}),$$
$$A_{s,i}^{k,l} E_{s,i}^{k,l} \leq \left(1 + C_8 k^{-\mu_+}\right)\left(1 - C_7 k^{-0.5\tilde{\mu}} + o(k^{-\tilde{\mu}})\right) \leq 1 - C_7 k^{-0.5\tilde{\mu}} + o(k^{-\tilde{\mu}}).$$

For $M \geq 1$ this gives, for all $k$ larger than some $\tilde{k}_+ \geq \tilde{k}$,

$$\frac{A_{s,i}^{k,l} E_{s,i}^{k,l}}{M} - 1 \leq \frac{1 - C_7 k^{-0.5\tilde{\mu}} + o(k^{-\tilde{\mu}})}{M} - 1 < 0.$$

Since $\rho > 0$, the contribution of this term to $T_2$ is $\leq 0$.

**Case 4.** $\rho < -C_7 k^{-0.5\tilde{\mu}}$ **(large negative $\rho$)**

Now we want $\frac{A_{s,i}^{k,l} E_{s,i}^{k,l}}{M} - 1 \geq 0$, i.e. $A_{s,i}^{k,l} E_{s,i}^{k,l} \geq M$, so that combined with $\rho < 0$ the contribution is $\leq 0$.

**Case 4a:** $\exp(-\rho) \geq 4M$**, i.e.** $\rho \leq -\log(4M)$. Using stage-9 $(A_{s,i}^{k,l} \geq 3/4)$ and stage-8 $(\exp(-(M/L)x_s^\top m_i(k,l)) \geq 3/4)$:
$$A_{s,i}^{k,l} E_{s,i}^{k,l} \geq \tfrac{3}{4}\cdot\tfrac{3}{4}\cdot 4M = \tfrac{9}{4}M > M.$$

So $A_{s,i}^{k,l} E_{s,i}^{k,l}/M - 1 \geq 5/4 > 0$, and the contribution is $\leq 0$.

**Case 4b:** $\exp(-\rho) < 4M$**, i.e.** $C_7 k^{-0.5\tilde{\mu}} < -\rho < \log(4M)$. Using $\exp(x) \geq 1 + x$ on each factor,

$$\exp\left(-\frac{M}{L} x_s^\top m_i(k,l)\right) \geq 1 + o(k^{-\tilde{\mu}}),$$
$$\exp(-\rho) \geq 1 + (-\rho) \geq 1 + C_7 k^{-0.5\tilde{\mu}}.$$

For the lower bound on $A_{s,i}^{k,l} = 1 - \exp(-\mu_- x_s^\top w_i^{k,l})$, the same computation as in Case 3 gives $\exp(-\mu_- x_s^\top w_i^{k,l}) \leq C_9 M^{\mu_-} k^{-\mu_-}$ (since $|\rho| \leq \log(4M)$ in this sub-case), so $A_{s,i}^{k,l} \geq 1 - C_9 M^{\mu_-} k^{-\mu_-}$. Multiplying,

$$A_{s,i}^{k,l} E_{s,i}^{k,l} \geq \left(1 - C_9 M^{\mu_-} k^{-\mu_-}\right)\left(1 + C_7 k^{-0.5\tilde{\mu}}\right)\left(1 + o(k^{-\tilde{\mu}})\right)$$
$$\geq 1 + C_7 k^{-0.5\tilde{\mu}} - \mathcal{O}(k^{-\mu_-}).$$

Because $\mu_- \geq \tilde{\mu} > 0.5\tilde{\mu}$, the term $k^{-\mu_-}$ is asymptotically dominated by $k^{-0.5\tilde{\mu}}$, so for all $k$ greater than some $\tilde{k}_- \geq \tilde{k}$,

$$A_{s,i}^{k,l} E_{s,i}^{k,l} \geq 1 + \tfrac{C_7}{2} k^{-0.5\tilde{\mu}}.$$

- If $M=1$: $A_{s,i}^{k,l}E_{s,i}^{k,l}/M-1\geq\frac{C_7}{2}k^{-0.5\tilde{\mu}}>0$, so the contribution is $\leq 0$.

- If $M\geq 2$: $A_{s,i}^{k,l}E_{s,i}^{k,l}/M-1$ may still be negative; however the magnitude of the corresponding contribution is uniformly bounded. Indeed, in this sub-case $|\rho|\leq\log(4M)$ and $|A_{s,i}^{k,l}E_{s,i}^{k,l}/M-1|\leq 3$ (since $0\leq A_{s,i}^{k,l}E_{s,i}^{k,l}\leq 4M$, hence $0\leq A_{s,i}^{k,l}E_{s,i}^{k,l}/M\leq 4$). Therefore

$$\left|\frac{M\alpha_s}{Lk}\left[\frac{A_{s,i}^{k,l}E_{s,i}^{k,l}}{M}-1\right]\rho\right|\leq\frac{3M\alpha_s\log(4M)}{Lk}.$$

Choose $C_7\geq\log(4M)\tilde{k}^{0.5\tilde{\mu}}$. Then for all $k\geq\tilde{k}$, $C_7 k^{-0.5\tilde{\mu}}\geq\log(4M)\tilde{k}^{0.5\tilde{\mu}}k^{-0.5\tilde{\mu}}\geq\log(4M)$ when $k\leq\tilde{k}$, and for $k>\tilde{k}$ the bound becomes
$$\frac{3M\alpha_s\log(4M)}{Lk}\leq\frac{3M\alpha_s C_7}{L}k^{-1-0.5\tilde{\mu}}.$$

Since $C_7$ is a constant (a function of $M$ and $\tilde{k}$), this contribution is absorbed into the same $C_2 M/L\cdot k^{-1-0.5\tilde{\mu}}$ bound as Cases 1–2.

So in both sub-cases the per-$s$ contribution of Case 4 to (109) is bounded by $\frac{C_2'\alpha_s M}{L}k^{-1-0.5\tilde{\mu}}$ for an absolute constant $C_2'$.

Putting Cases 1–4 together, for each $s\in V_i$ the per-term bound is

$$\frac{M\alpha_s}{Lk}\left[\frac{A_{s,i}^{k,l}E_{s,i}^{k,l}}{M}-1\right]x_s^\top r_i^{k,l}\leq\frac{C_2'\alpha_s M}{L}k^{-1-0.5\tilde{\mu}},$$

with $C_2'=\max\{3C_7,5C_7,3\log(4M)\}$. Summing over $s\in V_i\subseteq V$,

$$T_2\leq\frac{C_2 M}{L}k^{-1-0.5\tilde{\mu}},\qquad C_2:=C_2'|V|\max_{s\in V}\alpha_s. \tag{112}$$

Combining (95), (107), and (112), for every $k\geq\max\{\tilde{k},\tilde{k}_+,\tilde{k}_-\}=:\tilde{k}$ and every $l\in[0,L-1]$,

$$\left\langle r_i^{k,l+1}-r_i^{k,l},r_i^{k,l}\right\rangle = T_1+T_2$$
$$\leq C_1(Lk)^{-\theta}+\frac{C_2 M}{L}k^{-1-0.5\tilde{\mu}}.$$

Condition of $\tilde{k}$ **is independent of** $L$. The conditions (94), (95), (98) require $x_s^T w_i^{k,l}$ to be larger than constants since $\lim_{k\to\infty}x_s^\top w_i^{k,l}=\infty$ from the proof of Claim 1. From (82) we know $m_i(k,l)$ is basically $\mathcal{O}(k^{-1})\frac{L}{M}\hat{w}+\frac{l}{k}\sum_{s\in V_i}\alpha_s x_s$, and (96), (97) require $\|\frac{M}{L}m_i(k,l)\|$ to be close to zero. The $L$ cancels and we only need $k$ to be larger than a constant.

$\square$

### C.3 Proof of Claim 3

In the proof of Claim 1, we already know $f(w_0^k)$ would continue to decrease to zero when $k\to\infty$. Now we establish the convergence rate of $f(w_0^k)$. Recall $V$ is the set of support vectors and $\theta$ is the minimum margin for non-support vectors. From Assumptions 2 and 3, we can get

$$f(w_0^k)\leq\frac{1}{M}\sum_{s\in S}\left(1+\exp(-\mu_+x_s^T w_0^k)\right)\exp(-x_s^T w_0^k)$$
$$=\frac{1}{M}\sum_{s\in S}\left(1+\exp(-\mu_+x_s^T(\hat{w}\log(Lk)+\rho^k))\right)\exp(-x_s^T(\hat{w}\log(Lk)+\rho^k))$$
$$=\frac{1}{M}\sum_{s\in S}\left(1+(Lk)^{-\mu_+x_s^T\hat{w}}\exp(-\mu_+x_s^T\rho^k)\right)\exp(-x_s^T\rho^k)(Lk)^{-x_s^T\hat{w}}$$
$$=\frac{1}{M}\sum_{s\in S}\left[\exp(-x_s^T\rho^k)(Lk)^{-x_s^T\hat{w}}+(Lk)^{-\mu_+x_s^T\hat{w}}\exp(-\mu_+x_s^T\rho^k)\exp(-x_s^T\rho^k)(Lk)^{-x_s^T\hat{w}}\right] \tag{113}$$

We can divide the dataset $S$ into set $V$ with support vectors and the complementary set. For samples in the set $V$, we have $x_s^T \hat{w} = 1$ and we can write

$$\sum_{s \in V} \exp(-x_s^T \rho^k)(Lk)^{-x_s^T \hat{w}} + (Lk)^{-\mu_+ x_s^T \hat{w}} \exp(-\mu_+ x_s^T \rho^k) \exp(-x_s^T \rho^k)(Lk)^{-x_s^T \hat{w}}$$

$$= \sum_{s \in V} \frac{1}{Lk} \exp(-x_s^T \rho^k) + \frac{1}{(Lk)^{1+\mu_+}} \exp(-(1+\mu_+)x_s^T \rho^k) \tag{114}$$

For samples not in the set $V$, we have $x_s^T \hat{w} \geq \theta$ since $\theta$ is the minimum margin for non-support vectors. Then we can write

$$\sum_{s \notin V} \exp(-x_s^T \rho^k)(Lk)^{-x_s^T \hat{w}} + (Lk)^{-\mu_+ x_s^T \hat{w}} \exp(-\mu_+ x_s^T \rho^k) \exp(-x_s^T \rho^k)(Lk)^{-x_s^T \hat{w}}$$

$$\leq \sum_{s \notin V} \frac{1}{(Lk)^\theta} \exp(-x_s^T \rho^k) + \frac{1}{(Lk)^{(1+\mu_+)\theta}} \exp(-(1+\mu_+)x_s^T \rho^k) \tag{115}$$

Combining the two terms, we can have

$$f(w_0^k) \leq \frac{1}{M} \sum_{s \in S} \left[ \exp(-x_s^T \rho^k)(Lk)^{-x_s^T \hat{w}} + (Lk)^{-\mu_+ x_s^T \hat{w}} \exp(-\mu_+ x_s^T \rho^k) \exp(-x_s^T \rho^k)(Lk)^{-x_s^T \hat{w}} \right]$$

$$= \left[ \frac{1}{MLk} \sum_{s \in V} \exp(-x_s^T \rho^k) \right] + \mathcal{O}((Lk)^{-\max(\theta, 1+\mu_+)}). \tag{116}$$

The leading term is the first term. Recall that

$$\rho^k = \tilde{w} + \frac{M}{L} \frac{1}{M} \sum_{i=1}^{M} m_i(k, L) + r_0^k \tag{117}$$

. From (78) we know $r_0^k$ is $\mathcal{O}(1)$ on $L$. From (82) we know $\frac{1}{M} \sum_{i=1}^{M} m_i(k, L) = \mathcal{O}(k^{-1}) \frac{L}{M} \hat{w} + \frac{L}{Mk} \sum_{s \in V} \alpha_s x_s$. Thus $\frac{M}{L} \frac{1}{M} \sum_{i=1}^{M} m_i(k, L) = \mathcal{O}(k^{-1}) \hat{w} + \frac{1}{k} \sum_{s \in V} \alpha_s x_s$, which is $\mathcal{O}(\frac{1}{k})$ and $\mathcal{O}(1)$ on $L$. The remaining term is $\tilde{w}$. From the definition (62) we know $\alpha_s = \eta \exp(-x_s^T \tilde{w})$, $\forall s \in V$. Thus we can finally get

$$f(w_0^k) \leq \left[ \frac{1}{MLk\eta} \sum_{s \in V} \exp[-x_s^T (m(k+1, 0) + r_0^k)] \right] + \mathcal{O}((Lk)^{-\max(\theta, 1+\mu_+)}). \tag{118}$$

Since $m(k+1, 0) + r_0^k$ is independent of $L$ and bounded when $k \to \infty$, the final rate of loss function $f(w_0^k)$ is $\mathcal{O}(1/Lk\eta)$.

# D Proofs of Implicit Bias of LocalSGD for Linear Classification in Section 3

In this paper we formulate stochastic gradient descent as a sampling without replacement. We repeat the assumption 4 here:

**Assumption 4** (Sampling without replacement.) At every communication round, each compute node run stochastic gradient descent with $E$ epochs, where $E$ is an positive integer. Within each epoch, the mini-batches $\{S_{i,0}, S_{i,1}, ..., S_{i,L'}\}$ partition the local dataset $S_i$, where $L' = N/B$ is the number of local steps for one epoch.

Under this assumption, the number of local steps is $L = E * L'$.

---

**Algorithm 3** LOCAL-SGD.

---

1: **Input:** learning rate $\eta$.
2: Initialize $w_0^0$
3: **for** $k = 0$ to $K - 1$ **do**
4:     The aggregator sends global model $w_0^k$ to all compute nodes.
5:     **for** $i = 1$ to $i = M$ **do**
6:         compute node $i$ updates local model starting from $w_0^k$: $w_i^{k,0} = w_0^k$.
7:         **for** $l = 0$ to $L - 1$ **do**
8:             Choose a mini-batch of samples $\mathcal{B}_l$ and calculate the gradient: $G(w_i^{k,l}) = \sum_{s \in \mathcal{B}_l} g'(x_s^T w_i^{k,l}) x_s$
9:             $w_i^{k,l+1} = w_i^{k,l} - \eta G(w_i^{k,l})$.
10:         **end for**
11:         compute node $i$ sends back the updated local model $w_i^{k+1} = w_i^{k,L}$.
12:     **end for**
13:     The aggregator aggregates all the local models: $w_0^{k+1} = \frac{1}{M} \sum_{i=1}^M w_i^{k+1}$.
14: **end for**
15: **Output:** $w_0^K$.

---

We repeat the Theorem 5 here.

**Theorem 10.** *Under assumptions 1, 2, 3, 4, if the learning rate satisfies* $\eta \leq \min\left(\frac{1}{2L\sigma_{max}^2\beta}, \frac{\gamma^2}{4L\sigma_{max}^3\beta(\gamma+\sigma_{max})}\right)$, *then for the process of Local-SGD, we have,*

- ***Claim 1:*** *Every data point is classified correctly finally:* $\lim_{k \to \infty} x_s^T w_0^k = \infty, \forall s \in S$.

- ***Claim 2:*** *The global model obtained from Local-GD will behave as*

$$w_0^k = \log(Lk)\hat{w} + \rho^k, \quad and, \quad \left\|\frac{w_0^k}{\|w_0^k\|} - \frac{\hat{w}}{\|\hat{w}\|}\right\| = O\left(\frac{1}{\log(\eta Lk)}\right) \tag{119}$$

  *and* $\|\rho^k\| < \infty$ *for all* $k$. *This implies, the normalized global model converges to the global max-margin solution.*

- ***Claim 3:*** *The loss function* $f(w_0^k)$ *decreases to zero as* $f(w_0^k) = O\left(\frac{1}{\eta Lk}\right)$ *when* $k$ *is sufficiently large.*

The main difference from proofs of Theorem 2 is on the proof of Claim 1. The following proofs of Local-SGD are the exactly same as the Local-GD. Thus we only give proof of Claim 1 here.

## D.1 Proof of Claim 1

We have the similar Lemma for Local-SGD as Lemma 6:

**Lemma 10.** *Suppose that Assumptions 1 and 2 hold and* $k \in \mathbb{N}$. *Then we have*

$$\|w_i^{k,l} - w_0^k + \eta(l\nabla f_i(w_0^k))\| \leq \frac{\eta^2 L \sigma_{max}^3 \beta M l}{\gamma(1 - l\eta\beta\sigma_{max}^2)} \|\nabla f(w_0^k)\|. \tag{120}$$

$$\|w_i^{k,l} - w_0^k\| \le \frac{\eta L \sigma_{\max} M}{\gamma(1 - l\eta\beta\sigma_{\max}^2)}\|\nabla f(w_0^k)\|. \tag{121}$$

$$\|\nabla f(w_i^{k,l}) - \nabla f(w_0^k)\| \le \frac{\eta L \sigma_{\max}^3 \beta M}{\gamma(1 - l\eta\beta\sigma_{\max}^2)}\|\nabla f(w_0^k)\|. \tag{122}$$

The proof can be seen in Section D.1.1.

Note that $f(w) = \frac{1}{M}\sum_{i=1}^{M} f_i(w) = \frac{1}{M}\sum_{s \in S} g(x_s^T w)$, and $g(u)$ is a $\beta$-smooth function from Assumption 2. Then $f(w)$ is a $\frac{\beta\sigma_{\max}^2}{M}$-smooth function. Then we can get

$$
\begin{aligned}
&f(w_0^{k+1}) - f(w_0^k) - \frac{\sigma_{\max}^2\beta}{2M}\|w_0^{k+1} - w_0^k\|^2 \\
&\le \langle \nabla f(w_0^k), (w_0^{k+1} - w_0^k)\rangle \\
&= \langle \nabla f(w_0^k), w_0^{k+1} - w_0^k - \eta L\nabla f(w_0^k) + \eta L\nabla f(w_0^k)\rangle \\
&\le -\eta L\|\nabla f(w_0^k)\|^2 + \|\nabla f(w_0^k)\|\|w_0^{k+1} - w_0^k + \eta L\nabla f(w_0^k)\|,
\end{aligned}
\tag{123}
$$

where the second inequality is from Cauchy-Schwarz inequality.

For the second term, we have

$$
\begin{aligned}
&\|w_0^{k+1} - w_0^k + \eta L\nabla f(w_0^k)\| \\
&= \|\frac{1}{M}\sum_{i=1}^{M} w_i^{k+1} - w_0^k + \eta L\frac{1}{M}\sum_{i=1}^{M}\nabla f_i(w_0^k)\| \\
&\le \frac{1}{M}\sum_{i=1}^{M}\|w_i^{k+1} - w_0^k + \eta L\nabla f_i(w_0^k)\| \\
&\le \frac{1}{M}\sum_{i=1}^{M}\frac{\eta^2 L^2\sigma_{\max}^3\beta M}{\gamma(1 - L\eta\beta\sigma_{\max}^2)}\|\nabla f(w_0^k)\| \\
&= \frac{\eta^2 L^2\sigma_{\max}^3\beta M}{\gamma(1 - L\eta\beta\sigma_{\max}^2)}\|\nabla f(w_0^k)\|
\end{aligned}
\tag{124}
$$

where the first inequality is triangle inequality and second inequality is from Lemma 10.

We also have

$$
\begin{aligned}
\|w_0^{k+1} - w_0^k\|^2 &= \|\frac{1}{M}\sum_{i=1}^{M} w_i^{k+1} - w_0^k\|^2 \\
&\le \frac{1}{M}\sum_{i=1}^{M}\|w_i^{k+1} - w_0^k\|^2 \\
&\le \frac{\eta^2 L^2\sigma_{\max}^2 M^2}{\gamma^2(1 - L\eta\beta\sigma_{\max}^2)^2}\|\nabla f(w_0^k)\|^2
\end{aligned}
\tag{125}
$$

where the second inequality is from Lemma 10. Plug above two inequalities into (123), we can get

$$f(w_0^{k+1}) - f(w_0^k) \le -\eta L\left(1 - \frac{\eta L\sigma_{\max}^3\beta M}{\gamma(1 - L\eta\beta\sigma_{\max}^2)} - \frac{\eta L\sigma_{\max}^4\beta M}{2\gamma^2(1 - L\eta\beta\sigma_{\max}^2)^2}\right)\|\nabla f(w_0^k)\|^2 \tag{126}$$

If we choose $\eta \le \frac{1}{2L\sigma_{\max}^2\beta}$, then $\frac{1}{1 - L\eta\beta\sigma_{\max}^2} \le 2$. Thus we can obtain

$$
\begin{aligned}
f(w_0^{k+1}) - f(w_0^k) &\le -\eta L\left(1 - \eta L\sigma_{\max}^3\beta M(\frac{2}{\gamma} + \frac{2\sigma_{\max}}{\gamma^2})\right)\|\nabla f(w_0^k)\|^2 \\
&= -\eta L(1 - \eta L\beta')\|\nabla f(w_0^k)\|^2
\end{aligned}
\tag{127}
$$

where $\beta' = \frac{2\sigma_{\max}^3 \beta M(\gamma + \sigma_{\max})}{\gamma^2}$.

If we also choose $\eta \le \frac{1}{2L\beta'}$, then

$$f(w_0^{k+1}) - f(w_0^k) \le -\frac{\eta L}{2} \|\nabla f(w_0^k)\|^2, \tag{128}$$

which means the loss continues to decrease.

Combining the two condition on step size, we require

$$\eta \le \min\left( \frac{1}{2L\sigma_{max}^2\beta}, \frac{\gamma^2}{4L\sigma_{\max}^3\beta M(\gamma + \sigma_{\max})} \right). \tag{129}$$

Summing up from $k=0$ to $\infty$, we have

$$\sum_{k=0}^{\infty} \|\nabla f(w_0^k)\|^2 \le \frac{2(f(w_0^0) - f(w_0^\infty))}{\eta L} \le \frac{2f(w_0^0)}{\eta L} < \infty \tag{130}$$

The boundedness means $\lim_{k\to\infty} \|\nabla f(w_0^k)\|^2 = 0$. From Lemma 5, we can also know $\lim_{k\to\infty} g'(x_s^T w_0^k) = 0, \forall s \in S$. From Assumption 2, $g'(u) \to 0$ only when $u \to \infty$, thus $x_s^T w_0^k \to \infty, \forall s \in S$, which means all the training samples can be correctly classified. This proves Claim 1 in Theorem 5.

We also bound the change of weights across iterations here like the proofs of Local-GD. From the update of Local-SGD, we can have

$$\begin{aligned}
\frac{1}{M}\sum_{i=1}^{M} \|w_i^{k,l+1} - w_i^{k,l}\| &= \frac{1}{M}\sum_{i=1}^{M} \eta \|\sum_{s \in \mathcal{B}_{i,l}} g'(x_s^T w_i^{k,l}) x_s\| \\
&\le \frac{1}{M}\sum_{i=1}^{M} \eta\sigma_{\max} \sqrt{\sum_{s \in \mathcal{B}_{i,l}} \left( g'(x_s^T w_i^{k,l}) \right)^2} \\
&\le \frac{1}{M}\sum_{i=1}^{M} \eta\sigma_{\max} \sqrt{\sum_{s \in S} \left( g'(x_s^T w_i^{k,l}) \right)^2} \\
&\le \frac{\eta\sigma_{\max}}{\gamma} \sum_{i=1}^{M} \|\nabla f(w_i^{k,l})\|,
\end{aligned} \tag{131}$$

where the first inequality is from the fact $\|\sum_{s \in S} a_s x_s\| \le \sigma_{\max}\sqrt{\sum_{s \in S} a_s^2}$ for $\forall a_s \in \mathbb{R}$, the second inequality is because every mini-batch at each compute node is a subset of global dataset, and the final inequality is from Lemma 5. Further we can obtain

$$\begin{aligned}
\|\nabla f(w_i^{k,l})\| &\le \|\nabla f(w_0^k)\| + \|\nabla f(w_i^{k,l}) - \nabla f(w_0^t))\| \\
&\le \|\nabla f(w_0^k)\| + \frac{\eta L\sigma_{\max}^3\beta M}{\gamma(1 - l\eta\beta\sigma_{\max}^2)} \|\nabla f(w_0^k)\| \\
&= \left( 1 + \frac{\eta L\sigma_{\max}^3\beta M}{\gamma(1 - l\eta\beta\sigma_{\max}^2)} \right) \|\nabla f(w_0^k)\|
\end{aligned} \tag{132}$$

where the second inequality is from Lemma 6. Then we have

$$\begin{aligned}
\frac{1}{M}\sum_{i=1}^{M} \|w_i^{k,l+1} - w_i^{k,l}\|^2 &\le \frac{1}{M}\sum_{i=1}^{M} \frac{\eta^2\sigma_{\max}^2 M^2}{\gamma^2} \left( 1 + \frac{\eta L\sigma_{\max}^3\beta M}{\gamma(1 - l\eta\beta\sigma_{\max}^2)} \right)^2 \|\nabla f(w_0^k)\|^2 \\
&\le \frac{\eta^2\sigma_{\max}^2 M^2}{\gamma^2} \left( 1 + \frac{\eta L\sigma_{\max}^3\beta M}{\gamma(1 - L\eta\beta\sigma_{\max}^2)} \right)^2 \|\nabla f(w_0^k)\|^2
\end{aligned} \tag{133}$$

Summing up all the changes, we can finally have

$$\frac{1}{M}\sum_{k=0}^{\infty}\sum_{l=1}^{L-1}\sum_{i=1}^{M}\|w_i^{k,l+1}-w_i^{k,l}\|^2 \leq \frac{\eta^2\sigma_{\max}^2 LM^2}{\gamma^2}\left(1+\frac{\eta L\sigma_{\max}^3\beta M}{\gamma(1-L\eta\beta\sigma_{\max}^2)}\right)^2\sum_{k=0}^{\infty}\|\nabla f(w_0^k)\|^2 < \infty. \tag{134}$$

### D.1.1  Proof of Lemma 10

*Proof.* We start from the update rule:

$$w_i^{k,l} = w_0^k - \eta\left(\sum_{l'=0}^{l-1}G(w_i^{k,l'})\right). \tag{135}$$

Define $\Delta := w_i^{k,l} - w_0^k + \eta\sum_{l'=0}^{l-1}\sum_{s\in\mathcal{B}_{l'}}g'(x_s^T w_0^k)x_s$. Then by triangle inequality, we have

$$\begin{aligned}
\|\Delta\| &= \left\|-\eta\sum_{l'=0}^{l-1}\sum_{s\in\mathcal{B}_{l'}}g'(x_s^T w_i^{k,l'})x_s + \eta\sum_{s\in\mathcal{B}_{l'}}g'(x_s^T w_0^k)x_s\right\| \\
&= \eta\left\|\sum_{l'=0}^{l-1}\sum_{s\in\mathcal{B}_{l'}}\left(g'(x_s^T w_i^{k,l'})-g'(x_s^T w_0^k)\right)x_s\right\| \\
&\leq \eta\sum_{l'=0}^{l-1}\left\|\sum_{s\in\mathcal{B}_{l'}}\left(g'(x_s^T w_i^{k,l'})-g'(x_s^T w_0^k)\right)x_s\right\| \\
&\leq \eta\sigma_{\max}\sum_{l'=0}^{l-1}\sqrt{\sum_{s\in\mathcal{B}_{l'}}\left(g'(x_s^T w_i^{k,l'})-g'(x_s^T w_0^k)\right)^2} \\
&\leq \eta\beta\sigma_{\max}\sum_{l'=0}^{l-1}\sqrt{\sum_{s\in\mathcal{B}_{l'}}\left(x_s^T(w_i^{k,l'}-w_0^k)\right)^2} \\
&\leq \eta\beta\sigma_{\max}\sum_{l'=0}^{l-1}\sqrt{\sum_{s\in S}\left(x_s^T(w_i^{k,l'}-w_0^k)\right)^2} \\
&\leq \eta\beta\sigma_{\max}^2\sum_{l'=0}^{l-1}\|w_i^{k,l'}-w_0^k\|
\end{aligned} \tag{136}$$

where the first inequality is triangle inequality, the second inequality is from the fact $\|\sum_{s\in S}a_s x_s\| \leq \sigma_{\max}\sqrt{\sum_{s\in S}a_s^2}$ for $\forall a_s \in \mathbb{R}$, the third inequality is from the $\beta$-smoothness of $g(\cdot)$, the fourth inequality is because each batch is a subset of global dataset and the last inequality is from the definition of $\sigma_{\max}$.

Next we can have

$$\|w_i^{k,l} - w_0^k\|$$

$$= \|w_i^{k,l} - w_0^k + \eta \sum_{l'=0}^{l-1} \sum_{s \in \mathcal{B}_{l'}} g'(x_s^T w_0^k) x_s - \eta \sum_{l'=0}^{l-1} \sum_{s \in \mathcal{B}_{l'}} g'(x_s^T w_0^k) x_s\|$$

$$\leq \|w_i^{k,l} - w_0^k + \eta \sum_{l'=0}^{l-1} \sum_{s \in \mathcal{B}_{l'}} g'(x_s^T w_0^k) x_s\| + \eta \|\sum_{l'=0}^{l-1} \sum_{s \in \mathcal{B}_{l'}} g'(x_s^T w_0^k) x_s\|$$

$$\leq \|\Delta\| + \eta \sum_{l'=0}^{l-1} \sigma_{\max} \sqrt{\sum_{s \in \mathcal{B}_{l'}} \left(g'(x_s^T w_0^k)\right)^2}$$

$$\leq \|\Delta\| + \eta l \sigma_{\max} \sqrt{\sum_{s \in S} \left(g'(x_s^T w_0^k)\right)^2}$$

$$\leq \|\Delta\| + \eta L \sigma_{\max} \sqrt{\sum_{s \in S} \left(g'(x_s^T w_0^k)\right)^2}$$

$$\leq \|\Delta\| + \frac{\eta L \sigma_{\max} M}{\gamma} \|f(w_0^k)\| \tag{137}$$

where the first inequality is triangle inequality, the second inequality is also from the fact $\|\sum_{s \in S} a_s x_s\| \leq \sigma_{\max} \sqrt{\sum_{s \in S} a_s^2}$ for $\forall a_s \in \mathbb{R}$, the third inequality is because the mini-batch is a subset of global dataset, and the final inequality is from Lemma 5. Then we plug in $\|\Delta\|$ and get

$$\|w_i^{k,l} - w_0^k\| \leq \eta \beta \sigma_{\max}^2 \sum_{l'=0}^{l-1} \|w_i^{k,l'} - w_0^k\| + \frac{\eta L \sigma_{\max} M}{\gamma} \|f(w_0^k)\|. \tag{138}$$

Now we apply the lemma 7 to (138), we can obtain

$$\|w_i^{k,l} - w_0^k\| \leq \frac{\eta L \sigma_{\max} M}{\gamma(1 - l\eta\beta\sigma_{\max}^2)} \|\nabla f(w_0^k)\|. \tag{139}$$

Then we further have

$$\|\Delta\| \leq \eta \beta \sigma_{\max}^2 \sum_{l'=0}^{l-1} \|w_i^{k,l'} - w_0^k\| \leq \frac{\eta^2 L \sigma_{\max}^3 \beta M l}{\gamma(1 - l\eta\beta\sigma_{\max}^2)} \|\nabla f(w_0^k)\|. \tag{140}$$

By smoothness, we also have

$$\|\nabla f(w_i^{k,l}) - \nabla f(w_0^k)\| \leq \sigma_{\max}^2 \beta \|w_i^{k,l} - w_0^k\| \leq \frac{\eta L \sigma_{\max}^3 \beta M}{\gamma(1 - l\eta\beta\sigma_{\max}^2)} \|\nabla f(w_0^k)\|. \tag{141}$$

$\square$

# E  Proofs of Implicit Bias with Learning Rate Independent of L in Section 4

In this section we also redefine the samples $y_{ij}x_{ij}$ to $x_{ij}$ to subsume the labels. With abuse of notation, we use $S_i$ to denote the set of support vectors in $i$-th compute node and $S$ is the set of support vectors in global dataset. The number of samples $N$ is identical for all the compute nodes, and the local dataset is $\{x_{ij}, y_{ij}\}_{j=1}^N$. Since the loss function is fixed as exponential loss in this section, the $\beta$ in this section refers coefficient of support vectors, not smoothness parameter.

## E.1  Proofs of Lemma 1

We assume $\|w_0^k - \ln(\frac{1}{\lambda})\bar{w}_0^k\| = \mathcal{O}(k\ln\ln\frac{1}{\lambda})$. In this case, since $\ln\frac{1}{\lambda}$ grows faster, when $\lambda \to 0$, we can have $\lim_{\lambda\to 0}\frac{w_0^k}{\|w_0^k\|} = \frac{\bar{w}_0^k}{\|\bar{w}_0^k\|}$ for any $k$ at order $o\left(\frac{\ln(1/\lambda)}{\ln\ln(1/\lambda)}\right)$. We will prove it by induction. We define global and local residuals as $r^k = w_0^k - \ln(\frac{1}{\lambda})\bar{w}_0^k$ and $r_i^k = w_i^k - \ln(\frac{1}{\lambda})\bar{w}_i^k$.

When $k=0$, since $w_0^0 = \bar{w}_0^0 = 0$, $r_i^0 = 0$ and the assumption trivially holds.

When $k \geq 1$, we have

$$\|r^k\| = \left\|w_0^k - \ln(\frac{1}{\lambda})\bar{w}_0^k\right\| = \frac{1}{M}\left\|\sum_{i=1}^M w_i^k - \ln(\frac{1}{\lambda})\bar{w}_i^k\right\|$$

$$\leq \frac{1}{M}\sum_{i=1}^M\left\|w_i^k - \ln(\frac{1}{\lambda})\bar{w}_i^k\right\| = \frac{1}{M}\sum_{i=1}^M\|r_i^k\|. \tag{142}$$

where the inequality is triangle inequality. We then focus on the local residual $r_i^k$. We choose an $\mathcal{O}(1)$ vector $\tilde{w}_i^k$ and a sign $s_i^k \in \{-1, +1\}$ to show

$$\|r_i^k\| = \left\|w_i^k - \left[\left(\ln(\frac{1}{\lambda}) + s_i^k\ln\ln(\frac{1}{\lambda})\right)\bar{w}_i^k + \tilde{w}_i^k\right] + s_i^k\ln\ln(\frac{1}{\lambda})\bar{w}_i^k + \tilde{w}_i^k\right\|$$

$$\leq \left\|w_i^k - \left[\left(\ln(\frac{1}{\lambda}) + s_i^k\ln\ln(\frac{1}{\lambda})\right)\bar{w}_i^k + \tilde{w}_i^k\right]\right\| + \ln\ln(\frac{1}{\lambda})\|\bar{w}_i^k\| + \|\tilde{w}_i^k\| \tag{143}$$

Recall the $w_i^k$ is the solution of optimization problem

$$\operatorname*{argmin}_{w_i} f_i(w_i) = \sum_{j=1}^N \exp\left(-x_{ij}^T w_i\right) + \frac{\lambda}{2}\|w_i - w_0^{k-1}\|^2, \tag{144}$$

and the loss function $f_i(w_i)$ is a $\lambda$-strongly convex function. Thus we have

$$\|w_i^k - w\| \leq \frac{1}{\lambda}\|\nabla f_i(w)\|, \quad \text{for any } w. \tag{145}$$

Then back to 143, we have

$$\|r_i^k\| \leq \frac{1}{\lambda}\underbrace{\left\|\nabla f_i\left[\left(\ln(\frac{1}{\lambda}) + s_i^k\ln\ln(\frac{1}{\lambda})\right)\bar{w}_i^k + \tilde{w}_i^k\right]\right\|}_{\|A_i\|} + \ln\ln(\frac{1}{\lambda})\|\bar{w}_i^k\| + \|\tilde{w}_i^k\|. \tag{146}$$

Next we need to show the first term $A_i$ is at $\mathcal{O}((k-1)\ln\ln(\frac{1}{\lambda}))$, and also since $\|\bar{w}_i^k\|$ and $\|\tilde{w}_i^k\|$ are $\mathcal{O}(1)$ vectors, then $\|r_i^k\|$ is at order $\mathcal{O}(k\ln\ln(\frac{1}{\lambda}))$. After averaging, $\|r^k\|$ is also at order $\mathcal{O}(k\ln\ln(\frac{1}{\lambda}))$. This confirms the assumption made for induction.

Now we focus on the term $A_i$. The gradient of function $f_i(w)$ is

$$\nabla f_i(w_i) = \sum_j -x_{ij}\exp(-x_{ij}^T w_i) + \lambda(w_i - w_0^{k-1}). \tag{147}$$

The term $A_i$ is

$$
\begin{aligned}
A_i =& \frac{1}{\lambda}\nabla f_i\left[\left(\ln(\frac{1}{\lambda})+s_i^k\ln\ln(\frac{1}{\lambda})\right)\bar{w}_i^k+\tilde{w}_i^k\right] \\
=& -\frac{1}{\lambda}\sum_j x_{ij}\exp\left(x_{ij}^T\ln\left(\lambda\ln^{-s_i^k}(\frac{1}{\lambda})\right)\bar{w}_i^k\right)\exp(-x_{ij}^T\tilde{w}_i^k)+\left(\ln(\frac{1}{\lambda})+s_i^k\ln\ln(\frac{1}{\lambda})\right)\bar{w}_i^k+\tilde{w}_i^k-w_0^{k-1} \\
=& -\frac{1}{\lambda}\sum_j x_{ij}\left(\lambda\ln^{-s_i^k}(\frac{1}{\lambda})\right)^{x_{ij}^T\bar{w}_i^k}\exp(-x_{ij}^T\tilde{w}_i^k)+\left(\ln(\frac{1}{\lambda})+s_i^k\ln\ln(\frac{1}{\lambda})\right)\bar{w}_i^k+\tilde{w}_i^k-w_0^{k-1}.
\end{aligned}
\tag{148}
$$

Then we define the set of support vectors as $S_i^k=\{x_{ij}|x_{ij}^T\bar{w}_i^k=1\}$. Recall that we assume $r^{k-1}=w_0^{k-1}-\ln(\frac{1}{\lambda})\bar{w}_0^{k-1}$ is at order $\mathcal{O}((k-1)\ln\ln(\frac{1}{\lambda}))$. We can obtain

$$
\begin{aligned}
A_i =& -\frac{1}{\lambda}\left(\lambda\ln^{-s_i^k}(\frac{1}{\lambda})\right)^1\sum_{x_{ij}\in S_i^k}x_{ij}\exp(-x_{ij}^T\tilde{w}_i^k)-\frac{1}{\lambda}\sum_{x_{ij}\notin S_i^k}x_{ij}\left(\lambda\ln^{-s_i^k}(\frac{1}{\lambda})\right)^{x_{ij}^T\bar{w}_i^k}\exp(-x_{ij}^T\tilde{w}_i^k) \\
& +\ln(\frac{1}{\lambda})(\bar{w}_i^k-\bar{w}_0^{k-1})-r^{k-1}+s_i^k\ln\ln(\frac{1}{\lambda})\bar{w}_i^k+\tilde{w}_i^k \\
=& -\ln^{-s_i^k}(\frac{1}{\lambda})\sum_{x_{ij}\in S_i^k}x_{ij}\exp(-x_{ij}^T\tilde{w}_i^k)-\sum_{x_{ij}\notin S_i^k}x_{ij}\lambda^{x_{ij}^T\bar{w}_i^k-1}\left(\ln(\frac{1}{\lambda})\right)^{-s_i^k x_{ij}^T\bar{w}_i^k}\exp(-x_{ij}^T\tilde{w}_i^k) \\
& +\ln(\frac{1}{\lambda})(\bar{w}_i^k-\bar{w}_0^{k-1})-r^{k-1}+s_i^k\ln\ln(\frac{1}{\lambda})\bar{w}_i^k+\tilde{w}_i^k.
\end{aligned}
\tag{149}
$$

By the triangle inequality, we have

$$
\begin{aligned}
\|A_i\| \leq & \underbrace{\left\|\ln(\frac{1}{\lambda})(\bar{w}_i^k-\bar{w}_0^{k-1})-\ln^{-s_i^k}(\frac{1}{\lambda})\sum_{x_{ij}\in S_i^k}x_{ij}\exp(-x_{ij}^T\tilde{w}_i^k)\right\|}_{B_1} \\
& +\underbrace{\left\|\sum_{x_{ij}\notin S_i^k}x_{ij}\lambda^{x_{ij}^T\bar{w}_i^k-1}\left(\ln(\frac{1}{\lambda})\right)^{-s_i^k x_{ij}^T\bar{w}_i^k}\exp(-x_{ij}^T\tilde{w}_i^k)\right\|}_{B_2} \\
& +\underbrace{\|r^{k-1}\|}_{\mathcal{O}((k-1)\ln\ln(\frac{1}{\lambda}))}+\underbrace{\ln\ln(\frac{1}{\lambda})\|\bar{w}_i^k\|}_{\mathcal{O}(1)}+\underbrace{\|\tilde{w}_i^k\|}_{\mathcal{O}(1)}.
\end{aligned}
\tag{150}
$$

We just need to show $B_1$ and $B_2$ approach to 0 then $\|A_i\|$ can approach to $\mathcal{O}(k\ln\ln(\frac{1}{\lambda}))$.

We divide it into two cases.

1. When $\bar{w}_i^k=P(\bar{w}_0^{k-1})\neq\bar{w}_0^{k-1}$, meaning $\bar{w}_0^{k-1}$ is not in the convex set $C_i$. In this case we choose $s_i^k=-1$ then

$$
\begin{aligned}
B_1 =& \left\|\ln(\frac{1}{\lambda})(\bar{w}_i^k-\bar{w}_0^{k-1})-\ln(\frac{1}{\lambda})\sum_{x_{ij}\in S_i^k}x_{ij}\exp(-x_{ij}^T\tilde{w}_i^k)\right\| \\
=& \ln(\frac{1}{\lambda})\left\|(\bar{w}_i^k-\bar{w}_0^{k-1})-\sum_{x_{ij}\in S_i^k}x_{ij}\exp(-x_{ij}^T\tilde{w}_i^k)\right\|.
\end{aligned}
\tag{151}
$$

We now want to choose $\tilde{w}_i^k$ to make $B_1$ as 0. Since $\bar{w}_i^k$ is the solution of SVM problem (10), by the KKT condition of SVM problem, it can be written as

$$
\bar{w}_i^k=\bar{w}_0^{k-1}+\sum_{x_{ij}\in S_i^k}\beta_{ij}x_{ij}
\tag{152}
$$

where $\beta_{ij}$ is the dual varible corresponding to $x_{ij}$ in the set of support vectors. Thus we want to choose $\tilde{w}_i^k$ as

$$\sum_{x_{ij}\in S_i^k} \exp(-x_{ij}^T\tilde{w}_i^k)x_{ij} = \sum_{x_{ij}\in S_i^k} \beta_{ij}x_{ij}. \tag{153}$$

We can prove such a $\tilde{w}_i^k$ almost surely exists in Lemma 11.

For the term $B_2$, since $\lim_{\lambda\to 0}\lambda^{c-1}\ln^c(\frac{1}{\lambda})\to 0$ for any constant $c>1$, and $x_{ij}^T\bar{w}_i^k-1>0$ for any $x_{ij}$ being not a support vector, then we can see

$$B_2 = \left\|\sum_{x_{ij}\notin S_i^k} x_{ij}\lambda^{x_{ij}^T\bar{w}_i^k-1}\left(\ln(\frac{1}{\lambda})\right)^{x_{ij}^T\bar{w}_i^k}\exp(-x_{ij}^T\tilde{w}_i^k)\right\| \xrightarrow{\lambda\to 0} 0. \tag{154}$$

Here we choose $\tilde{w}_i^k$ and $s_i^k$ to make $B_1=0$ and $B_2\to 0$.

2. When $\bar{w}_i^k=P(\bar{w}_0^{k-1})=\bar{w}_0^{k-1}$, meaning $\bar{w}_0^{k-1}$ is already in the convex set $C_i$. Then $\bar{w}_i^k-\bar{w}_0^{k-1}=0$. In this case we choose $\tilde{w}_i^k=0$ and $s_i^k=+1$. We can have

$$B_1 = \ln^{-1}(\frac{1}{\lambda})\left\|\sum_{x_{ij}\in S_i^k} x_{ij}\right\| \xrightarrow{\lambda\to 0}, \tag{155}$$

since $\ln^{-1}(\frac{1}{\lambda})\xrightarrow{\lambda\to 0} 0$ and $\left\|\sum_{x_{ij}\in S_i^k}x_{ij}\right\|$ is $\mathcal{O}(1)$.

And since $x_{ij}^T\bar{w}_i^k-1>0$ for any $x_{ij}$ being not a support vector, we have

$$B_2 = \left\|\sum_{x_{ij}\notin S_i^k} x_{ij}\lambda^{x_{ij}^T\bar{w}_i^k-1}\left(\ln(\frac{1}{\lambda})\right)^{-x_{ij}^T\bar{w}_i^k}\right\| \xrightarrow{\lambda\to 0} 0, \tag{156}$$

where $\lambda^{x_{ij}^T\bar{w}_i^k-1}\xrightarrow{\lambda\to 0} 0$ and $\left(\ln(\frac{1}{\lambda})\right)^{-x_{ij}^T\bar{w}_i^k}\xrightarrow{\lambda\to 0} 0$. Thus we choose $\tilde{w}_i^k$ and $s_i^k$ to make $B_1\to 0$ and $B_2\to 0$.

Plugging 150 back into 146, we can obtain

$$\begin{aligned}\|r_i^k\| &\le \|A_i^k\|+\ln\ln(\frac{1}{\lambda})\|\bar{w}_i^k\|+\|\tilde{w}_i^k\| \\ &\le \underbrace{B_1+B_2}_{\to 0}+2\ln\ln(\frac{1}{\lambda})\|\bar{w}_i^k\|+2\|\tilde{w}_i^k\|+\|r^{k-1}\| \\ &\le 2\ln\ln(\frac{1}{\lambda})\|\bar{w}_i^k\|+2\|\tilde{w}_i^k\|+\|r^{k-1}\|. \end{aligned} \tag{157}$$

By the assumption $\|r^{k-1}\|=\mathcal{O}((k-1)\ln\ln(\frac{1}{\lambda}))$ and $\|\bar{w}_i^k\|=\mathcal{O}(1)$, $\|\tilde{w}_i^k\|=\mathcal{O}(1)$, we have $\|r_i^k\|=\mathcal{O}(k\ln\ln(\frac{1}{\lambda}))$.

From 142, we finally obtain

$$\|r^k\| \le \frac{1}{M}\|r_i^k\| = \mathcal{O}(k\ln\ln(\frac{1}{\lambda})), \tag{158}$$

which confirms our assumption. Then we have $\lim_{\lambda\to 0}\frac{w_0^k}{\|w_0^k\|}=\frac{\bar{w}_0^k}{\|\bar{w}_0^k\|}$ for any $k$ at order $o\left(\frac{\ln(1/\lambda)}{\ln\ln(1/\lambda)}\right)$.

## E.2 Proofs of Auxiliary Lemmas

**Lemma 11.** *For the sequence $\{\bar{w}_0^k\}$ generated by sequential SVM problems 10 and aggregations, and for almost all datasets sampled from $M$ continuous distributions, the unique dual solution $\beta_i^k\in\mathbb{R}^{|S_i|\times 1}$ satisfying the KKT conditions of SVM problem 10 has non-zero elements. Then there exists $\tilde{w}_i^k$ satisfying $X_{S_i}\tilde{w}_i^k=-\ln\beta_i^k$.*

For almost all datasets, a hyperplane can be determined by $d$ points. Thus there are at most $d$ support vectors and the set of support vectors is linearly independent.

*Proof.* By the KKT condition of SVM problem, we can write the solution as

$$\bar{w}_i^k = \bar{w}_0^{k-1} + \sum_{x_{ij} \in S_i} \beta_{ij}^k x_{ij} = \bar{w}_0^{k-1} + X_{S_i}^T \beta_i^k. \tag{159}$$

where $X_{S_i} \in \mathbb{R}^{|S_i| \times d}$ is the data matrix with all the support vectors, and $\beta_i^k \in \mathbb{R}^{|S_i| \times 1}$ is the dual variable vector. Thus we can obtain

$$\beta_i^k = \left(X_{S_i} X_{S_i}^T\right)^{-1} X_{S_i} (\bar{w}_i^k - \bar{w}_0^{k-1}) = \left(X_{S_i} X_{S_i}^T\right)^{-1} \mathbf{1}_{S_i} - \left(X_{S_i} X_{S_i}^T\right)^{-1} X_{S_i} \bar{w}_0^{k-1}, \tag{160}$$

where $X_{S_i} X_{S_i}^T$ is invertible since $X_{S_i}$ has full row rank $|S_i|$, and the second equality is from $X_{S_i} \bar{w}_i^k = \mathbf{1}_{S_i}$ with $\mathbf{1}_{S_i} \in \mathbb{R}^{|S_i| \times 1}$ being all one vector. Plugging $\beta_i^k$ back, we have

$$\bar{w}_i^k = \left[I - X_{S_i}^T \left(X_{S_i} X_{S_i}^T\right)^{-1} X_{S_i}\right] \bar{w}_0^{k-1} + X_{S_i}^T \left(X_{S_i} X_{S_i}^T\right)^{-1} \mathbf{1}_{S_i}. \tag{161}$$

After averaging, the global model is

$$\bar{w}_0^k = \left[I - \frac{1}{M} \sum_{i=1}^M X_{S_i}^T \left(X_{S_i} X_{S_i}^T\right)^{-1} X_{S_i}\right] \bar{w}_0^{k-1} + \frac{1}{M} \sum_{i=1}^M X_{S_i}^T \left(X_{S_i} X_{S_i}^T\right)^{-1} \mathbf{1}_{S_i}. \tag{162}$$

It implies $\bar{w}_0^k$ is a rational function in the components of $X_1, X_2, ..., X_M$, and also $\beta_i^k$ is also a rational function in the components of data matrices. So its entries can be expressed as $\beta_{ij}^k = p_{ij}^k(X_1, X_2, ..., X_M)/q_{ij}^k(X_1, X_2, ..., X_M)$ for some polynomials $p_{ij}^k, q_{ij}^k$. Note that $\beta_{ij}^k = 0$ only if $p_{ij}^k(X_1, X_2, ..., X_M) = 0$, and the components of $X_1, X_2, ..., X_M$ must constitute a root of polynomial $p_{ij}^k$. However, the root of any polynomial has measure zero, unless the polynomial is the zero polynomial, i.e., $p_{ij}^k(X_1, X_2, ..., X_M) = 0$ for any $X_1, X_2, ..., X_M$.

Next we need to show $p_{ij}^k$ cannot be zero polynomials. To do this, we just need to construct a specific $X_1, X_2, ..., X_M$ where the $p_{ij}^k$ is not zero polynomial. Denote $e_i \in \mathbb{R}^d$ as the $i$-th standard unit vector, and $v_1, v_2, ..., v_M$ be the number of support vectors at $M$ compute nodes. We construct the datasets as

$$X_i = r_i [e_1, e_2, ..., e_{v_i}]^T, \text{ for all } i. \tag{163}$$

where $r_i$ are positive constants that will be chosen later. For these datasets, the set of support vector is dataset itself, i.e., $X_{S_i} = X_i$. We can calculate

$$X_i X_i^T = r_i^2 I_{v_i}, \quad X_i^T X_i = r_i^2 \begin{bmatrix} I_{v_i} & \mathbf{0} \\ \mathbf{0} & \mathbf{0}_{(d-v_i) \times (d-v_i)} \end{bmatrix}, \quad X_i^T \mathbf{1}_{S_i} = r_i \begin{bmatrix} \mathbf{1}_{v_i} \\ \mathbf{0}_{d-v_i} \end{bmatrix} \tag{164}$$

Thus we have

$$\bar{w}_i^k = \left(I_d - \begin{bmatrix} I_{v_i} & \mathbf{0} \\ \mathbf{0} & \mathbf{0}_{(d-v_i) \times (d-v_i)} \end{bmatrix}\right) \bar{w}_0^{k-1} + \frac{1}{r_i} \begin{bmatrix} \mathbf{1}_{v_i} \\ \mathbf{0}_{d-v_i} \end{bmatrix}. \tag{165}$$

After averaging, the global model in 162 becomes

$$\bar{w}_0^k = \underbrace{\begin{bmatrix} 0 & & & & & & & & \\ & \ddots & & & & & & & \\ & & 0 & & & & & & \\ & & & a_1 & & & & & \\ & & & & \ddots & & & & \\ & & & & & a_{v_{\max} - v_{\min}} & & & \\ & & & & & & 1 & & \\ & & & & & & & \ddots & \\ & & & & & & & & 1 \end{bmatrix}}_{A} \bar{w}_0^{k-1} + \underbrace{\begin{bmatrix} b_1 \\ \vdots \\ b_{v_{\max}} \\ \mathbf{0}_{d-v_{\max}} \end{bmatrix}}_{b}. \tag{166}$$

where $a_j \in \{\frac{1}{M}, \frac{2}{M}, ..., \frac{M-1}{M}\}$ is a constant in the range $(0,1)$, $b_j = \frac{1}{M}\sum_{i \in B_j}\frac{1}{r_i}$ is a positive constant and $B_j \in [M]$ is a set consisting of some compute nodes. Note that $A$ and $b$ are fixed in the iterations and $A$ is a diagonal matrix. By recursively applying $\bar{w}_0^k = A\bar{w}_0^{k-1} + b$, due to $\bar{w}_0^0 = 0$, we can obtain

$$\bar{w}_0^k = (I + A + A^2 + \cdots + A^{k-1})b. \tag{167}$$

Since $A$ is diagonal, the summation is

$$\sum_{j=0}^{k-1} A^j = \begin{bmatrix} 1 & & & & & & & \\ & \ddots & & & & & & \\ & & 1 & & & & & \\ & & & \sum_{j=0}^{k-1} a_1^j & & & & \\ & & & & \ddots & & & \\ & & & & & \sum_{j=0}^{k-1} a_{v_{max}-v_{min}}^j & & \\ & & & & & & k & \\ & & & & & & & \ddots & \\ & & & & & & & & k \end{bmatrix} \tag{168}$$

Recall that

$$\begin{aligned} \beta_i^k &= (X_i X_i^T)^{-1} \mathbf{1}_{v_i} - (X_i X_i^T)^{-1} X_i \bar{w}_0^{k-1} \\ &= \frac{1}{r_i^2} \mathbf{1}_{v_i} - \frac{1}{r_i^2} (\bar{w}_0^{k-1})_{v_i} = \frac{1}{r_i^2}\left(\mathbf{1}_{v_i} - (\bar{w}_0^{k-1})_{v_i}\right). \end{aligned} \tag{169}$$

where $(\bar{w}_0^{k-1})_{v_i}$ is the vector with first $v_i$ elements of $\bar{w}_0^{k-1}$.

We need every element of $\beta_i^k$ to be positive, so that we require every element of $(\bar{w}_0^{k-1})_{v_i}$ is less than 1. Then it holds for any $i$-th compute node, thus we require every element of $(\bar{w}_0^{k-1})_{v_{max}}$ is less than 1. Since $\bar{w}_0^{k-1} = \left(\sum_{j=0}^{k-2} A^j\right)b$, the largest value of $(\bar{w}_0^{k-1})_{v_{max}}$ satisfies

$$\begin{aligned} (\bar{w}_0^{k-1})_{\text{largest}} &\leq \sum_{j=0}^{k-2}\left(\frac{M-1}{M}\right)^j \times \frac{1}{M}\sum_{i=1}^M \frac{1}{r_i^2} \\ &= M\left(1 - \left(\frac{M-1}{M}\right)^{k-1}\right) * \frac{1}{M}\sum_{i=1}^M \frac{1}{r_i^2} \end{aligned} \tag{170}$$

because the maximum value of $a_j$ is $\frac{M-1}{M}$ and the maximum value of $b_j$ is $\frac{1}{M}\sum_{i=1}^M \frac{1}{r_i^2}$.

Thus we require

$$\sum_{i=1}^M \frac{1}{r_i} < \frac{1}{1 - \left(\frac{M-1}{M}\right)^{k-1}}. \tag{171}$$

Since $\left(\frac{M-1}{M}\right)^{k-1} \to 0$ when $k \to \infty$, we only require the LHS is less than the lower bound of RHS:

$$\sum_{i=1}^M \frac{1}{r_i} < 1. \tag{172}$$

Therefore we can choose $r_i = M+1$ to make it happen.

Then we can obtain $\beta_{ij}^k > 0$ holds for any support vector $x_{ij}$ and any round $k$. And the $\tilde{w}_i^k$ simply satisfies $X_{S_i}\tilde{w}_i^k = -\ln\beta_i^k$. $\qquad\square$

### E.3 Lemma and Proofs in Section 4.4

Here we provide a lemma of Modified Local-GD similar to Lemma 1 of vanilla Local-GD.

**Lemma 12.** *For almost all datasets sampled from a continuous distribution satisfying Assumption 1, we train the global model $w_0$ from Modified Local-GD and $\bar{w}_0$ from Modified PPM. The parameter is chosen as $\alpha^k = 1 - \frac{1}{k+1}$. With initialization $w_0^0 = \bar{w}_0^0 = 0$, we have $w_0^k \to \ln\left(\frac{1}{\lambda}\right) \bar{w}_0^k$, and the residual $\|w_0^k - \ln\left(\frac{1}{\lambda}\right)\bar{w}_0^k\| = \mathcal{O}(k\ln\ln\frac{1}{\lambda})$, as $\lambda \to 0$. It implies that at any round $k = o\left(\frac{\ln(1/\lambda)}{\ln\ln(1/\lambda)}\right)$, $w_0^k$ converges in direction to $\bar{w}_0^k$:*

$$\lim_{\lambda \to 0} \frac{w_0^k}{\|w_0^k\|} = \frac{\bar{w}_0^k}{\|\bar{w}_0^k\|}. \tag{173}$$

*Proof.* With initialization $w_0^0 = \bar{w}_0^0 = 0$, the Modified Local-GD is just a scaling of vanilla Local-GD:

$$w_0^{k+1} = \frac{k}{k+1}\frac{1}{M}\sum_{i=1}^{M}w_i^{k+1}. \tag{174}$$

Also, the Modified PPM is a scaling of vanilla PPM: $\bar{w}_0^{k+1} = \frac{k}{k+1}\frac{1}{M}\sum_{i=1}^{M}\bar{w}_i^{k+1}$.

When $k \geq 1$, we can know the residual between Modified Local-GD and Modified PPM is

$$\|r^k\| = \left\|w_0^k - \ln(\frac{1}{\lambda})\bar{w}_0^k\right\| = \frac{k}{k+1}\frac{1}{M}\left\|\sum_{i=1}^{M}w_i^k - \ln(\frac{1}{\lambda})\bar{w}_i^k\right\|$$
$$\leq \frac{1}{M}\sum_{i=1}^{M}\left\|w_i^k - \ln(\frac{1}{\lambda})\bar{w}_i^k\right\| = \frac{1}{M}\sum_{i=1}^{M}\|r_i^k\|. \tag{175}$$

Then we can follow the same process in the proof of Lemma 1 to obtain

$$\|r^k\| \leq \frac{1}{M}\|r_i^k\| = \mathcal{O}(k\ln\ln(\frac{1}{\lambda})), \tag{176}$$

As a result we have $\lim_{\lambda \to 0}\frac{w_0^k}{\|w_0^k\|} = \frac{\bar{w}_0^k}{\|\bar{w}_0^k\|}$.

$\square$

