# OpenReview forum: "Effectiveness of Distributed Gradient Descent with Local Steps for Overparameterized Models"
_TMLR — Rejected by TMLR_

### Review · Reviewer_kxPb · 2026-04-05

**Summary Of Contributions:**

This paper studies the implicit bias of Local Gradient Descent (Local-GD) — also known as FedAvg — for overparameterized linear models. The central question is: among the many zero-training-loss solutions available in the interpolation regime, which one does Local-GD converge to? The authors answer this for binary classification with linearly separable data, showing that the aggregated global model converges *in direction* to the same max-margin (SVM) solution that centralized GD would find, regardless of the number of local steps $L$ and data heterogeneity across nodes. The main results are:

1. Theorem 2 (Local-GD, $\eta = O(1/L)$): The global model converges in direction to the global max-margin solution at rate $O(1/\log(Lk))$, and the loss converges at rate $O(1/(Lk))$.
2. Theorem 5 (Local-SGD): The same result extends to Local-SGD with sampling without replacement.
3. Theorems 6 & 7 (Learning rate independent of $L$): Using a weakly regularized local subproblem solved exactly, Local-GD converges to the global feasible set; a Modified Local-GD (with a time-varying aggregation) converges exactly to the centralized max-margin solution, with a learning rate independent of $L$.
4. Linear regression (Theorem 1): In the globally overparameterized regime, Local-GD converges exactly to the centralized minimum-norm solution at an exponential rate.

---

**Key strengths:**
- The paper addresses a well-motivated and practically relevant question: why Local-GD works well even with many local steps and heterogeneous data.
- The theoretical framework is clean: connecting Local-GD to Parallel Projection Methods (PPM) is an elegant insight.
- The results for linear regression and classification are complementary and paint a coherent picture.
- The extension to Local-SGD and the discussion of non-separable data add breadth.

---

**Key weaknesses:**
- The results are restricted to linear models and exponentially-tailed losses, which limits direct applicability to deep learning practice.
- The $\eta = O(1/L)$ learning rate requirement in Theorem 2 is quite restrictive and common in prior work; the "learning rate independent of $L$" result (Section 4) requires solving a modified regularized subproblem exactly, which departs from standard Local-GD.
- The experimental validation is limited in scope and does not thoroughly probe the practical implications of the theoretical findings.

**Additional Comments:**

This is a solid theoretical contribution that brings tools from implicit bias analysis and convex feasibility theory to bear on an important question in distributed learning. The core insight, where overparameterization eliminates the harm of heterogeneous local updates, is valuable and clearly communicated. The paper is generally well-written, though the experimental section needs strengthening to match the ambition of the practical claims made in the introduction.

A few additional technical questions for the authors:

1. The convergence results are asymptotic (requiring $k$ to be sufficiently large). Can the authors provide any finite-time bounds, or at least characterize how large $\bar{k}$ needs to be before the asymptotic regime kicks in?

2. In practice, not all compute nodes participate in every round. Can the results be extended to partial participation settings?

3. The paper mentions linear models as relevant for last-layer fine-tuning. Has there been any attempt to extend the analysis to the Neural Tangent Kernel (NTK) regime, where overparameterized neural networks behave approximately linearly?

Overall, I believe this paper makes a meaningful contribution to the theoretical understanding of Local-GD in overparameterized settings and, with the requested revisions, would be suitable for TMLR.

**Audience:**

Yes

**Audience Explanation:**

This paper addresses a fundamental question in distributed optimization and federated learning that is of interest to both the theoretical and applied ML communities. The finding that Local-GD implicitly converges to the centralized max-margin solution, regardless of the number of local steps, provides a novel theoretical explanation for the empirical success of FedAvg with large numbers of local steps, which is increasingly relevant given the rise of distributed LLM training methods like DiLoCo. The connection to Parallel Projection Methods is intellectually interesting and opens new analytical avenues. TMLR's audience, which includes researchers working on optimization theory, federated learning, and implicit bias, would find value in these results.

That said, the restriction to linear models and the specific learning rate requirement temper the practical impact somewhat. The paper would be of primary interest to the theoretical optimization subcommunity rather than practitioners.

**Broader Impact Concerns:**

No significant ethical concerns. The work is purely theoretical, analyzing properties of a well-established distributed optimization algorithm. The results could marginally contribute to more efficient distributed training of ML models, which has both positive and negative downstream implications, but these are indirect and well-understood.

**Claims And Evidence:**

Yes

**Claims Explanation:**

The theoretical claims are supported by detailed proofs in the appendices. The proof framework for Theorem 2 (Claims 1–3) is carefully constructed, building on the implicit bias analysis of SGD by Nacson et al. (2019) and extending it to the distributed setting with local steps and aggregation. The key technical challenge, i.e., controlling the residual $\rho^k$ across local steps and aggregation rounds, is handled rigorously.

A few observations on the evidence quality:

1. The proofs in Appendices B–E appear technically sound. The induction argument in Lemma 1 (Appendix E) for the equivalence between Local-GD with exact local solves and PPM is well-structured, though the argument that $B_1 \to 0$ and $B_2 \to 0$ as $\lambda \to 0$ relies on a careful choice of auxiliary vectors $\tilde{w}_i^k$ whose existence is justified via Lemma 11. The almost-sure genericity argument in Lemma 11 (showing dual variables are nonzero for almost all datasets) is valid but somewhat delicate; it would benefit from a brief intuitive explanation in the main text.

2. The authors claim their convergence rates match centralized GD (Soudry et al., 2018) with total steps $Lk$. This is a meaningful observation supporting the claim that local steps are "free" in terms of convergence, but it should be noted that this tightness is asymptotic and the constants hidden in $O(\cdot)$ may differ.

3. The experiments confirm the theoretical predictions for linear models (Figs. 1–3). The neural network fine-tuning experiment (Section 5.3) is suggestive but limited: only a single architecture (ResNet50) and dataset (CIFAR-10) are tested, with only the last linear layer trained. This experiment validates the linear theory rather than demonstrating broader applicability.

4. There is a notable gap between Theorem 6 (convergence to some point in the global feasible set) and Theorem 7 (convergence to the centralized model via Modified Local-GD). The vanilla Local-GD with learning rate independent of $L$ is only shown to converge to the feasible set, not to the centralized solution. While the authors are transparent about this, the practical significance of the Modified Local-GD aggregation rule deserves further empirical investigation.

**Requested Changes:**

1. **Clarify the practical gap between $\eta = O(1/L)$ and constant $\eta$.** The paper presents two regimes but the transition between them is unclear. In Theorem 2, the learning rate shrinks with $L$, so the effective step per round is $\eta L = O(1)$. This means more local steps do not actually move the model further per round, and the benefit is purely in the convergence rate counting total gradient evaluations. Please discuss this explicitly and clarify whether the benefit of large $L$ is computational (more parallelism) rather than statistical.

2. **Strengthen the experimental section.** The current experiments are adequate for illustrating the theory but insufficient for supporting the paper's claims about practical implications. Specifically:
   - Include experiments with more varied heterogeneity levels and data distributions beyond Gaussian and Dirichlet.
   - For the neural network experiment, test with more architectures and datasets, and compare against baselines like SCAFFOLD or FedProx to contextualize the results.
   - Show how the convergence rate to the centralized model depends on the number of compute nodes $M$.

3. **Discuss limitations more thoroughly.** The paper should include a dedicated limitations section (or expand the conclusions) addressing:
   - The gap between linear models and practical deep learning.
   - The requirement of global linear separability (Assumption 1) and its restrictiveness.
   - The fact that the Modified Local-GD requires a non-standard aggregation rule.
   - Whether the $O(1/\log(Lk))$ directional convergence rate is fast enough to be practically meaningful.

4. **Notation consistency.** There are minor notation issues: $M$ in the learning rate condition of Theorem 2 does not appear in the theorem statement but appears in the proof (e.g., $\beta' = 2\sigma_{\max}^3 \beta M(\gamma + \sigma_{\max})/\gamma^2$ in Eq. (49)). The dependence on $M$ should be explicit in the theorem statement.

---

> ### Author Response · Authors · 2026-05-04
>
> Thanks for the positive comments and constructive suggestions. Please kindly find our responses below.
>
> ### $\textbf{Benefit of local steps.}$
> In Section 3, we indeed need learning rate to be at the order of $\mathcal{O}(1/L)$. Since the loss convergence rate is $\mathcal{O}(1/\eta Lk)$, the final rate is $\mathcal{O}(1/k)$. But this matches the lower bound and it is the best we can achieve with such a small learning rate. In the revised version we have added a detailed discussion of the learning rate in Section 5. Please kindly check there for more insights. Our main message in this work is that even you use arbitrarily large number of local steps, the Local-GD/SGD can still converge to the centralized max-margin solution, which have good generalization properties.
>
> ### $\textbf{Additional experiments.}$
> Thanks for the suggestions. To further support our theoretical claims and consider more realistic heterogeneous setting, we have added experiments on FEMNIST dataset from Lead benchmark, which is a standard and widely-used dataset for heterogeneous federated learning. The results on FEMNIST dataset align with our results and CIFAR10 dataset. We also tested the impact of number of compute nodes $M$ in the Appendix as the reviewer suggested.
>
> ### $\textbf{Limitations.}$
> Thanks for the suggestion. We have expanded the conclusions to include a discussion of limitations. One thing to note is that the directional convergence even for centralized GD is $\mathcal{O}(1/\log t)$. Thus we already recover this rate for local steps with $\mathcal{O}(1/ \log Lk)$ rate.
>
> ### $\textbf{Large $\tilde{k}$ of asymptotic regime.}$
> To explicitly show the $\tilde{k}$ required to guarantee implicit bias result, we have added a full proof of Lemma 9 in the Appendix C2.2. We show $\tilde{k}$ is the maximum value satisfying all the conditions (94)-(98). We note that $\tilde{k}$ is independent of number of local steps $L$, and is related to the property of dataset itself.
>
> ### $\textbf{Partial participation.}$
> It is possible to extend the implicit bias results to partial participation case since we assume the global linear separability. Even in every round there are only a fraction of compute nodes participating in the communication, the asymptotic convergence should not be impacted. But it needs refined analysis to consider the set of support vectors of partial compute nodes in each communication round, which is a good future work to explore.
>
> ### $\textbf{Extension to NTK regime.}$
> That is a good idea worth exploring since NTK separability assumption is very similar to linear separability. But the main difference is that the model needs to stay in the NTK ball to ensure kernal regime. For our implicit bias analysis, the $k$ should be large enough to enter an asymptotic regime, which is not guaranteed to stay in the kernel regime. Thus we are not sure the results can be directly extended to NTK regime.

---

### Review · Reviewer_tsC8 · 2026-04-15

**Summary Of Contributions:**

The paper studies the implicit bias of Local-GD / Local-SGD in overparameterized distributed learning. It shows that the aggregated global model obtained from Local-GD, with an arbitrary number of local steps, converges in direction to the centralized model. The paper further extends the analysis to Local-SGD and also discusses a modified Local-GD variant.

Strengths:

1.The paper studies an important problem in distributed learning. The main result is interesting and provides a clear characterization of the implicit bias of Local-GD.

2.The paper includes both theoretical analysis and experimental results to support its claims.

Weaknesses:

1.The experimental setup is relatively simple compared with the paper’s practical motivation.

2.The connection between Section 3 and Section 4 is not fully clear, which affects the overall consistency of the paper.

**Audience:**

Yes

**Audience Explanation:**

The paper studies an important question in distributed learning, namely the implicit bias and convergence behavior of Local-GD / Local-SGD in overparameterized settings. I believe these findings would be of interest to the TMLR audience.

**Claims And Evidence:**

Yes

**Claims Explanation:**

The main claims are generally supported by the theoretical analysis and the experimental results, and the overall evidence is reasonably convincing.

However, there appears to be a inconsistency between Theorem 1 / Theorem 8 and the discussion in Appendix B.2. In both theorems, the convergence result is stated under the condition that, when $d>MN$, the minimum eigenvalue of $\bar P$  is strictly positive. However, Appendix B.2 explicitly states that $rank(\bar P) \leq MN$, and further notes that $\bar P$ is singular in the globally overparameterized regime. If $\bar P $ is singular, then its smallest eigenvalue cannot be strictly positive. Therefore, the theorem statement seems inconsistent with the appendix discussion.

**Requested Changes:**

1. The experiments are currently not strong enough to fully support the paper’s practical motivation. Since the paper discusses overparameterized models and explicitly mentions practical large-scale settings such as LLMs in practical implications, it would be valuable to include a more realistic LLM fine-tuning experiment.

2. Sections 3 and 4 study noticeably different settings, but the paper does not sufficiently explain the relationship between them. Section 3 studies Local-GD / Local-SGD for classification under one setting, while Section 4 moves to a different formulation with exponential loss, weak regularization, and exact solution of local subproblems. A clearer discussion is needed to improve the consistency of the presentation and help readers understand why both formulations are necessary.

3. Section 4 only presents GD-type results. It would be helpful for the authors to discuss whether the proposed idea and the resulting conclusions can be extended to Local-SGD as well.

---

> ### Author Response · Authors · 2026-05-04
>
> Thanks for the suggestions and constructive comments. Please kindly find our responses below.
>
> ### $\textbf{Additional experiments.}$
>  In this work we mainly theoretically analyzed the asymptotic implicit bias of Local-GD/SGD for linear models. Thus we choose to perform experiments with training linear models and fine-tuning last layer of neural networks to support our theoretical claims. Since our results are on the convergence of a linear model \emph{in direction}, the important metric is the vector similarity between aggregated model from Local-GD and centralized model. Thus it is harder to measure the convergence in direction for large-scale neural networks, including LLM training. Nevertheless, we have added experiments on FEMNIST dataset from Leaf benchmark to further support our results, which is a standard and widely-used dataset for heterogeneous federated learning. The results on FEMNIST dataset align with our results and CIFAR10 dataset. We also tested the impact of number of compute nodes $M$ in the Appendix A.2.
>
> ### $\textbf{Eigenvalues of $\bar{P}$.}$
> We thank the reviewer for the careful reading. We would like to clarify that the issue is from ambiguous description. The quantity $\theta_{\min}$ in the theorems refers to the smallest non-zero eigenvalue of $\bar{P}$, not the smallest eigenvalue of the full $d\times d$ matrix. This is made explicit in the proof in Appendix B.4.2. We perform the eigendecomposition $\bar{P} = Q \Sigma Q^T$, where $n' = rank(\bar{P}) \leq MN$. By construction, $\Sigma$ contains only the non-zero eigenvalues and the $d - n'$ zero eigenvalues are separated out into the orthogonal complement $Q^\prime$. The proof involves $\lambda_{\min}$ as the smallest non-zero eigenvalue of $\bar{P}$. The zero eigenvalues do not affect convergence because with the initialization $w_0^0 = 0$, the initial residual lies entirely in the column space of $\bar{P}$. The component in the null space of $\bar{P}$ is zero at initialization and stays zero throughout training. We have corrected the description in Theorem 1.
>
> ### $\textbf{Claim of Section 4.}$
> We agree the problem and setting in Section 4 is an analogy of Local-GD, so we have revised the title of Section 4 and the contribution of this part to avoid misleading. In Section 4 we actually analyzed the exactly solved local problem in distributed training, which shows several differences compared to standard Local-GD algorithm due to proof techniques. We now emphasize this point and explicitly explain it in the Section 4. But the main message is the same: distributed gradient descent with many local steps can converge to the centralized solution in direction.
>
> ### $\textbf{Local-SGD extension of Section 4.}$
> In Section 4 we only analyzed deterministic Local-GD since we can rely on the Parallel Projection Method to help establish a clear results. Considering Local-SGD brings stochastic noise to the parallel projection, which is not even clear in the literature about convex projection. Unlike analysis in Section 3 of exact Local-GD, the extension is not direct and needs further exploration, which can be good future work.

---

### Review · Reviewer_ASZj · 2026-04-18

**Summary Of Contributions:**

This paper analyzes the implicit bias of local gradient descent for binary classification with linearly separable data. The main result shows that Local GD converges to the maximum margin classifier of the global dataset (which is the same implicit bias as GD on the global dataset), and provides asymptotic rates of convergence for the parameter direction and for the loss. This result is also extended to Local SGD.  The authors also analyze an idealized version of Local GD in which every communication rounds involve the exact solution of a regularized loss for each client, which is intended as a stand-in for Local GD with a large number of local steps.

Strengths:

**1**. The paper considers an important problem: binary classification is a fundamental problem in machine learning, and I'm happy to see more theoretical work at the intersection of distributed optimization and classification problems.

**2**. Characterizing the implicit bias of Local GD is a solid contribution.  Understanding the implicit bias of Local GD is an important part of understanding the algorithm itself, so I think the authors have filled a gap in the literature with these results.

Weaknesses:

**1**. There are major issues with the interpretation of Theorem 2 and the corresponding claims of contributions, which stem from a questionable use of big $\mathcal{O}$ notation. The central issue is that Theorem 2 does not actually provide a finite-time guarantee for approximately minimizing the loss: Theorem 2 shows that the loss goes to zero in the limit as the number of iterations goes to infinity, but it cannot provide any bound on the number of iterations required to make the loss small.

The bound on the loss from Theorem 2 is stated fully in the equation just before Equation (91). You can see that going from this line to Equation (91) introduces a big $\mathcal{O}$, but importantly, there is no bound provided on the time at which the big $\mathcal{O}$ rate "kicks in". It is very possible that this time depends on $\eta$ and $L$, and we need to know this dependence in order to prove that a particular choice of $\eta$ and $L$ can make the loss small in a finite number of steps.

What this means for their results: the convergence rate $f(w_0^k) \leq \mathcal{O}(1/Lk)$ does not "kick in" until some unspecified time which may depend on $L$, so there is simply no finite-time guarantee here. The authors say (on page 6, paragraph "Impact of local steps") that they can achieve $\epsilon^{-1/2}$ complexity as a consequence of their $\mathcal{O}(1/Lk)$ rate by choosing $L = \Theta(\epsilon^{-1/2})$, but by the same logic, we can achieve $\epsilon^{-p}$ for any $p \in (0, 1]$ by choosing $L = \Theta(\epsilon^{p-1})$. The same logic can go even further to say that the loss can be made smaller than $\epsilon$ after a constant number of iterations by choosing $L = \Theta(\epsilon^{-1})$. These are clearly not valid consequences of the $\mathcal{O}(1/Lk)$ rate.

The authors try to sidestep this issue by saying (on page 6, paragraph "Impact of local steps") that both their Theorem 2 and the prior work (Crawshaw et al, 2025) require that the number of rounds reach some threshold $\bar{k}$ after which the asympotic rate kicks in. This is a false equivalence. The critical difference between this work and that of (Crawshaw et al, 2025), is that in their theorem they actually bound the threshold $\bar{k}$ in terms of the parameters $\eta, L$, so they can indeed show a bound on the number of steps untill the loss is small. The authors continue by saying that they simply assume $\epsilon$ is small enough that the number of rounds $\epsilon^{-1/2}$ is much larger than this $\bar{k}$, but again this is just ignoring the dependence on the number of local steps: If we choose $L$ in terms of $\epsilon$, then $\bar{k}$ depends on $\epsilon$ through $L$, so you cannot just ignore $\bar{k}$.

So several claims made by the authors are incorrect. The complexity is not $\epsilon^{-1/2}$, in fact there is no complexity result here at all. The claims made by the authors are at worst very misleading, and at best a fundamental misunderstanding of big $\mathcal{O}$ notation and finite-time complexity.

Lastly, I have serious doubts about not just the interpretation, but the correctness of the rate $f(w_0^k) \leq \mathcal{O}(1/Lk)$. It was already shown in [1] that after a certain number of steps, Local GD with stepsize $\eta$ and local steps $L$ is nearly equivalent to GD with stepsize $\eta L$, for which the convergence rate is $f(w_0^k) \leq \mathcal{O}(1/(\eta L k))$. Given that the authors require $\eta \leq \mathcal{O}(1/L)$, the resulting convergence rate should actually be $\mathcal{O}(1/k)$, i.e. local steps has no effect here. I do not have more time to continue digging into their proof and find mistakes, so I cannot pinpoint the cause of this discrepancy, though it likely originates from usage of $\mathcal{O}$ to omit some occurrences of $L$ and not others. Based on the approximate equivalence with GD, I feel confident that the requirement $\eta \leq \mathcal{O}(1/L)$ must eliminate any possible acceleration from local steps, and therefore I seriously doubt the convergence rate $f(w_0^k) \leq \mathcal{O}(1/Lk)$ is correct.

**2**. The paper contains several inaccurate descriptions of their contributions, especially in comparison with prior work.

**2a**. Section 4 analyzes an idealized version of Local GD that does not include any parameters corresponding to learning rate or local steps, yet it is described as analyzing Local GD with local steps independent from the learning rate. That is not an accurate description. I think the authors should edit their claimed contributions and description to avoid referring to this result as analyzing Local GD with a large number of local steps.

**2b**. The contributions in the intro states that the work is "Providing a theoretical explanation to the phenomenon that Local-GD can work well with a very large number of local steps in practice". In fact, more recent papers on Local GD (which were not cited by the authors) already show that Local GD can converge with an arbitrary number of local steps: [1] showed this with any $\eta$ and any $L$ for logistic regression, and [2] showed this for general convex, smooth functions as long as one uses $\eta \leq \mathcal{O}(1/L)$ (which is also used by the authors in this work). Similarly, in the paragraph "Practical implications", the authors state that the existing analyses of Local GD require the number of local steps to be not too large.

**2c**. It is stated in the introduction (paragraph "Comparisons") that Theorem 2 recovers the rate of (Crawshaw et al, 2025) as a direct corollary. As I discussed in my Weakness 1, Theorem 2 does not recover the rate of two-stage Local GD as a corollary; this claim is based on incorrect interpretation of big $\mathcal{O}$ notation and a possibly incorrect convergence rate in Theorem 2. Similarly, it is stated in "Contributions" that this work shows "the local steps can benefit the convergence rate", which again is unjustified.

**2d**. The authors are missing a citation to [1], which is a very closely related work. [1] analyzes Local GD for logistic regression and already proves convergence with a learning rate independent of the number of local steps, and does not require any modifications such as regularization or exact solution of intermediate problems as in this submission's Section 4.

[1] Michael Crawshaw, Blake Woodworth, & Mingrui Liu (2025). Constant Stepsize Local GD for Logistic Regression: Acceleration by Instability. In Forty-second International Conference on Machine Learning.

[2] Woodworth, Blake E., Kumar Kshitij Patel, and Nati Srebro. "Minibatch vs local sgd for heterogeneous distributed learning." Advances in Neural Information Processing Systems 33 (2020): 6281-6292.

**Additional Comments:**

None.

**Audience:**

Yes

**Audience Explanation:**

Yes, I think that the convergence rate and implicit bias of Local GD is definitely of interest for people working in distributed optimization, optimization theory, or ML theory in general.

**Broader Impact Concerns:**

I have no concerns about the broader impact of this work; it is largely theoretical and fits naturally into the existing research landscape.

**Claims And Evidence:**

No

**Claims Explanation:**

The claims of improved convergence rates of the loss are not justified, as I explained in my weakness #1. Certain statements of contribution and comparisons with prior work are also unsupported, sometimes missing citations and other times using "optimistic" language to describe the results, as I explained in my weakness #2.

**Requested Changes:**

1. The interpretation and claimed contributions of Theorem 2 need to be amended, as the theorem does not provide any finite-time complexity result. This is critical for acceptance for me. The authors need to remove or amend several claims:
- Page 2, paragraph "Comparisons": Theorem 2 recovers the accelerated rate of (Crawshaw et al, 2025) as a direct corollary.
- Page 2, paragraph "Contributions": Theorem 2 shows that local steps can benefit the convergence rate.
- Page 6, paragraph "Impact of local steps": Theorem 2 achieves $\epsilon^{-1/2}$ complexity as a consequence of their $\mathcal{O}(1/Lk)$ rate by choosing $L = \Theta(\epsilon^{-1/2})$.
- Page 6, paragraph "Impact of local steps": The $\bar{k}$ requirement of Theorem 2 is the same as in (Crawshaw et al, 2025).
- Anywhere in the paper that refers to Theorem 2 as providing a complexity result for minimizing the loss.
2. Several other claims of contribution and comparison with previous work need to be amended. All here are critical for me except the one about Section 4, which I leave as a suggestion:
- Page 2, contributions: This work is "providing a theoretical explanation to the phenomenon that Local-GD can work well with a very large number of local steps in practice".
- Page 2, practical implications: "In the existing convergence analysis of Local-GD, the number of local steps $L$ should not be very large for heterogeneous data".
- Section 4 should not be described as analyzing Local GD with a learning rate independent of $L$. I am fine with the authors drawing an analogy between the algorithm they consider and Local GD with a learning rate independent of $L$, but it must be treated as what it is: an analogy.
- The authors must cite and discuss [1] (which analyzes Local GD for logistic regression with any learning rate and any local steps) and [2] (which analyzes Local GD for smooth, convex, heterogeneous objectives without requiring a bound on the number of local steps).
3. The authors need to explain how they can possibly achieve a convergence rate of $f(w_0^k) \leq \mathcal{O}(1/Lk)$ when the learning rate must obey $\eta \leq \mathcal{O}(1/L)$. I have serious doubts about the correctness of this rate (see the last paragraph of my weakness \#1). Given that this is a matter of technical correctness, this is critical for acceptance.

---

> ### Author Response · Authors · 2026-05-04
>
> We sincerely thank the reviewer for the careful reading and constructive comments. We are very happy to discuss the technical details with the reviewer. Please kindly find our responses below.
>
> ### $\textbf{Loss convergence rate $1/Lk$.}$
> First of all, we would like to note all of our results are \emph{asymptotic} convergence when number of rounds $k$ is sufficiently large. Under this regime we can derive the implicit bias of Local-GD and, based on the implicit bias result, the loss convergence rate $\mathcal{O}(1/Lk)$. It is not a non-asymptotic rate that in general holds, and we emphasize this point in the revised version to avoid misleading. On top of that, we are very willing to discuss the rate with the reviewer.
>
> 1. When $k$ is large enough so that our implicit bias results hold, we derive the loss convergence rate in detail in Appendix C.3. We do not hide any information in the big O notation. The derivation from (88) to (91) ((111) to (114) in the revised version) is completely presented in the proof and we clearly show why there is $O((Lk)^{- \max (\theta, 1+ \mu_+)})$ order by combining (112) and (113). The first line of (114) ((91) in the original version) is just a copy of (111) and we split it into two parts in (112) and (113).
>
> 2. The remaining issue is how large the $k$ should be. This is hidden in the proof of Lemma 9. So in the revised version we have added the full proof of Lemma 9 in Appendix C2.2, although it is just an adaption of Lemma 6 in Nacson et al. (2019). In the proof the $k$ should be larger than $\tilde{k}$, which is the maximum value satisfying conditions (94)-(98). At the end of C2.2 we clearly show the $\tilde{k}$ is independent of number of local steps $L$. Nevertheless, we still note that $\tilde{k}$ can be a large value so that it is just an asymptotic result. But it does not impact the $\mathcal{O}(1/Lk)$ rate under this asymptotic regime.
>
> 3. Comparison with (Crawshaw et al. 2025b). Both (Crawshaw et al. 2025b) and (Crawshaw et al. 2025a) are attempts to extend the analysis of GD on logistic regression with large step sizes, from (Wu et al., 2024) to the distributed case. Note that these large step sizes do not come for free, either in the centralized case (Wu et al. 2024) or in the distributed case (Crawshaw et al. 2025b) and (Crawshaw et al. 2025a). GD or Local-GD on logistic regression with large step sizes requires several warm-up steps/rounds, so as to handle the initial instability due to the step size. The number of warm-up rounds is proportional to the step-size, see Theorem 1 in (Wu et al. 2024), the lower bound on $R$ in Corollary 4.3 in (Crawshaw et al. 2025b) and Theorem 1 in (Crawshaw et al. 2025a). In contrast, our analysis does not focus on the large step sizes, and thus does not pay the price of this warm-up due to instability. Instead, the only lower bounds required for the number of rounds in our results is for the asymptotics to kick in. This resembles the case of using small learning rate in logistic regression in (Soudry et al 2018). We quantify this lower bound above.
> Finally, the convergence rate of Local-GD on logistic regression with large step-sizes in (Crawshaw et al. 2025b) after the instability phase is $\mathcal{O}((\eta L k)^{-1}$, which required them to use large step-sizes. Our rates are $\mathcal{O}((Lk)^{-1})$ without the term of $\eta$, thus allowing us to use even small step-sizes. (Crawshaw et al. 2025b) use this rate to set $L$ appropriately to recover an accelerated rate, which we can also do. However, due to our small step sizes, we never pay the price of the additional initial unstable rounds, thus allowing more flexibility in our choice of $L$. Indeed, we need $k$ to be large to enter the asymptotic implicit bias stage, so we still say it is an asymptotic rate, but independent of $L$. We have added the comparison at Introduction in the revised version.
>
> (Crawshaw et al. 2025a) Michael Crawshaw, Blake Woodworth, and Mingrui Liu. Local steps speed up local GD for heterogeneous distributed logistic regression. In The Thirteenth International Conference on Learning Representations, 2025.
>
> (Crawshaw et al. 2025b) Michael Crawshaw, Blake Woodworth, and Mingrui Liu (2025). Constant Stepsize Local GD for Logistic Regression: Acceleration by Instability. In Forty-second International Conference on Machine Learning.
>
> (Wu et al. 2024) Wu, J., Bartlett, P.L., Telgarsky, M. and Yu, B., 2024, June. Large stepsize gradient descent for logistic loss: Non-monotonicity of the loss improves optimization efficiency. In The Thirty Seventh Annual Conference on Learning Theory (pp. 5019-5073). PMLR.
>
> (Soudry et al 2018) Daniel Soudry, Elad Hoffer, Mor Shpigel Nacson, Suriya Gunasekar, and Nathan Srebro. The implicit bias of gradient descent on separable data. The Journal of Machine Learning Research, 19(1):2822–2878, 2018.

---

> ### Author Response · Authors · 2026-05-04
>
> 4. Comparison with (Woodworth et al. 2020). Note that the lower bounds for convex and smooth objectives in Theorems 2 and 4 of (Woodworth et al. 2020) do not apply to the problem of logistic regression. The reason for this is the term $B$, which is an upper bound on the $\ell_2$ norm of the optimum. For logistic regression, the optimum is achieved at $||w^*||$ to $\infty$, hence all the bounds in Theorems 2 and 4 don't yield any non-trivial information about logistic regression. Further, in the overparametrized setting, all the datapoints can share an optimum, so the terms of noise at optimum,  $\zeta^\star=\sigma_{\star, m} =0$ in (Woodworth et al. 2020). This further makes their bounds invalid for the problem of overparametrized logistic regression. We have added the discussion at Related Works.
>
> (Woodworth et al. 2020) Woodworth, Blake E., Kumar Kshitij Patel, and Nati Srebro. "Minibatch vs local sgd for heterogeneous distributed learning." Advances in Neural Information Processing Systems 33 (2020): 6281-6292.
>
> ### $\textbf{Claim of Section 4.}$
> Thanks for the suggestion. We have revised the title of Section 4 and the contribution of this part in the Introduction to avoid misleading. In Section 4 we actually analyzed the exactly solved local problem in distributed training, which is an analogy to Local-GD. We now emphasize this point in the main text. But the main message is the same: distributed gradient descent with many local steps can converge to the centralized solution in direction.
>
> ### $\textbf{Changes of claims.}$
> Based on the explanation of asymptotic convergence rate above and the reviewer's requested changes, we have corrected several claims in the revised version as the reviewer requested. All changes are marked as blue color in the revised version.
>
> We thank again for the reviewer's detailed suggestions and would love to discuss if the reviewer has further questions.

---

> > ### Comment · Reviewer_ASZj · 2026-05-04
> >
> > I appreciate the efforts of the authors to amend their paper. The new version is an improvement in terms of the language: it is now clearly communicated that the results are asymptotic rates rather than finite-time complexity results. However, I still have serious doubts about the correctness of the $\mathcal{O}(1/Lk)$ rate of Theorem 2; in fact below I included a lower bound that contradicts the rate of Theorem 2. The lower bound shows that there can be no benefit to the convergence rate (even asymptotically) as long as $\eta \leq \mathcal{O}(1/L)$, which is required in Theorem 2. I conclude that the rate of Theorem 2 is incorrect, and therefore many claims in the paper about the impact of local steps are not justified. This is a major issue with the paper that prevents my recommendation of acceptance. See the proof below, which is short and easily checked.
> >
> > Let $\tau$ be the first communication round where $f(w_{\tau}) \leq \min(\gamma/(70 \eta LM), 5/(12 \eta L))$. We will derive a lower bound for all $k \geq \tau$.
> >
> > The idea of the proof is as I mentioned in my review: When the loss is sufficiently small, (Crawshaw et al, 2025b) already showed an approximate equivalence between (1) Local GD with stepsize $\eta$ and interval $L$, and (2) GD with stepsize $\eta L$. And we can just apply convexity and a few properties of the logistic loss to lower bound the loss decrease for each individual step. To make this precise, recall that $w_k$ are the iterates of Local GD, and define $$
> >     b_k = \frac{w_{k+1} - w_k}{-\eta L} - \nabla f(w_k),
> > $$ so that $w_{k+1} = w_k - \eta L (\nabla f(w_k) + b_k)$. So $b_k$ is the approximation error of the previously stated equivalence. There are a few key properties of the trajectory that we will use:
> > - $f(w_{k+1}) \leq f(w_k)$ for all $k \geq \tau$. This is immediate given the assumed upper bound on $\eta$ from Theorem 2, but it also comes from Eq (34) of (Crawshaw et al, 2025b) given our bound on $f(w_\tau)$.
> > - From the previous bullet and the definition of $\tau$, we have $f(w_k) \leq \min(\gamma/(70 \eta LM), 5/(12 \eta L))$ for all $k \geq \tau$.
> > - $\lVert \nabla f(w) \rVert \leq f(w)$ for all $w$ (Lemma 25 from (Crawshaw et al, 2025a).
> >
> > We can now lower bound the loss decrease at each step using convexity: $$
> >     f(w_{k+1}) \geq f(w_k) + \langle \nabla f(w_k), w_{k+1} - w_k \rangle = f(w_k) - \eta L \lVert \nabla f(w_k) \rVert^2 - \eta L \langle \nabla f(w_k), b_k \rangle
> > $$ $$
> >     f(w_{k+1}) \geq f(w_k) - \eta L \lVert \nabla f(w_k) \rVert^2 - \eta L \lVert \nabla f(w_k) \rVert  \lVert b_k \rVert \geq f(w_k) - \frac{6}{5} \eta L \lVert \nabla f(w_k) \rVert^2,
> > $$ where the last inequality uses that $\lVert b_k \rVert \leq \frac{1}{5} \lVert \nabla f(w_k) \rVert$ when $f(w_k) \leq \gamma/(70 \eta LM)$. We can also apply the third bullet to get $$
> >     f(w_{k+1}) \geq f(w_k) - \frac{6}{5} \eta L f(w_k)^2.
> > $$ We will manipulate and unroll this recursion, but first we collect a fact for later. Note that $$
> >     f(w_{k+1}) \geq \left( 1 - \frac{6}{5} \eta L f(w_k) \right) f(w_k) \geq \frac{1}{2} f(w_k),
> > $$ where we used the second bullet above. Now to unroll the recursion, divide both sides by $f(w_{k+1}) \cdot f(w_k)$ and rearrange: $$
> >     \frac{1}{f(w_{k+1})} \leq \frac{1}{f(w_k)} + \frac{6}{5} \eta L \frac{f(w_k)}{f(w_{k+1})} \leq \frac{1}{f(w_k)} + \frac{12}{5} \eta L,
> > $$ where the last inequality uses $f(w_{k+1}) \geq \frac{1}{2} f(w_k)$ from above. We can now unroll the recursion back to $\tau$ to get $$
> >     \frac{1}{f(w_k)} \leq \frac{1}{f(w_\tau)} + \frac{12}{5} \eta L (k - \tau),
> > $$ or $$
> >     f(w_k) \geq \frac{1}{1/f(w_\tau) + (12/5) \eta L (k - \tau)} \geq \frac{5}{12 \eta L (k - \tau)} \geq \frac{5}{12 \eta Lk}.
> > $$ This inequality matches exactly my claim from my review, that the convergence rate of Local GD for this problem cannot be any better than $f(w_k) \geq \Omega(1/(\eta Lk))$. Since Theorem 2 from this submission requires that $\eta \leq \mathcal{O}(1/L)$, this means that the true rate is $f(w_k) \geq \Omega(1/k)$, so there is no benefit of local steps, even to the asymptotic convergence rate.

---

> > > ### Author Response · Authors · 2026-05-12
> > >
> > > We are very grateful to the reviewer's detailed comments. After careful check of our proofs we finally found one ignored item in the rate of loss convergence. After correction, the loss convergence rate in our analysis should be $\mathcal{O}(1/\eta Lk)$, which matches the reviewer's provided lower bound. Since the required learning rate is $\mathcal{O}(1/L)$, the final rate is $\mathcal{O}(1/k)$. Below we briefly point out the ignored term in the analysis and then correct contributions of this work.
> > >
> > > ### The ignored term and final rate
> > >
> > > In the proof of loss convergence rate (Appendix C.3 Proof of Claim 3), everything is correct until we get the expression in (116) in current version:
> > > \begin{align}
> > >     f(w_0^k) \leq \left[\frac{1}{MLk} \sum_{s \in V} \exp(-x_s^T \rho^k)\right] + \mathcal{O}((Lk)^{- \max (\theta, 1+ \mu_+)}).
> > > \end{align}
> > > The issue is not the big O expression, but the $\exp(-x_s^T \rho^k)$ term. Recall that the residual term is
> > > \begin{align}
> > >     \rho^k = \tilde{w} + \frac{M}{L} \frac{1}{M} \sum_{i=1}^M m_i(k,L) + r_0^k.
> > > \end{align}
> > > Before we focused on $r_0^k$ and checked that it is $\mathcal{O}(1)$ on $L$. However, we ignored the term $\tilde{w}$, which is the solution of
> > > \begin{align}
> > >     \alpha_s = \eta \exp(- x_s^T \tilde{w} ) \quad \forall s \in V.
> > > \end{align}
> > > It is actually related to learning rate $\eta$. Plugging in this equation, we can finally get
> > > \begin{align}
> > >     f(w_0^k) \leq & \left[\frac{1}{MLk\eta} \sum_{s \in V} \exp[-x_s^T (m(k+1,0) + r_0^k)]\right] + \mathcal{O}((Lk)^{- \max (\theta, 1+ \mu_+)}).
> > > \end{align}
> > > Since $m(k+1,0) + r_0^k$ is independent of $L$ and bounded when $k \to \infty$, the final rate of loss function $f(w_0^k)$ is $\mathcal{O}(1/Lk\eta)$.
> > >
> > > We have corrected the proof in the revised version and every place the rate is mentioned. Also, the directional convergence rate is $\left \| \frac{w_0^k}{\|w_0^k\|} - \frac{\hat{w}}{\|\hat{w}\|} \right \| = \mathcal{O}\left (\frac{1}{\log(\eta Lk)} \right)$.

---

> > > > ### Author Response · Authors · 2026-05-12
> > > >
> > > > ### Contributions of this work
> > > >
> > > > Thanks to the reviewer's comments, we are motivated to discuss further about the contribution of this work and its comparison to (Crawshaw et al. 2025b), which is perhaps the closest paper to our work.
> > > >
> > > >
> > > > Firstly, we would like to emphasize that the core question of this work is which solution the aggregated model trained by Local-GD would converge to in {\em overparameterized} regime, as claimed in Introduction. Our error in the convergence rates does not impact the implicit bias results in our work. In the overparametrized regime, several models can achieve $0$ loss, only one of which is the centralized model. Existing works (Crawshaw et al. 2025b), characterize convergence in loss, but they don't characterize the implicit bias of Local-GD.
> > > >
> > > >
> > > > Secondly, we show that for small learning rate, $\eta = \mathcal{O}(1/L)$, Local-GD can finally converge to the centralized model. This utilizes the intuition that the reviewer mentioned previously, that $\mathcal{O}(1/L)$ learning rate makes every round a ``large GD step''. While this small step-size guarantees that we converge to the centralized model, it forces the convergence to be slower, at a rate dependent on $k$, the number of rounds.
> > > >
> > > >
> > > > Thirdly, for large learning rate, it is not necessary that Local-GD converges to the centralized model. In this work we considered a special case where a local problem with exponential loss is exactly solved. It is not vanilla Local-GD algorithm but it includes the case of infinite local steps with larger step size independent of $L$. We conclude even with exactly solved local problem, the vanilla average aggregation is not guaranteed to converge to centralized model. Only with the proposed Modified Aggregation we can ensure the same implicit bias result as centralized GD. In this case, we provide a concrete counter-example, where the average of local max-margin solutions, which would correspond to the output of Local-GD,  is not the global max-margin solution.
> > > >
> > > > Counter-Example : Consider the dimension is $d>4$. Suppose $e_1$ and $e_2$ are unit vectors with only 1 non-zero entry. For compute node 1, the data samples are $(x=e_1, y=1)$ and $(x=-e_1, y=-1)$. For compute node 2, the data samples are $(x= 0.5 (e_1 + e_2), y=1)$ and $x=(-0.5(e_1 + e_2), y=-1)$. Thus the max-margin unit-norm classifier of compute node 1 is $e_1$, and that of compute node 2 is $(e_1 + e_2)/\sqrt{2}$. Their average is not equal to the max-margin classifier of the centralized solution, which is $(e_1 + e_2)/\sqrt{2}$, as the centralized solution's support vectors are only from compute node 2 and not from compute node 1.
> > > >
> > > > Finally, we believe that studying the implicit bias of vanilla Local-GD with large learning rates, is a promising direction of future work, especially in light of existing results where even gradient descent can diverge catastrophically under the exponential loss for large learning rates (Theorem 4.2 in (Wu et al 2023)).
> > > >
> > > > Since the above discussion and comparisons are important, we added a new section after all the theoretical results (Section 5 in the revised version) and moved the comparison to (Crawshaw et al. 2025b) there. Please check there for the detailed discussions. Hopefully we have reached the agreement on the contributions of this work now.
> > > >
> > > > (Wu et al 2023) Wu, Jingfeng, Vladimir Braverman, and Jason D. Lee. "Implicit bias of gradient descent for logistic regression at the edge of stability." Advances in Neural Information Processing Systems 36 (2023): 74229-74256.

---

### Author Response · Authors · 2026-05-04

We thank the AE and reviewers for their careful reading, constructive comments, and valuable suggestions. We have revised the manuscript accordingly, with all changes marked in blue. Below, we provide point-by-point responses to the reviewers’ comments. We would be happy to address any further questions or clarifications the reviewers may have.

---

### Decision · Action_Editor_SYGJ · 2026-06-01

**Recommendation:** Reject

**Audience:**

Yes

**Audience Explanation:**

The proof that local GD on globally linearly separable data converges to the maximum margin solution for logistic regression will be of interest to at least part of TMLR's audience, as is the analysis of the implicit bias of exactly solved local problems.

**Claims And Evidence:**

No

**Claims Explanation:**

Despite a robust discussion between the authors and reviewer ASZj, who really went above and beyond the call of duty by providing a lower-bound proof contradicting Theorem 2 in the original manuscript, which led the authors to find an error in their proofs, this paper still has problems with the way it states results.

1. By the logic in the authors' comment (https://openreview.net/forum?id=nqG7naNjBB&noteId=tQ3uEQ2p5G) the statement of Theorem 2 (and also Theorem 5) is incorrect. It states that $w_{0}^{k} = \\log(Lk)\\hat{w} + \rho^{k}$ but the correct statement is $w_{0}^{k} = \\log(\\eta Lk)\\hat{w} + \rho^{k}$.
2. Theorem 9 has not been corrected at all.
3. The comparison of the results in this manuscript to those in Crawshaw, Woodworth, and Liu, "Constant stepsize local GD for logistic regression: Acceleration by instability," in Proc. ICML, 2025 in Section 3.2 under "Impact of Local Steps" is misleading because the proof of Theorem 2 requires that $\\eta$ be upper bounded by a term proportional to $1/L$ while Crawshaw, Woodworth, and Liu analyze local GD with a step size independent of $L$.

Beyond these specific concerns, my impression is that the authors have attempted to make *de minimis* changes to the manuscript in response to the reviewers' feedback when far more extensive changes were required given the errors in the proofs in the original version.

**Resubmission Of Major Revision:**

The authors may consider submitting a major revision at a later time.